# Polar stratospheric cloud climatology based on CALIPSO spaceborne lidar measurements from 2006-2017

Michael C. Pitts[1], Lamont R. Poole[2], and Ryan Gonzalez[3, 4]

[1]NASA Langley Research Center, Hampton, Virginia, 23681, USA
[2]Science Systems and Applications, Inc., Hampton, Virginia, 23666, USA
[3]Universities Space Research Association, NASA Langley Research Center, Hampton, VA, 23681 USA
[4]Now at Department of Atmospheric Science, Colorado State University, Fort Collins, CO, 80523 USA

*Correspondence to*: Michael C. Pitts (michael.c.pitts@nasa.gov)

**Abstract.**   The Cloud-Aerosol Lidar with Orthogonal Polarization (CALIOP) on the CALIPSO (Cloud-Aerosol Lidar and Infrared Pathfinder Satellite Observations) satellite has been observing polar stratospheric clouds (PSCs) from mid-June 2006 until the present. The spaceborne lidar profiles PSCs with unprecedented spatial (5-km horizontal x 180-m vertical) resolution and its dual-polarization capability enables classification of PSCs according to composition.   Nearly coincident Aura Microwave Limb Sounder (MLS) measurements of the primary PSC condensables ($HNO_3$ and $H_2O$) provide additional
constraints on particle composition.  A new CALIOP Version 2 (v2) PSC detection and composition classification algorithm has been implemented that corrects known deficiencies in previous algorithms and includes additional refinements to improve composition discrimination.  Major v2 enhancements include dynamic adjustment of composition boundaries to account for effects of denitrification and dehydration, explicit use of measurement uncertainties, addition of composition confidence indices, and retrieval of particulate backscatter, which enables simplified estimates of particulate surface area density (SAD)
and volume density (VD).  The 11+ years of CALIOP PSC observations in each v2 composition class conform to their expected thermodynamic existence regimes, which is consistent with previous analyses of data from 2006-2011 and underscores the robustness of the v2 composition discrimination approach.

The v2 algorithm has been applied to the CALIOP dataset to produce a PSC reference data record spanning the 2006-2017 time period, which is the foundation for a new comprehensive, high resolution climatology of PSC occurrence and composition
for both the Antarctic and Arctic.  Time series of daily-averaged, vortex-wide PSC areal coverage versus altitude illustrate that Antarctic PSC seasons are similar from year to year, with about 25% relative standard deviation in Antarctic PSC spatial volume at the peak of the season in July and August.  Multi-year average, monthly zonal mean cross sections depict the climatological patterns of Antarctic PSC occurrence in latitude/altitude and also equivalent latitude/potential temperature coordinate systems, with the latter system better capturing the microphysical processes controlling PSC existence.  Polar maps
of the multi-year mean geographical patterns in PSC occurrence frequency show a climatological maximum between longitudes 90° W and 0°, which is the preferential region for forcing by orography and upper tropospheric anticyclones.  The

climatological mean distributions of particulate SAD and VD also show maxima in this region due to the large enhancements from the frequent ice clouds.

Stronger wave activity in the Northern Hemisphere leads to a more disturbed Arctic polar vortex, whose evolution and lifetime vary significantly from year to year. Accordingly, Arctic PSC areal coverage is distinct from year to year with no "typical" year, and the relative standard deviation in Arctic PSC spatial volume is > 100% throughout most of the season. When PSCs are present in the Arctic, they most likely occur between longitudes 60°W and 90°E, which is consistent with the preferential location of the Arctic vortex.

Comparisons of CALIOP v2 and Michelson Interferometer for Passive Atmospheric Sounding (MIPAS) Antarctic PSC observations show excellent correspondence in the overall spatial and temporal evolution, as well as for different PSC composition classes. Climatological patterns of CALIOP v2 PSC occurrence frequency in the vicinity of McMurdo Station, Antarctica, and Ny-Ålesund, Spitsbergen, are similar in nature to those derived from local ground-based lidar measurements. To investigate the possibility of longer term trends, appropriately subsampled and averaged CALIOP v2 PSC observations from 2006-2017 were compared with PSC data during the 1978-1989 period obtained by the spaceborne solar occultation instrument SAM II (Stratospheric Aerosol Measurement II). There was good consistency between the two instruments in column Antarctic PSC occurrence frequency, suggesting that there has been no long-term trend. There was less overall consistency between the Arctic records, but it is very likely due to the high degree of interannual variability in PSCs rather than a long-term trend.

## 1 Introduction

The overall role of polar stratospheric clouds (PSCs) in the depletion of stratospheric ozone is well established (Solomon, 1999). Heterogeneous reactions on PSC particles convert the stable chlorine reservoirs HCl and $ClONO_2$ to chlorine radicals that destroy ozone catalytically (Solomon et al., 1986; Crutzen et al., 1992; Solomon et al., 1999). Rates of these reactions depend on particle surface area density (SAD) and composition, which can include binary ($H_2SO_4/H_2O$) or ternary ($HNO_3/H_2SO_4/H_2O$, or STS) liquid droplets; solid nitric acid trihydrate (NAT) particles; and $H_2O$ ice particles (Lowe and MacKenzie, 2008). PSCs also impact polar ozone chemistry by temporarily removing gas-phase $HNO_3$ from the polar stratosphere through uptake by the particles during formation and growth (denoxification). In addition, sedimentation of large NAT particles (Fahey et al., 2001; Northway et al., 2002; Molleker et al., 2014) can permanently remove $HNO_3$ (denitrification), which prolongs ozone depletion by delaying reformation of the stable chlorine reservoirs. A substantial recovery of the ozone layer is expected by the middle of this century with reduced global production of ozone depleting substances in accordance with the Montreal Protocol and subsequent amendments and adjustments (WMO, 2015). But as climate changes, leading to a colder and perhaps wetter stratosphere and upper troposphere (e.g., Shindell, 2001), reliable model predictions of recovery of the Antarctic ozone hole and of potentially more severe ozone depletion in the Arctic are

challenging because many global models use simple parameterizations that do not accurately represent PSC processes (e.g. Peter and Grooß, 2012; Morgenstern et al., 2017).

Fortunately, our knowledge of the temporal and geographic distribution of PSCs and their particle composition has expanded greatly in the 21st century with the advent of three satellite instruments with extensive polar measurement coverage: the Michelson Interferometer for Passive Atmospheric Sounding (MIPAS) on Envisat (2002-2012), the Microwave Limb Sounder (MLS) on Aura (2004-present), and the Cloud-Aerosol Lidar with Orthogonal Polarization (CALIOP) on CALIPSO (Cloud-Aerosol Lidar and Infrared Pathfinder Satellite Observations, 2006-present). CALIPSO flies in a 98° inclination orbit at an altitude of 705 km as part of the NASA A-train satellite constellation (Stephens et al., 2002), along with the Aqua, Aura, CloudSat, and Orbiting Carbon Observatory-2 (OCO-2) satellites. Although PSC studies are not a primary mission objective, CALIPSO is an ideal platform for studying polar processes, collecting data along 14-15 orbits per day with coverage from 82° S to 82° N latitude on each orbit. CALIOP data collection began in mid-June 2006 and continues at the time of this writing.

The foundation for PSC detection and composition classification using CALIOP data was laid out in papers by Pitts et al. (2007) and Pitts et al. (2009), with additional refinements appearing in Pitts et al. (2011) and Pitts et al. (2013). We will refer to these papers herein as P07, P09, P11, and P13, respectively, and refer to the P07/P09/P11/P13 algorithm sequence as Version 1.0 (v1). Data products from v1 have compared favorably with PSC observations from MIPAS (Höpfner et al., 2009), ground-based lidar (Achtert and Tesche, 2012), and Aura MLS (P13; Lambert et al, 2016). However, several known deficiencies in the v1 algorithm were highlighted in P13, one being that the effects of measurement noise on the inferred PSC composition were not explicitly considered. In addition, the boundary separating NAT and ice PSCs in the CALIOP optical measurement space needed to be adjusted in the event of denitrification and dehydration, a shortcoming that was also noted in comparisons of v1 products with model simulations by Zhu et al. (2017).

In the present paper, we introduce the CALIOP Version 2.0 (v2) PSC algorithm, which addresses these known deficiencies and includes additional refinements to increase the robustness of the inferred PSC composition. These refinements include: (1) correction for crosstalk between the CALIOP parallel and perpendicular polarization channels; (2) estimation of random uncertainties in the measured and derived optical quantities using the noise-scale factor (NSF) approach (Liu et al., 2006); (3) adoption of less conservative PSC detection thresholds to better match features detected by the naked eye in CALIOP orbital curtain images; (4) redefined PSC composition classes with indices denoting statistical confidence in the inferred composition; and (5) retrieval of 532-nm particulate backscatter, which corrects the CALIOP measurements for attenuation by overlying particle layers and enables simplified estimates of the bulk particle microphysical quantities SAD and volume density (VD) to facilitate comparisons with PSC measurements by other instruments as well as with theoretical model representations of heterogeneous chemical processes. We show several examples that illustrate the top-level differences between v1 and v2 data products. We then present a state-of-the-art PSC reference data record and climatology constructed by applying the v2 algorithm to the 11+ year CALIOP spaceborne lidar dataset spanning 2006-2017. This work is part of a larger effort to compile

new reference PSC climatologies based on the contemporary CALIOP, MIPAS, and MLS datasets that is being performed under the auspices of the Stratospheric-tropospheric Processes and their Role in Climate (SPARC) Polar Stratospheric Cloud initiative (PSCi: http://www.sparc-climate.org/activities/polar-stratospheric-clouds/). A separate MIPAS PSC climatology has been compiled by Spang et al. (2018). These new climatologies represent the first observational-based records of PSC occurrence, composition, and particle characteristics on vortex-wide spatial scales covering decadal time scales and are a valuable resource for testing and validating current and future global models.

In Section 2 we describe the datasets utilized in the CALIOP v2 PSC algorithm and in constructing the CALIOP PSC climatology. In Section 3 we describe in detail the modifications implemented in the v2 algorithm, illustrate the top-level differences between v1 and v2, and show that the v2 PSC composition classes conform well to their expected temperature existence regimes. In Section 4 we present the CALIOP v2 PSC reference data record and climatology in terms of overall and composition-specific areal coverage and occurrence frequency. Multi-year average, monthly zonal mean cross sections in both latitude/altitude and equivalent latitude/potential temperature coordinate systems are shown along with polar maps that illustrate the detailed temporal and spatial patterns in PSC occurrence and composition. In Section 5 we show examples of the SAD climatology. In Section 6 we discuss comparisons of CALIOP with MIPAS and ground-based lidar PSC observations and investigate the possibility of long-term trends in PSC occurrence by comparing the CALIOP data record to the historical (1978-1989) SAM II (Stratospheric Aerosol Measurement II) solar occultation PSC occurrence record. Finally, in Section 7 we summarize the key findings and discuss the results.

## 2 Datasets

The A-Train satellite constellation offers a unique opportunity for PSC analyses through the combination of CALIOP data and nearly coincident Aura MLS measurements of the primary PSC condensable vapors, $HNO_3$ and $H_2O$. Additional context is provided by ancillary meteorological information from the Modern Era Retrospective-Analysis for Research (MERRA-2) reanalysis products (Gelaro et al., 2017) and the Aura MLS Derived Meteorological Products (DMPs) (Manney et al., 2007; Manney et al., 2011a). A brief description of these datasets is provided below.

### 2.1 CALIOP

CALIOP, the primary instrument on the CALIPSO satellite, is a dual wavelength polarization-sensitive lidar that provides high vertical resolution profiles of backscatter coefficients at 532 and 1064 nm (Winker et al., 2009). Figure 1 illustrates the typical CALIPSO orbital coverage for a single day (17 July 2008) over the Antarctic polar region. A curtain of CALIOP 532-nm total attenuated backscatter coefficient measurements along a single orbit from this day is shown in Fig. 2 and illustrates the unique capability of the CALIOP spaceborne lidar to probe clouds and aerosols at very high spatial resolution. Although not specifically designed for stratospheric applications, PSCs generally produce detectable enhancements in CALIOP backscatter

profiles as can be seen at altitudes above ~12 km along the orbit curtain in Fig. 2. The CALIOP measurements of 532-nm perpendicular backscatter coefficient provide additional information on particle shape, from which PSC composition can be inferred. The v2 CALIOP PSC data products are derived from night-time-only CALIOP v4.10 Level 1B 532-nm parallel and perpendicular backscatter coefficient measurements; daytime measurements contain elevated background noise due to scattered sunlight, which greatly inhibits the detection of PSCs. Ancillary meteorological data from MERRA-2, including temperature, pressure, ozone number density, and tropopause height at the CALIOP measurement locations are included in the CALIOP v4.10 Level 1B data products and utilized in the PSC algorithm as described in Section 3. Further details on the CALIPSO v4.10 Level 1 data processing and calibration approach can be found in Kar et al. (2018).

**2.2 Aura MLS**

The Aura satellite flies in formation with CALIPSO in the A-train satellite constellation providing a nearly coincident dataset of gas-phase $HNO_3$ and $H_2O$ from the MLS instrument with spatial and temporal differences between the CALIOP and MLS measurements less than 10 km and 30 s after a repositioning of the Aura satellite in April 2008 and about 200 km and 7-8 min. prior to 2008 (see Lambert et al., 2012). The Aura MLS detects thermal microwave emission from the Earth's limb along the line-of-sight in the forward direction of the Aura spacecraft flight track (Waters et al., 2006). Vertical scans are made from the Earth's surface up to a 90 km tangent height every 24.7 s, providing a total of 3500 vertical profiles per day with a horizontal along track spacing of 1.5 degrees (~165 km) and nearly global latitude coverage from 82° S–82° N. The limb radiance measurements are inverted using a 2-D optimal estimation retrieval (Livesey et al., 2006) to yield atmospheric profiles of temperature and gas-phase constituents in the vertical range 8–90 km. Herein we use the MLS version 4.2 products (Livesey et al., 2017). For the vertical range relevant for PSCs, the MLS version 4.2 measurements have typical single-profile precisions (accuracies) of 4-15% (4-7%) for $H_2O$ (Read et al. 2007; Lambert et al., 2007) and 0.6 ppbv (1-2 ppbv) for $HNO_3$ (Santee et al., 2007). Vertical and horizontal along-track resolutions are 3.1-3.5 km and 180-290 km for $H_2O$, and 3.5-5.5 km and 400-550 km for $HNO_3$.

The Aura MLS Version 2 (v2) derived meteorological products (DMPs) (Manney et al., 2007; Manney et al., 2011a) have been calculated from the MERRA-2 reanalyses and consist of meteorological variables (winds, temperature, potential temperature, potential vorticity) and derived fields (e.g., equivalent latitude and vortex edge location) interpolated to the locations and times of the Aura MLS observations.

To better facilitate the utilization of the MLS data in the PSC analyses, the MLS gas-phase $HNO_3$ and $H_2O$ measurements are interpolated to the CALIOP PSC orbit grid using an average of the two nearest MLS profiles weighted by the inverse of the relative distance of each to the CALIPSO location. In addition, meteorological parameters from the Aura MLS v2.0 DMPs are also mapped onto the CALIOP PSC orbit grid. The MLS $HNO_3$ and $H_2O$, and meteorological parameters from the DMPs are included in the archived v2 CALIOP PSC data product files.

## 3 CALIOP Version 2.0 PSC Algorithm

### 3.1 Data Pre-processing

Data pre-processing generally follows the procedure discussed in detail in P07 and P09, but with additional steps (see Appendix A) to correct for the small amount of crosstalk between the two CALIOP polarization channels and to estimate uncertainties
[$u(x)$] in the basic CALIOP measurements and derived quantities. The initial step is to ingest nighttime-only profiles of CALIOP V4.10 Lidar Level 1B 532-nm attenuated parallel ($\beta'_\parallel$) and perpendicular ($\beta'_\perp$) backscatter coefficients over the altitude range 8.4-30 km. The data are smoothed to a uniform 5-km horizontal (along the orbit track) by 180-m vertical resolution grid to remove the altitude dependence of the resolution of the downlinked CALIOP data (Winker et al., 2007). The data are then corrected for molecular and ozone attenuation using the MERRA-2 molecular and ozone number density profiles
reported in the CALIOP V4.10 Level 1B data product files. The MERRA-2 molecular number density is also used in the theoretical relationship from Hostetler et al. (2006) to calculate molecular backscatter $\beta_{mol}$, which is then used to calculate the 532-nm attenuated scattering ratio

$$R'_{532} = (\beta'_\parallel + \beta'_\perp) / \beta_{mol}. \tag{1}$$

### 3.2 PSC Detection

Detection of PSCs in the v2 algorithm generally follows the approach of v1 in that PSCs are detected as statistical outliers in either $\beta'_\perp$ or $R'_{532}$ relative to the background stratospheric aerosol population. We also use successive horizontal averaging (5, 15, 45, and 135 km) to ensure that strongly scattering PSCs (e.g., fully developed STS and ice) are found at the finest possible spatial resolution while also enabling the detection of more tenuous PSCs (e.g., low number density liquid-NAT mixtures) through additional averaging. PSC features found at finer spatial resolution are masked out of the profiles of $\beta'_\perp$
and $R'_{532}$ that undergo additional averaging. Successive averaging minimizes optical aliasing that can result from grouping fine-resolution pixels having vastly different optical properties into a single coarser-resolution average.

Visual comparison of many CALIOP v1 orbital composition images (e.g., Fig. 13 of P11) with corresponding images of $\beta'_\perp$ and $R'_{532}$ indicated that the PSC statistical thresholds used in the v1 algorithm were too conservative, so we have made appropriate adjustments in v2. The thresholds for the background aerosol - assumed to be those data at MERRA-2 temperatures
above 200 K - are now defined as the daily median plus one median absolute deviation of $\beta'_\perp$ and $R'_{532}$. These are computed in overlapping 100 K-thick potential temperature ($\theta$) layers over the range from $\theta$ = 250-750 K. The region of the South Atlantic Anomaly, defined here as a wedge between longitudes 60° W and 45° E, is excluded from the background aerosol threshold calculations due to excessive noise in the CALIOP 532-nm data in this region (Hunt et al., 2009). Then for a candidate CALIOP data point to be identified as a PSC, its value of $\beta'_\perp$ (or $R'_{532}$) must exceed the background aerosol threshold
by at least $u(\beta'_\perp)$ (or $u(R'_{532})$). We also impose a spatial coherence test that requires that more than 11 of the points in a 5-point horizontal by 3-point vertical box centered on the candidate feature exceed the current PSC detection threshold or to

have been identified as a PSC at a previous (finer) averaging scale. This revised approach does a better job overall of capturing PSC clusters identified by the naked eye in CALIOP orbital images while continuing to eliminate false PSC identifications stemming from positive noise spikes in the data. Spot checks of the v2 Antarctic PSC database from early May - when no PSCs observations are expected - indicate that the v2 false positive rate is much less than 0.01%.

Tropopause height information included in the CALIOP Level 1b lidar data product files is based on the MERRA-2 "blended" tropopause altitudes. The MERRA-2 "blended" tropopause is the lower (in altitude) of the temperature-based ("thermal") tropopause and potential vorticity (PV) based ("dynamic") tropopause (Bosilovich et al., 2016; Ott et al., 2016). In practice, the MERRA-2 "blended" tropopause is usually the "dynamic" tropopause in mid- and high-latitudes, but switches to the "thermal" tropopause in the tropics. The tropopause is often difficult to locate in the polar regions, especially during the polar

nights (e.g., Highwood et al., 2000), so any tropopause height determination in these regions should be used with caution. In fact, it is our experience that the transition from upper tropospheric cirrus to PSCs is often ambiguous with no clear separation across the reported tropopause. Consequently, using the MERRA-2 tropopause as a hard boundary to separate tropospheric cirrus from PSCs will certainly lead to misclassifications. Hence, we make no explicit attempt to distinguish tropospheric cloud from stratospheric cloud in the 8.5-30 km altitude range over which we produce the CALIOP PSC cloud mask.

However, we do tag each observation in the database with a feature flag that identifies its altitude location relative to the reported "blended" tropopause as one of three possibilities: (1) below the tropopause, (2) between the tropopause and tropopause + 4 km, or (3) above the tropopause + 4 km. This allows data users to perform some crude separation between tropospheric and stratospheric cloud as desired.

### 3.3 PSC Composition Classification

PSC composition classification is based on comparing CALIOP data with temperature-dependent theoretical optical calculations for non-equilibrium mixtures of liquid (binary $H_2SO_4$-$H_2O$ or STS) droplets and NAT or ice particles. To illustrate the differences between the v2 and v1 algorithms, we show a more extensive set of theoretical results for 50 hPa atmospheric pressure, 10 ppbv $HNO_3$, and 5 ppmv $H_2O$. For these conditions, the NAT equilibrium temperature $T_{NAT} \cong 195.7$ K (Hanson and Mauersberger, 1988); $T_{STS}$, the temperature at which liquid particle volume starts to increase markedly $\cong 192$ K (Carslaw

et al., 1995); and the frost point temperature $T_{ice} \cong 188.5$ K (Murphy and Koop, 2005). The total particle number density ($N_{total}$) is fixed at 10 cm$^{-3}$ (Weigel et al., 2014), partitioned between liquid droplets ($N_{liq}$) and either NAT ($N_{NAT}$) or ice ($N_{ice}$) for the various mixtures being considered. The liquid particle size distribution is assumed to be a single-mode lognormal with geometric standard deviation $\sigma$=1.6, whose mode radius is calculated as a function of $N_{liq}$ and the equilibrium condensed liquid particle VD (Carslaw et al., 1995) for temperatures $T_{ice}$ - 3 K $< T <$ 196 K. We assume that non-equilibrium liquid-NAT

mixtures exist at $T < T_{NAT}$ and that non-equilibrium liquid-ice mixtures exist at $T < T_{ice}$ and that both NAT and ice particle size distributions are single-mode lognormals with $\sigma$=1.38. We consider a range of $N_{NAT}$ ($N_{ice}$) from 0.0001-1.0 cm$^{-3}$ (0.001-10 cm$^{-3}$) and a range of volume-equivalent radii ($r_{NAT}$ and $r_{ice}$) from 0.25-15 μm. However, only those combinations of [$N$, $r$]

with NAT (ice) particle volumes less than or equal to the temperature-dependent equilibrium NAT (ice) volume are physically possible. For the liquid-NAT mixtures, $N_{liq} = N_{total} - N_{NAT}$, and the equilibrium liquid particle VD is reduced to account for the condensed $HNO_3$ and $H_2O$ in the co-existing NAT particles. For the liquid-ice mixtures, $N_{liq} = N_{total} - N_{ice}$, and the equilibrium liquid particle VD is reduced to account for the condensed $H_2O$ in the co-existing ice particles. All particle optical properties were calculated using the database of T-matrix results compiled by Scarchilli et al. (2005) and based on the original work of Mishchenko and Travis (1998), with fixed refractive indices of $1.44 + i0.0$ for binary $H_2SO_4$-$H_2O$ and STS, $1.48 + i0.0$ for NAT, and $1.31 + i0.0$ for ice. We assumed spherical liquid droplets (aspect ratio = 1.0) and assumed both NAT and ice particles to be prolate spheroids with an aspect (diameter-to-length) ratio of 0.9, which Engel et al. (2013) showed to produce best agreement with maximum values observed by CALIOP of particulate depolarization ratio $\delta_{particulate}$:

$$\delta_{particulate} = \beta_{\perp,particulate} / \beta_{\parallel,particulate} = [\beta_\perp - \delta_{mol}\,\beta_{mol}] / [\beta_\parallel - (1 - \delta_{mol})\,\beta_{mol}] \tag{2}$$

where $\delta_{mol}$ is the theoretical molecular depolarization ratio = 0.00366 (see Appendix A).

Figure 3 shows the theoretical results plotted in the coordinate system of $\delta_{particulate}$ vs. inverse scattering ratio ($1/R_{532}$) used in the v1 algorithm. In v1, it was assumed implicitly that attenuation of the CALIOP laser beam due to PSC particles themselves was negligible, i.e. that $R'_{532}$ and $\beta'_\perp$ could be considered equivalent to $R_{532}$ and $\beta_\perp$ for the purpose of composition classification. The individual "streaks" of points in Fig. 3 represent physically possible [$N_{NAT}$, $r_{NAT}$] or [$N_{ice}$, $r_{ice}$] combinations, with temperature decreasing from upper left to lower right along each streak. P09 defined fixed $\delta_{particulate}$ vs. $1/R_{532}$ boundaries separating the composition classes STS, ice (our abbreviated name for liquid-ice mixtures), and Mix1 and Mix2, the latter denoting liquid-NAT mixtures with lower and higher NAT number densities/volumes, respectively. P11 added two additional subclasses: Mix2-enhanced, those liquid-NAT mixtures with optical properties ($2 < R_{532} < 5$ and $\delta_{particulate} > 0.1$) similar to the so-called Type 1a enhanced clouds observed during earlier airborne field missions (e.g., Tsias et al., 1999); and wave ice, PSCs presumably induced by mountain waves. P11 defined wave ice conservatively as those PSCs with $R_{532} > 50$, but noted that the subclass is not all-inclusive, i.e. some additional ice PSCs are likely associated with mountain waves, but do not meet the stringent ($R_{532} > 50$) wave-ice classification criterion. P13 changed the boundary separating STS from liquid-NAT mixtures and ice to a $\beta_\perp$ threshold instead of a fixed value of $\delta_{particulate}$; hence that boundary is shown as a dashed magenta line in Fig. 3. For comparison purposes, Fig. 3 also shows a dashed purple curve representing the lower boundary of the v2 "enhanced NAT mixtures" class, as discussed below.

Two known deficiencies in the v1 composition classification scheme were pointed out in P13. Due to uncertainty in the CALIOP measurements, the optical space boundaries between PSC composition classes are actually "fuzzy" rather than sharp. Thus, toggling between inferred composition classes over small spatial scales may be due to measurement noise rather than a true change in composition. This is especially true in the case of separating liquid-NAT mixtures into the Mix1, Mix2, and Mix2-enhanced categories. P13 also pointed out that the boundary separating ice and liquid-NAT mixtures must be shifted to

larger values of $1/R_{532}$ (smaller values of $R_{532}$) in the event of denitrification and dehydration to avoid ice PSCs being misclassified as liquid-NAT mixtures (also noted by Zhu et al., 2017). To address these deficiencies, we have significantly improved the composition classification scheme in v2. The improvements are discussed below in the context of Fig. 4, where the theoretical optical results are re-plotted in the coordinate system $\beta_\perp$ vs. $R_{532,}$ surrogates for the measured attenuated CALIOP quantities $\beta'_\perp$ and $R'_{532}$ used for PSC detection. As discussed below in Section 3.4, the v2 algorithm also incorporates a retrieval of 532-nm particulate backscatter, $\beta_{particulate}$, through which $\beta'_\perp$ and $R'_{532}$ are later corrected for attenuation due to overlying particulate layers (i.e. the "primes" are removed), allowing for a more robust comparison with the theoretical results. The families of points representing physically possible [$N_{NAT}$, $r_{NAT}$] or [$N_{ice}$, $r_{ice}$] pairs lie at constant $\beta_\perp$ in Fig. 4, with temperature again decreasing from left to right along each family of points. The following points are to be noted in our revised algorithm:

- The former Mix1 and Mix2 classes of liquid-NAT mixtures have been combined into a single class named "NAT mixtures" for brevity. (Note that the line and curve separating Mix1 and Mix2 in Fig. 3 disappears with the v2 definition of NAT mixtures.)

- The former Mix2-enhanced class has been renamed "enhanced NAT mixtures" and it is now defined as the sub-class of NAT mixtures with $R_{532} > 2$ and $\beta_\perp > 2\times10^{-5}$ km$^{-1}$sr$^{-1}$. This conservative boundary was determined empirically by comparing CALIOP Antarctic PSC data to contemporaneous MIPAS observations with and without a belt of NAT clouds formed by heterogeneous nucleation on wave ice PSCs over the Antarctic Peninsula (Höpfner et al., 2006). MIPAS data from 2008 May 27/28/30 (M. Höpfner, Karlsruhe Institute of Technology, private communication) showed no evidence of these NAT clouds, and only about 2% of CALIOP NAT mixture data from those days had $R_{532} > 2$ and $\beta_\perp > 2\times10^{-5}$ km$^{-1}$sr$^{-1}$. In contrast, NAT belt clouds were clearly evident in MIPAS data on 2008 May 29 and 2008 June 01/02, and their locations were matched extremely well by CALIOP NAT mixtures with $R_{532} > 2$ and $\beta_\perp > 2\times10^{-5}$ km$^{-1}$sr$^{-1}$. In theoretical terms, CALIOP enhanced NAT mixture points correspond roughly to those NAT mixtures with $r_{NAT} < 3$ µm and NAT VD > 1.0 µm$^3$cm$^{-3}$, which match the MIPAS NAT detection limits ($r_{NAT} < 3$ µm and NAT VD > 0.3 µm$^3$cm$^{-3}$) reasonably well. Since our criteria defining enhanced NAT mixtures are conservative, the enhanced NAT mixtures sub-class is not all-inclusive, i.e., it does not capture all NAT mixture PSCs heterogeneously nucleated in wave ice PSCs.

- The wave ice class remains the same as in P11, i.e. ice PSCs with $R_{532}>50$. We reemphasize that this definition of wave ice is not all-inclusive, i.e. some additional ice PSCs are likely associated with mountain waves, but do not meet our stringent wave-ice classification criterion.

- The dashed horizontal line labeled $\beta_{\perp,thresh}$ represents qualitatively the CALIOP statistical threshold for detection of PSCs containing non-spherical particles. In practice, this threshold changes with horizontal averaging scale and differs from point to point due to its dependency on $u(\beta_\perp)$. Each data point is assigned a non-spherical particle confidence index $CI_{NS}$ = $[\beta_\perp-u(\beta_\perp)]/u(\beta_\perp)$. Points with $CI_{NS} > 1$ are presumed to be PSCs containing non-spherical particles.

- The dashed magenta vertical line labeled $R_{thresh}$ represents qualitatively the CALIOP statistical threshold for detection of liquid PSCs. In practice, $R_{thresh}$ also changes with horizontal averaging scale and differs from point to point due to its dependency on $u(R_{532})$. Data points classified as STS are those with $CI_{NS} \leq 1$, but with $R_{532} > R_{thresh}$. Each is assigned an STS confidence index $CI_{STS} = [R_{532} - u(R_{532})])/u(R_{532})$; $CI_{STS} > 1$.

- Note that in practice, there is not a distinct separation between histograms of $\beta_{\perp}$ for v2 STS and NAT mixtures. We estimate that 10-15% of data points in either class may fall in the overlap region and thus could be misclassified.

- Points in the grey box at the lower left fall below both CALIOP PSC detection thresholds and are classified as non-features. It should be noted that all measured and derived quantities for non-features are also retained in the v2 data product. A comprehensive discussion of so-called "sub-visible" PSCs can be found in the paper by Lambert et al. (2016), who show that they often can be detected through gas-phase uptake of $HNO_3$ as observed by MLS even though they are not detectable as PSCs by CALIOP.

- The position of the boundary separating NAT mixtures and enhanced NAT mixtures from ice (labeled $R_{NAT|ice}$) now is calculated dynamically according to the total abundances of $HNO_3$ and $H_2O$ vapors. $R_{NAT|ice}$ is based on a parameterization of theoretical calculations of $R_{532}$ for fully developed STS (assumed to be points between $T_{ice}$ and $T_{ice}$-1 K) over a wide range of atmospheric pressures and $HNO_3$ and $H_2O$ mixing ratios. Total $HNO_3$ and $H_2O$ abundances are determined on a daily basis as a function of altitude and DMP equivalent latitude based on nearly coincident "cloud-free" Aura MLS data, where the CALIOP PSC data themselves are used to filter out MLS data affected by uptake in the cloud particles. Then each point with $CI_{NS} > 1$ is assigned a NAT|ice confidence index $CI_{NAT|ice} = (R_{532} - R_{NAT|ice})/u(R_{532})$. For points classified as ice or wave ice, $CI_{NAT|ice} > 0$. For NAT mixtures or enhanced NAT mixtures, $CI_{NAT|ice} < 0$.

- The v2 composition classification extends downward in altitude to the 215 hPa pressure level (~10 km), the lowest reliable level for Aura MLS $HNO_3$ data that is required to define the location of the NAT mixture/ice boundary ($R_{NAT|ice}$) in our classification scheme. All clouds at altitudes below this pressure level are assumed to be ice.

To illustrate that these theoretical calculations resemble actual measurements, composite 2-D histograms of CALIOP PSC data from 10-18 July 2008 (which includes the orbital curtain of Fig. 2) are shown in Figs. 5 and 6 for the v1 and v2 coordinate systems, respectively. In the interest of examining a quasi-homogeneous ensemble, the data have been restricted to latitudes from 65° S to 75° S and $\theta$ from 475 K to 525 K. The most noteworthy feature in Fig. 5 is that the separation between NAT mixtures and ice is not at the fixed value of $1/R_{532} = 0.2$ used in the v1 algorithm, but instead occurs at $1/R_{532} \approx 0.25$-$0.35$. The separation appears to be better captured in Fig. 6, where the average dynamically calculated $R_{NAT|ice} \approx 2.75$ as a result of denitrification and dehydration.

### 3.4 Retrieval of 532-nm Particulate Backscatter

By retrieving the 532-nm particulate backscatter, $\beta_{particulate}$, the observed quantities $\beta'_{\perp}$ and $R'_{532}$ can be corrected for attenuation due to overlying particulate layers (i.e. the "primes" are removed). This allows for a more robust comparison with the

theoretical results and final adjustments in $u(\beta'_\perp)$, $u(R'_{532})$, and the assigned PSC composition class. It also enables the development of approximate relationships (Section 3.5) relating $\beta_{particulate}$ to the bulk particle microphysical quantities SAD and VD. The retrieval procedure we have implemented in v2 follows the general CALIOP particulate extinction retrieval approach outlined by Young and Vaughan (2009). The CALIOP total attenuated 532-nm backscatter profile, with the correction for molecular and ozone attenuation previously applied, can be expressed as follows:

$$\beta'(z) = [\beta_{particulate}(z) + \beta_{mol}(z)]\, \exp[-2\eta(z)\,\tau(0,z)_{particulate}] \tag{3}$$

where $\tau(0,z)_{particulate}$ is the particulate optical depth between the lidar (altitude 0) and altitude z, and $\eta(z)$ is a factor accounting for multiple scattering. By definition,

$$\tau(0,z)_p = \int_0^z \alpha_{particulate}(z)dz' \tag{4}$$

where $\alpha_{particulate}(z)$ is the particulate extinction coefficient.

Making the usual assumption that $\alpha_{particulate} = S_{particulate} \times \beta_{particulate}$, where $S_{particulate}$ is the particulate extinction-to-backscatter (lidar) ratio, leads to an equation of the form $\beta_{particulate}(z) = f\{\beta_{particulate}(z)\}$, which is solved bin by bin in PSCs layers using an unconstrained top-down Newtonian iterative numerical approach as discussed by Young and Vaughan (2009). The multiple scattering factor $\eta(z)$ is calculated as a function of temperature from a spline fit to results from Garnier et al. (2015) for semi-transparent cirrus clouds; for temperatures <190 K (>240 K) $\eta(z)$ is fixed at 0.9 (0.5). Based on ground-based 355-nm lidar measurements in a mountain-wave PSC event at Esrange, Sweden, Reichardt et al. (2004) derived layer-average values of $S_{particulate}$ ranging from 67-82 sr (20-35 sr) for PSC layers with small (large) scattering ratios. These are consistent with 532-nm $S_{particulate}$ values of 50-80 sr derived by Prata et al. (2017) for binary $H_2SO_4$-$H_2O$ aerosols and with values of 14-26 sr derived by Platt et al. (2011) for cold (-80 °C) high-altitude cirrus. For implementation in the v2 algorithm, we derived a parameterization based on the theoretical results described in Section 3.3 for binary $H_2SO_4$-$H_2O$ and STS droplets by which $S_{particulate}$ varies smoothly between the observed bounds:

$$S_{particulate} = 16 + 66/R_{532} - 12/(R_{532})^2 \tag{5}$$

The minimum value of $S_{particulate}$ was set at 16 to ensure that retrievals did not terminate at altitudes above where PSCs were detected in the CALIOP Level 1 attenuated backscatter data.

### 3.5 Estimation of Particle SAD and VD

To estimate SAD and VD using CALIOP data, we followed the methodology applied to stratospheric aerosols by Gobbi (1995), in which functional relationships linking $\beta_{particulate}$, SAD, and VD were determined by averaging the scattering properties of a large set of stratospheric aerosol size distributions. Actual PSC particle size measurements are somewhat limited and have often been obtained under mountain wave conditions (e.g., Deshler et al., 2003; Schreiner et al., 2003; Voigt et al., 2003), and

there is little or no information on the actual shapes of NAT or ice PSC particles. Therefore, we made the simplifying assumption that useful functional relationships could be based on the averaged scattering properties of a range of size distributions for liquid spherical $H_2SO_4$-$H_2O$ and STS particles, the characteristics of which are much better constrained. As described in Section 3.3, the equilibrium VD for $H_2SO_4$-$H_2O$ or STS particles can be calculated as a function of temperature for given atmospheric pressure and $HNO_3$ and $H_2O$ mixing ratios from Carslaw et al (1995). Assuming a single-mode lognormal size distribution with number density $N_{liq}$ and geometric standard deviation $\sigma$, the mode radius and SAD can then be calculated from VD using standard relationships between lognormal moments (e.g., Heintzenberg, 1994). With the particle size distribution fully specified, $\beta_{particulate}$ can be calculated using the database of optical properties for spherical particles compiled by Scarchilli et al. (2005). To explore the sensitivity of the results to size distribution parameters, we performed calculations for other values of $N_{liq}$ (5 and 15 $cm^{-3}$; Wilson et al., 1990; Campbell and Deshler, 2014) and $\sigma$ (1.3 and 1.8) in addition to our standard conditions of $N_{liq}$=10 $cm^{-3}$, $\sigma$ =1.6, 50 hPa atmospheric pressure, 10 ppbv $HNO_3$, and 5 ppmv $H_2O$. The results are shown in Figs. 7 and 8, along with the 3$^{rd}$ order polynomial least-squares fits to the two sets of curves. Note that increases (decreases) in atmospheric pressure, $HNO_3$, or $H_2O$ do not produce different curves, but shift the results for a given curve to the right (left). The RSS uncertainty in liquid particle SAD due to measurement error and lack of knowledge of the size distribution parameters $N_{liq}$ and $\sigma$ is on the order of ±1 (±2.5, ±5) $\mu m^2 cm^{-3}$ for $\beta_{particulate}$ = $10^{-5}$ ($10^{-4}$, $5 \times 10^{-4}$) $km^{-1} sr^{-1}$. The corresponding RSS uncertainty in liquid particle VD is on the order of ±0.05 (±0.15, ±1.0) $\mu m^3 cm^{-3}$ for $\beta_{particulate}$ = $10^{-5}$ ($10^{-4}$, $5 \times 10^{-4}$) $km^{-1} sr^{-1}$.

These liquid particle approximate expressions can be applied to the full suite of CALIOP data, including "sub-visible" PSCs as well as background aerosols. However, there are large uncertainties in the case of NAT mixtures and ice PSCs due to the dearth of information on NAT or ice particle size and shape. Figures 9 and 10 show the complex behavior of SAD and VD versus $\beta_{particulate}$ from the full set of theoretical results for NAT mixtures and ice discussed in Section 3.3 and compare those to the liquid particle approximations shown in Figs. 7 and 8. For a given value of $\beta_{particulate}$, the liquid particle approximation for SAD is an upper limit for the actual SAD in NAT mixtures and a lower limit for the actual SAD in ice PSCs. The level of over/underestimation of SAD may be as much as a factor of 3. For a given value of $\beta_{particulate}$, the liquid particle approximation for VD is a lower limit for the actual VD in ice PSCs and in most NAT mixtures, the exception being those with small $r_{NAT}$ (< ~1.5 μm). The level of underestimation of VD can be as much as a factor of 10 for NAT mixtures and up to a factor of 30 for ice PSCs. To test the validity of our approach, we used bimodal lognormal size distribution fits to in situ optical particle counter measurements within STS, NAT, and ice PSC layers (Deshler et al., 2003) to compute SAD, VD, and $\beta_{particulate}$. These are the blue symbols (labeled according to PSC composition) in Figs. 9 and 10 and show that our estimates of SAD and VD are reasonable.

### 3.6 Illustration of Difference Between v1 and v2 Algorithms

In this section, we illustrate top-level changes in CALIOP PSC data products between the v1 and v2 algorithms. Figures 11(a) and 11(b) present curtains of the retrieved v2 values of the two optical signals used in PSC detection and composition

discrimination, $R_{532}$ and $\beta_\perp$, for the orbit shown in Fig. 2, and Fig. 11(c) is the resultant v2 PSC composition curtain. Spatially-coherent regions of NAT mixtures/enhanced NAT mixtures (yellow/red) and ice (blue) identified along the orbit track correspond directly to regions of enhancements in both $R_{532}$ and $\beta_\perp$, while regions of liquid STS (green) show no enhancements in $\beta_\perp$ since they are spherical droplets and are identified solely through enhancements in $R_{532}$ (i.e. near left edge of orbit curtain).

Also notice the mountain wave ice (dark blue) with its distinctive tilted layer structure over the Antarctic Peninsula (75° S, 300° W). For comparison, the v1 PSC composition curtain is shown in Fig. 11(d). In general, there is much more ice and much less enhanced NAT mixtures in v2 compared with v1, where much of the ice was misclassified as NAT mixtures due to the fixed boundary separating the two composition classes in v1. In the lower stratosphere/upper troposphere, the v2 composition classification is producing more ice than with v1 due to the improved characterization of the $HNO_3$ and $H_2O$ condensables in this region. In addition, v2 substantially fills in holes that are present in the v1 image and reduces the pixel-to-pixel variation in inferred PSC composition. As noted in Section 3.3, all clouds at altitudes below the 215 hPa pressure level are assumed to be ice in v2.

Figure 12 compares v1 and v2 in terms of the total number of CALIOP measurement samples within PSCs (180-m vertical by 5-km horizontal pixels) for the Antarctic in 2009 at altitudes 4 km or more above the tropopause (to avoid contamination from
cirrus), as well as the breakdown of those observations by PSC composition. There are about 19% more total PSC measurements with v2 due to the less conservative PSC detection thresholds. In terms of PSC composition, the STS, NAT mixture, and wave ice fractions are similar in v1 and v2, but there is a significant decrease in v2 relative to v1 in enhanced NAT mixtures (5.8% versus 18.8%) and an accompanying increase in ice PSCs (21.4% versus 10.6%). We estimate that about 75% of the additional v2 ice PSCs come from a reclassification of v1 Mix2-enhanced PSCs, and the remaining 25% from a
reclassification of v1 Mix1 and Mix2 PSCs. Differences between v1 and v2 Antarctic PSC composition for other years and groups of years are comparable to those shown in Fig. 12.

### 3.7 Consistency of v2 PSC Observations with Expected Thermodynamic Regimes

Similar to the approach used by Lambert et al. (2012), P13, and Lambert and Santee (2018), we have examined the consistency of the v2 PSC composition classes with respect to their expected thermodynamic existence regimes through combined analyses
of the CALIOP and Aura MLS data. Here, we extend these previous studies to now include the entire 2006-2017 v2 CALIOP data record. To avoid potentially biased MLS $HNO_3$ measurements where the relatively large geometric field-of-view (FOV) is only partially filled with PSCs (e.g. see P13 and Lambert and Santee, 2018), the analyses are limited to cases where CALIOP PSCs cover at least 75% of the Aura MLS FOV (assumed to be 165 km x 2.16 km) and a single CALIOP composition is dominant. Composite histograms of PSC occurrence vs. $T - T_{ice}$ over the 20-22 km altitude range are shown in Fig. 13(a) and
Fig. 13c) for the Arctic (2006-07 to 2016-17) and Antarctic (2006–2017), respectively. Here, $T$ is the ambient temperature at the CALIOP observation point determined from the MERRA-2 gridded analyses, and $T_{ice}$ is calculated using the Murphy and Koop (2005) relationship with the coincident Aura MLS gas-phase $H_2O$ abundance. Histograms are shown for the CALIOP

STS, NAT mixture (including enhanced NAT mixtures), and ice (including wave ice) composition classes, and each histogram is normalized to a maximum value of 1.0. Figures 13(b) and 13(d) show the same composite histograms transformed to $T$-$T_{eq}$ space, where $T_{eq}$ is defined as $T_{NAT}$, $T_{STS}$, or $T_{ice}$, depending on the CALIOP composition classification, and is calculated using the Hanson and Mauersberger (1988) ($T_{NAT}$), Carslaw et al. (1995) ($T_{STS}$), and Murphy and Koop (2005) ($T_{ice}$) relationships

with the coincident $HNO_3$ and $H_2O$ abundances observed by MLS. $T_{eq}$ for ice PSCs remains the same as in Figs. 13(a) and 13(c) and the ice PSC distributions are identical to those in Figs. 13(a) and 13(c). The STS and NAT mixture histograms are restricted to observations with MLS $HNO_3$ values greater than 1 ppbv to avoid the region where the NAT and STS equilibrium $HNO_3$ uptake curves converge (e.g. see Fig. 3 in P13).

The mode of the ice PSC distribution for both hemispheres is located at a temperature slightly below the frost point with a full-
width-half-maximum of about 1 K. The longer positive tail in the ice PSC distributions may be associated with wave ice events induced by small-scale temperature perturbations that aren't fully resolved in the MERRA-2 temperature fields (e.g., Hoffman et al., 2017a) or NAT mixtures at warmer temperatures misclassified as ice due to measurement noise. STS PSCs in both hemispheres occur over a relative narrow temperature range centered slightly below the STS equilibrium temperature. The relatively narrow widths of the ice and STS histograms with modes near $T_{eq}$ are an indication that these particles are near
equilibrium, as would be expected. The ice and STS histogram mode peaks occurring below $T_{eq}$ are consistent with a small cold bias in the MERRA-2 temperature analyses as noted by Lambert et al. (2012) and Lambert and Santee (2018). The NAT mixture distributions are broader and roughly bimodal with one mode slightly below the NAT equilibrium temperature and a second more populous mode at 3-4 K below NAT equilibrium, which corresponds approximately to the STS equilibrium temperature. As discussed in P13, this bimodality is likely a consequence of different exposure times of air parcels to
temperatures below $T_{NAT}$. The mode near the STS equilibrium temperature represents air parcels with relatively brief exposure to temperatures below $T_{NAT}$. These parcels contain non-equilibrium liquid-NAT mixtures with a detectable enhancement in $\beta_\perp$, but the uptake of $HNO_3$ is dominated by the much more numerous liquid droplets at the lower temperatures. The NAT mixture mode near the NAT equilibrium temperature corresponds to parcels that have been exposed to temperatures below $T_{NAT}$ for extended periods of time, allowing a larger fraction of the gas-phase $HNO_3$ to condense onto the thermodynamically-
favored NAT particles and bringing the mixture closer to NAT equilibrium. These composite histograms, which incorporate over 11 years of CALIOP PSC measurements, demonstrate behavior consistent with theoretical expectations for each composition class, providing confidence that the v2 composition classification scheme is robust.

## 4 PSC Climatologies

Applying the v2 detection and composition classification algorithm to the CALIOP V4.10 Lidar Level 1B data from June 2006
through October 2017, we have created a new PSC reference data record which covers 12 Antarctic PSC seasons (May-October) and 11 Arctic PSC seasons (December-March). It is archived as the CALIPSO Lidar Level 2 Polar Stratospheric Cloud Mask Version 2.0 (v2) data product and publicly available through the NASA Langley Atmospheric Science Data

Center (ASDC) (https://eosweb.larc.nasa.gov/project/calipso/lidar_l2_polar_stratospheric_cloud_table). In this section, we present representative figures drawn from this data record that depict the seasonal and interannual variability of PSC spatial coverage in the Antarctic and Arctic, climatological mean geographic patterns of PSC occurrence, and overall differences between the hemispheres. We also show how Antarctic PSC composition varies climatologically over the season and relate climatological zonal mean cross-sections of PSC occurrence to analogous cross-sections of temperature and the PSC condensables $HNO_3$ and $H_2O$. We reiterate that in general we make no explicit attempt to separate upper tropospheric cirrus from PSCs in the climatologies, but do include the location of the MERRA-2 tropopause in many of the figures as a guide to the reader for the approximate upper extent of cirrus. Only in the PSC spatial volume analyses (Figs. 16 and 23) and composition pie charts (Figs. 12 and 26) do we exclude data within 4 km of the tropopause because inclusion of the omnipresent cirrus would skew the statistics on the temporal evolution of PSC occurrence and relative fraction of ice PSCs.

## 4.1 Antarctic

### 4.1.1 PSC Areal and Spatial Volume Coverage

A depiction of the vortex-wide, seasonal evolution of PSC occurrence is given by a measure of the total areal coverage of PSCs over the polar region as a function of altitude and time. To mitigate the effects of irregular sampling density due to the CALIPSO orbit geometry, the daily total PSC areal coverage is estimated as the sum of the occurrence frequency (number of PSC detections divided by the total number of observations) in ten equal-area latitude bands spanning the 50°-90° S latitude range, multiplied by the area of each band. This estimate implicitly assumes that the CALIOP observations from the approximately 15 orbits per day are representative of the PSC coverage within each latitude band. Note that the highest equal-area latitude band covers 77.8°-90° S, so CALIOP measurements between 77.8°-82° S are assumed to be representative of the entire 77.8°-90° S latitude band. A similar approach has been used to estimate PSC area statistics by P09 for CALIOP observations from 2006-2008 and also by Spang et al. (2018), who found that MIPAS and CALIOP PSC areas from the 2009 Antarctic PSC season were in excellent agreement in spite of the fundamentally different measurement approaches.

The seasonal evolution of PSC areal coverage during each of the 12 seasons in the CALIOP Antarctic data record is shown in Fig. 14. The full altitude range of the PSC data product (8.4 – 30.0 km) is presented with no attempt here to distinguish PSCs from upper tropospheric cirrus clouds that are commonly observed below ~12 km throughout the entire season. Temperatures low enough for PSC existence typically occur inside the stratospheric polar vortex, which in the case of the Antarctic is large, relatively axisymmetric, and generally similar from year-to-year (e.g. Waugh and Randel, 1999). Hence, it is not surprising that the seasonal evolution of PSC coverage in the Antarctic follows a similar pattern from year-to-year, with PSCs first occurring in mid to late May and persisting into October. The total areal extent of PSCs typically peaks in July and August when the vortex is largest and coldest and then diminishes markedly in September and approaches zero in October. PSCs extend in altitude from near the tropopause up to > 25 km, but there is a downward trend in the altitude of maximum areal coverage over time from above 20 km early in the season to near 15 km by September. This corresponds to a downward shift

in the axis of coldest temperatures as the vortex warms at higher altitudes, as was also noted by Poole and Pitts (1994). An interesting feature seen in most years is the apparent merging of the upper tropospheric and lower stratospheric cloud layers in July and August associated with CALIOP observations of deep synoptic-scale clouds extending from the troposphere into the stratosphere to altitudes well above 20 km. These episodic events are likely produced by large-scale adiabatic cooling

along upwardly displaced isentropic surfaces above upper tropospheric anticyclones (e.g. Teitelbaum and Sadourny, 1998; Teitelbaum et al., 2001; Kohma and Sato, 2013). Distinctive tilted cloud layers formed in the cold phases of strong orographic gravity waves (e.g. Cariolle et al., 1989; Höpfner et al., 2006; Orr et al., 2015) are also occasionally observed to extend from the troposphere well into the stratosphere, primarily over the Antarctic Peninsula. Both of these phenomena can be seen in the CALIOP orbit curtain shown in Fig. 2.

Although the general seasonal evolution is similar from year to year, there is a moderate amount of year-to-year variability in PSC coverage during the season that is primarily driven by the dynamical processes that control the size, thermal structure, and stability of the vortex, as well as the strength and frequency of orographic and upper tropospheric forcing events. For instance, 2006 was characterized by an especially large and cold vortex (e.g. WMO, 2007) and showed the largest PSC areas observed by CALIOP to date, while in 2010 and 2012 the vortex was relatively warm with concomitantly much smaller PSC

areas. The climatological mean seasonal evolution of Antarctic PSC areal coverage compiled for the 2006-2017 period is shown in Fig. 15. The climatological daily maximum tropopause height is indicated on Fig. 15 by the dashed white line and provides an approximate upper limit to the extent of cirrus during the season. While it is a reasonable approximation to the seasonal evolution of PSC coverage in any given year, the dynamic variability of the vortex and orographic/upper tropospheric forcing can produce significant deviations from this mean picture. To better quantify the interannual variability in PSC

coverage, we calculated the 12-year mean, standard deviation, and range of daily values of PSC spatial volume (daily area coverage integrated over altitude, e.g. see P09). These PSC spatial volumes are shown in Fig. 16, with maximum and minimum values color-coded according to the year in which they occurred. To avoid contamination from the underlying cirrus, the volume calculations include only those CALIOP observations at altitudes more than 4 km above the reported tropopause. Most of the maximum values in PSC spatial volume are from the very cold 2006 season, and many of the minimum values are from

the warmer 2010/2012 seasons. At the peak of the season in July, the relative standard deviation in PSC spatial volume is about ±25%.

The v2 CALIOP PSC data record can also be exploited to differentiate the seasonal evolution of PSC areal coverage by composition class. Figure 17 shows the 12-year mean relative spatial coverage (composition-specific area normalized by total PSC area) for (a) STS; (b) NAT mixtures, including enhanced NAT mixtures; and (c) ice, including wave ice. To provide

additional perspective, Fig. 17(d) shows the 12-year mean contour plot of $T$-$T_{NAT}$, where again $T$ is the ambient temperature from MERRA-2 gridded analyses and $T_{NAT}$ is calculated using the Hanson and Mauersberger (1988) relationship with cloud-free Aura MLS gas-phase $HNO_3$ and $H_2O$ abundances. To put better focus on PSCs, we limit the lower altitude in Fig. 17 to 12 km, near the climatological maximum tropopause as shown in Fig. 15. The onset of the PSC season in the Antarctic depends

on the details of the evolving Antarctic polar vortex such as its shape, location, and coldness, which vary significantly from year-to-year. Lambert et al. (2016) showed that from 2006-2015, synoptic-scale $HNO_3$ uptake by PSCs was first observed by Aura MLS as early as May 13 and as late as May 28. Furthermore, these initial PSCs are often "sub-visible" and only become detectable by CALIOP some 1-6 days later. Thus we chose to avoid the highly variable onset period in terms of presenting a

representative climatology and restricted our analyses to days and altitudes where PSCs were observed in at least 6 of the 12 Antarctic seasons covered by CALIOP demarcated by the thick black contour line on each of the color panels in Fig. 17. This provides an indication of the climatological temporal and altitude extent of the PSC season. For STS and NAT mixtures (Figs 17a-b), PSC onset in at least six years occurred by approximately 20 May. The onset of ice PSCs (Fig. 17c) is delayed until temperatures drop below the frost point which is typically mid-June. STS (panel a) is the most prevalent composition above

20 km until mid-June and then again at lower altitudes in September and October. The early-season predominance of STS above 20 km corresponds to the region of largest temperature departures below $T_{NAT}$ in panel (d), which is consistent with an enhanced liquid particle growth regime. The predominance of STS late in the season may be an indication that efficient NAT nuclei have been removed through sedimentation of PSC particles during the winter. NAT mixtures (panel b) are by far the dominant composition observed below 20 km in May and early June, comprising >80 % of the total observed PSC area below

17 km, and are also the prevailing composition above 20 km during July through mid-September. The early season maximum of NAT mixtures below 17 km corresponds to a region of temperatures near or just below $T_{NAT}$ where liquid particle growth would not be expected. The onset of ice PSCs (panel c) is delayed 3-4 weeks relative to STS and NAT mixtures, typically occurring around mid-June as temperatures fall below the frost point. The areal extent of ice PSCs is largest in July and August primarily at altitudes below 20 km, but ice is rarely the predominant composition. Note that cirrus contamination is still

apparent in the ice distributions (panel c) above 12 km. The 12-year mean relative PSC composition breakdown shown in Fig. 17 is remarkably similar to the 2006-2008 compilation shown by P09, highlighting the robustness of these results.

### 4.1.2    Zonal Mean and Geographical Distributions of PSC Occurrence

The PSC areal coverage and spatial volume plots capture quite well the seasonal evolution and interannual variability of PSCs from a vortex-wide point of view, but offer no information on the actual geographical patterns of occurrence. To gain this

insight, we now examine monthly zonal mean cross sections and polar maps of PSC occurrence frequency. Latitude/altitude cross sections of monthly zonal mean PSC occurrence frequency compiled from the 12-year CALIOP Antarctic data record are shown in Fig. 18 (top row) for the four primary Antarctic PSC months of June-September. To indicate potential PSC existence regimes, we show corresponding cross sections of zonal mean cloud-free MLS $HNO_3$ (second row) and $H_2O$ (third row), MERRA-2 $T$ (fourth row), and $T$-$T_{NAT}$ (bottom row). For reference, the mean location of the edge of the vortex from

the Aura MLS DMPs and the tropopause altitude from MERRA-2 are indicated on the panels by the black dashed and dotted lines, respectively. In June, PSCs are observed at latitudes poleward of about 65° S from near the tropopause up to about 26 km in altitude, with maximum mean occurrence frequency > 60% near 18 km at the highest latitudes. PSC occurrence peaks during July and August, with the region of highest occurrence frequency expanding in both altitude and latitude in response to

the continued cooling of the polar vortex. There is also a hint of a double peak in occurrence frequency with altitude during these months with the dominant peak near 15 km and a secondary peak above 20 km. PSC occurrence declines significantly in both magnitude and spatial extent in September with only a small region of occurrence frequency > 40% at 14 km near 82° S and overall occurrence restricted to altitudes below 23 km as the vortex warms at higher altitudes. As was observed in the vortex-wide PSC areal coverage plots, there is a systematic shift downward in the altitude of maximum zonal mean PSC occurrence from near 18-20 km in June to below 15 km in September. Upper tropospheric cirrus cloud occurrence frequency is > 10-15% throughout the season at all latitudes.

Although these conventional latitude/altitude zonal means are correct in a statistical sense, the Eulerian view has the disadvantage of possibly averaging together air masses from different, physically distinct regions of the vortex or even from inside and outside of the vortex. Consequently, the latitude/altitude zonal means are difficult to interpret in the context of meteorological and microphysical processes within the vortex that control PSC occurrence. This is especially true when the vortex is elongated and/or not centered over the South Pole. An alternative approach is to average data in the more physically based quasi-Lagrangian coordinate system of equivalent latitude (EqLat) versus potential temperature ($\theta$). This coordinate system roughly captures the motion of air parcel ensembles and is widely used by the stratospheric chemistry and dynamics community in studies of polar processes (e.g., Butchart and Remsberg, 1986; Manney et al., 1999).

Figure 19 shows the EqLat/$\theta$ cross-sectional representations of the 12-year average, monthly zonal mean Antarctic PSC occurrence frequency, cloud-free MLS $HNO_3$ and $H_2O$, MERRA-2 $T$, and $T$-$T_{NAT}$. During most months, the center of the polar vortex is shifted off the pole so that the conventional latitude/altitude cross-sectional monthly means (Fig. 18) blur the sharp gradients in $HNO_3$ and $H_2O$ between the interior and "collar" regions (e.g. Wespes et al., 2009) of the vortex that are much more clearly captured in the EqLat/$\theta$ cross-sections (Fig. 19). Gas-phase $HNO_3$ and $H_2O$ are severely depleted by July in the interior of the vortex at EqLat < −75° and $\theta$ = 400-500 K. Although there is relatively cold air present in this region, the lack of condensables sufficiently lowers the particle thermodynamic existence temperatures (e.g. $T_{NAT}$) to near or below ambient temperatures, limiting PSC existence. Consequently, the highest PSC frequency more typically occurs at equivalent latitudes closer to the vortex edge where there is an optimal combination of sufficient condensables and cold temperatures, which corresponds reasonably well with the minima in the $T$-$T_{NAT}$ distributions (bottom row of Fig. 19). The reason that a double-peak vertical structure in PSC occurrence appears at high latitudes in July and August in the latitude/altitude cross sections (Fig. 18) is much clearer in the EqLat/$\theta$ coordinate system, which show a relative minimum in PSC occurrence at $\theta$ = 450 K (~18 km) corresponding to the layer of depleted condensables.

PSC occurrence is not typically zonally symmetric in either geographic or equivalent latitude coordinate systems, but instead exhibits distinct longitudinal patterns. To illustrate these preferred patterns of PSC occurrence, 12-year average, monthly mean polar maps of Antarctic PSC frequency at $\theta$ = 500 K (~20 km altitude) are shown in Fig. 20. The top row shows the occurrence frequency for all PSCs, while the subsequent rows display the occurrence frequencies of STS, NAT mixtures, and ice,

respectively. Overlaid in the figure are the mean location of the edge of the vortex (black line) and the boundaries of the regions where the mean temperature is below $T_{NAT}$ (solid red line) and below $T_{ice}$ (dashed black line). In general, PSC occurrence is roughly bounded by the region where mean temperature is below $T_{NAT}$ and increases poleward with the highest occurrence frequencies (>60%) generally located within the region of $T<T_{ice}$ at the highest latitudes. The contours of PSC occurrence frequency and cold pool are not symmetric around the pole, but instead pushed slightly off the pole towards the Greenwich Meridian (GM) longitude quadrant. This zonal asymmetry in PSC occurrence is especially pronounced in July-September with the maximum occurrence frequency at 0°-90° W longitude near the base of the Antarctic Peninsula. The enhancement in PSC occurrence at longitudes near the Antarctic Peninsula is due to frequent mountain wave activity in this region (Alexander et al., 2011; Alexander et al., 2013; Hoffman et al., 2017b) and the large-scale upper tropospheric forcing events which are more frequent at these longitudes (Kohma and Sato, 2013).

The mean geographical distributions of STS, NAT mixtures, and ice PSCs at $\theta = 500$ K (Fig. 20, rows 2-4) also exhibit preferred occurrence patterns. STS-only observations are widespread during June at this level, but more limited afterwards with occurrence frequencies generally less than 10% in the interior of the vortex during July-August and less than 5% during September. NAT mixtures, on the other hand, are relatively widespread over much of the vortex at this level, especially during July and August when occurrence frequencies exceed 35%. The ubiquitous NAT mixtures and concomitant limited STS-only observations may be an indication that air parcels well inside the vortex have been exposed to temperatures below $T_{NAT}$ for sufficiently long periods of time to allow the condensed $HNO_3$ to migrate from the STS droplets to the more thermodynamically-favored NAT particles. The ring of increased occurrence of NAT mixtures in July over East Antarctica between 70°-75° S latitude is consistent with the so-called NAT belt that evolves downstream of ice PSCs that frequently occur over the Antarctic Peninsula (e.g. Höpfner et al., 2006). Ice PSC occurrence aligns reasonably well with the region of mean temperatures below $T_{ice}$ that occurs over the interior of the vortex at latitudes generally poleward of 70° S with a distinct maximum in July and August near the base of the Antarctic Peninsula arising from the frequent mountain wave and upper-tropospheric forcing events in this region.

## 4.2 Arctic

### 4.2.1 PSC Areal and Spatial Volume Coverage

The more irregular underlying surface topography in the Northern Hemisphere leads to stronger upward-propagating wave activity than in the Southern Hemisphere, causing a weaker and more distorted Arctic vortex compared to the Antarctic (e.g. Waugh et al., 2017). As a result, the Arctic polar vortex is warmer and exhibits greater temporal variability than its Antarctic counterpart, including sudden stratospheric warmings, which can severely disrupt or even completely break down the vortex in mid-winter (Charlton and Polvani, 2007). Not surprisingly then, Arctic PSC occurrence varies significantly from year to year as is illustrated in Fig. 21, which shows the daily mean PSC areal coverage during each of the 11 Arctic seasons in the CALIOP data record. As in Fig. 14, the full altitude range of the PSC data product (8.4 – 30.0 km) is presented with no attempt

here to distinguish PSCs from upper tropospheric cirrus clouds. The 2010-11 season was marked by persistent periods of PSCs from December-March that set the stage for record ozone depletion over the Arctic (Manney et al., 2011b). During the 2015-16 season, CALIOP observed the largest areal coverage of PSCs over the Arctic to date, including areas of synoptic ice PSCs, which have only been observed by CALIOP in the Arctic in only one other season (2009-10). In contrast to these remarkable Arctic PSC seasons, the warm 2014-15 winter was almost devoid of PSCs. Also note the merging of the upper tropospheric and lower stratospheric cloud layers during some winters (e.g. 2015-16). As in the Antarctic, this is indicative of upper tropospheric forcing events in the Arctic (Fromm et al., 2003; Achtert et al., 2012) that produce deep synoptic-scale cloud layers extending from the troposphere into the stratosphere.

For comparison with the Antarctic multi-year mean (Fig. 15), the mean seasonal evolution of Arctic PSC areal coverage compiled for the 2006-2017 period is shown in Fig. 22 with the climatological maximum daily tropopause height indicated by the dashed white line. Clearly the multi-year Arctic mean is unlike any year in the CALIOP record and, hence, would not be very meaningful in itself as guidance for representing Arctic PSCs in a model. The large year-to-year variability in Arctic PSC coverage is further quantified in Fig. 23, which depicts the time series of 11-year mean daily PSC spatial volumes over the Arctic along with the standard deviations, maxima, and minima. All the maxima in January correspond to the anomalous 2015-16 season while the majority of the maxima in February and March correspond to the 2010-11 season. The year-to-year variability in the PSC spatial volume in the Arctic is much larger than in the Antarctic, with the relative standard deviations exceeding 100% for most days. In terms of the climatology of Arctic PSC composition, we feel that it meaningful to show only the composite season-long vertical profile of relative spatial coverage (composition-specific area normalized by total PSC area) in Fig. 24. STS and NAT mixtures are the major Arctic PSC compositions as expected, with STS (NAT mixtures) predominant above (below) 24 km. Note that upper tropospheric cirrus produces the ice maximum near 12 km.

### 4.2.2 Geographical Distributions of PSC Occurrence

In spite of the high interannual variability in PSC areal coverage, the geographical pattern of PSC occurrence in the Arctic is quite regular from year to year, with PSCs primarily confined to longitudes from about 60° W to 120° E as illustrated in the 11-year average, monthly mean Arctic PSC occurrence frequency maps for December and January shown in Figure 25. This region corresponds to the climatologically favored location of the Arctic vortex in recent decades (e.g. Zhang et al., 2016) which has been influenced by enhanced zonal wavenumber 1 activity, pushing the vortex off the North Pole towards Eurasia.

### 4.3 Differences between Antarctic and Arctic

As discussed in Sections 4.1 and 4.2, the Antarctic polar vortex is a much more conducive environment for PSC existence than its Arctic counterpart. The Antarctic PSC season is longer and more regular with PSCs present every year from mid-May to early October, while in the Arctic, PSC occurrence is possible from December-March but not guaranteed in any of these months. The contrast between CALIOP PSC observations in the two hemispheres can be seen in Figure 26, which shows the

total number of measurement samples within PSCs over the entire 2006-2017 data record (12 Antarctic seasons and 11 Arctic seasons), as well as the average, minimum, and maximum percentage of measurements by composition class. On average, about 14 times more PSCs were sampled during a season in the Antarctic than in the Arctic. The largest differences in composition are in ice, which comprised nearly 25% of Antarctic PSCs compared to less than 5% in the Arctic (a result of the

much colder Southern vortex) and in NAT mixtures, which comprised nearly 60% of Arctic PSCs, but only about 40% of Antarctic PSCs. The percentages of STS, enhanced NAT mixtures, and wave ice are not vastly different between the two hemispheres.

## 5 Particle Surface Area Density and Volume Density

As described in Section 3.5, estimates of the bulk particle microphysical quantities SAD and VD are included in the new
CALIOP v2 PSC data record. The estimates assume liquid particles (binary $H_2SO_4$-$H_2O$ or STS) only and thus have large uncertainties when NAT mixtures or ice are present. Our estimated SAD is likely an upper (lower) limit for the actual SAD in NAT mixture (ice) PSCs, while our estimated VD is likely a lower limit for the actual VD in ice PSCs and in most NAT mixtures. Nonetheless, they represent the first long-term, vortex-wide observational-based record of SAD and VD and can be used to compare CALIOP stratospheric data with in situ particle measurements and to test parameterizations of the chemical
and radiative effects of particles in current and future theoretical models. Since the SAD and VD estimates cover the full range of CALIOP data, including "sub-visible" PSCs as well as background aerosols, they may prove especially valuable in studies of the role of PSCs relative to that of cold background aerosols in early-season chlorine activation (e.g., Wegner et al., 2016; Drdla and Müller, 2012).

The climatological, 12-year mean depiction of the temporal evolution of vortex-averaged SAD over the Antarctic is shown in
the top panel in Figure 27. The vortex-mean SAD begins to rise in mid-May, which may be an indication of binary aerosol deliquescence as the vortex cools and/or the initial onset of PSCs. SAD increases more significantly in June as PSCs become widespread below 25 km. The maximum SAD and its quasi-periodic nature are associated with ice PSCs that are most prevalent in July and August below 20 km. Twelve-year average, monthly mean polar maps of SAD at 18 km altitude are shown in the bottom row of Figure 27. Since ice PSCs produce the largest enhancements in SAD, the mean geographical
distribution of SAD closely mirrors the highly zonally asymmetric pattern of ice PSC occurrence, with largest values in the 90° W to 0° longitude sector where ice PSC occurrence is most prevalent, especially in July-September. The spatial and temporal patterns in estimated VD (not shown) are very similar to those in estimated SAD, as expected.

## 6 Comparison with Other PSC Data Sets

In this section we discuss comparisons of CALIOP v2 PSC data with MIPAS PSC observations over the period 2002-2012
and with contemporaneous and historical ground-based lidar PSC observations from McMurdo Station, Antarctica (77.85° S,

166.67° E) and Ny-Ålesund, Spitsbergen (79° N, 12° E). Also, to investigate the possibility of longer-term trends, we compare CALIOP PSC data from 2006-2017 with the SAM II (Stratospheric Aerosol Measurement II) solar occultation PSC record from the period 1978-1989 (Poole and Pitts, 1994; Fromm et al., 2003).

PSCs are detected and classified in MIPAS data based on differences in IR limb emission spectral measurements from different
atmospheric window regions (Spang et al., 2005; 2016). Since this approach is completely different from that of CALIOP, comparisons with MIPAS PSC observations provide an independent test of the validity of the CALIOP PSC results. The first such assessment was presented by Höpfner et al. (2009), who showed a high degree of correlation between MIPAS NAT and CALIOP v1 NAT mixtures for the 2006–2007 Antarctic and 2006/07–2007/08 Arctic winters. Spang et al. (2018; hereafter S18) recently published a climatology of PSC occurrence and composition classification based on MIPAS data from 2002-
2012. S18 compared MIPAS and CALIOP v2 observations of daily, altitude-resolved PSC areal coverage for the 2009 Antarctic season and found excellent correspondence in the overall spatial and temporal evolution as well as for different PSC composition classes. The 10-year mean MIPAS daily PSC areal coverage shown by S18 is very similar to the 11-year mean CALIOP v2 areal coverage (Fig. 15). Additionally, S18 showed the predominant PSC composition to be STS in May and early June and NAT over most altitudes from early July through the end of the season, which is consistent with the CALIOP
v2 relative composition areal coverages shown in Fig. 17. In another recent paper, Höpfner et al. (2018) compared retrieved/estimated vertical profiles of PSC particle VD from coincident (within 200 km distance and 2 h time) MIPAS and CALIOP measurements during the 2009 Antarctic winter. For STS PSCs, the comparisons showed very good agreement between the instruments in terms of the vertical profile shape as well as the absolute values of particle VD.

To facilitate comparisons with the ground-based lidar data, we calculated the frequency of CALIOP v2 PSC observations
within ±1.5° latitude and ±15° longitude of McMurdo and Ny-Ålesund, respectively. Figure 28 shows CALIOP PSC occurrence frequencies near McMurdo for June-September, 2006-2010 compiled in 15-d × 1.5-km bins, which is similar in nature to the ground-based McMurdo 2D PSC occurrence frequency histogram for the same time period given in Fig. 3 in Di Liberto et al. (2014). Figure 29 extends this to show CALIOP PSC occurrence frequencies near McMurdo for May-September, 2006-2017 compiled in 15-d × 2-km bins. This figure is quite similar to the 2D histogram of ground-based lidar PSC
occurrence frequencies at McMurdo for the earlier period 1995-2001 shown in Fig. 3b in Massoli et al. (2006). Finally, Figure 30 shows CALIOP PSC occurrence frequencies near Ny-Ålesund for December-February, 2006-2017 compiled in 7-d × 2-km bins. This figure is qualitatively similar to the 2D histogram of Ny-Ålesund ground-based lidar PSC occurrence frequencies for five earlier Arctic seasons from 1995/96 and 2002/03 shown in Fig. 3a in Massoli et al. (2006). It is not surprising that the absolute sighting frequencies at Ny-Ålesund by CALIOP and the ground-based lidar do not agree well given the different time
periods and the high degree of interannual variability in PSC occurrence in the Arctic (as illustrated earlier in Fig. 23).

The first spaceborne sightings of PSCs were by SAM II (McCormick et al., 1982), which was a single-channel (1-µm) sun photometer (McCormick et al., 1979) that operated on the Nimbus 7 satellite from October 1978 – December 1993 (orbit

degradation led to significant data gaps after 1989). SAM II measured the solar radiance in a small (0.01°) field of view along a tangential ray path through the Earth's atmosphere during each sunrise and sunset encountered by the satellite. The radiance data were reduced to give transmittance profiles, which were then inverted by the methods of Chu and McCormick (1979) to produce 1-km vertical resolution profiles of particulate extinction ($\alpha_{particulate}$) at the 1-µm wavelength. Due to the orbital

characteristics of Nimbus 7, all sunrise events occurred in the Southern Hemisphere at latitudes from 64° S to 81° S, and all sunset events occurred in the Northern Hemisphere between 65° to 84° N. The measurement locations progressed slowly in latitude (1°-2° per week) from one extreme to the other over a period of three months with the minimum and maximum latitudes measured at the solstices and equinoxes, respectively. There were approximately 14 measurements in each hemisphere per day, each separated by about 26° longitude. The SAM II extinction values represent an average over a measurement volume

near the tangent point of the optical path, which is approximately 230 km long × 1-km thick. Because of this long horizontal path through the atmosphere, SAM II was very sensitive to the presence of thin cloud that would be transparent to many nadir-viewing instruments. While the inversion treats the atmosphere as a series of concentric shells where the particulate matter in each shell is assumed to be homogeneously distributed throughout the shell, in reality clouds can occur at any point along the ray path at an altitude equal to or higher than the tangent point. With a 12-bit digitizer (Chu and McCormick, 1979), SAM II

could not measure through clouds with optical depths greater than about 6 (maximum $\alpha_{particulate} \cong 0.02$ km$^{-1}$).

The first step in comparing CALIOP with SAM II is to produce a subset of CALIOP data (which we shall call CALIOP-SO) matched to the nominal SAM II solar occultation sampling geometry. For each day from May-November in the Antarctic and from November-March in the Arctic, we determined the longitudes at which the CALIPSO orbits crossed the nominal SAM II measurement latitude(s). We then defined a 230 km along track × 1 km vertical subset of CALIOP data centered at the SAM

II measurement latitude to represent the large occultation sampling volume. If CALIOP detected PSCs in at least five measurement pixels (5 km horizontal × 180-m vertical) within this volume, then the CALIOP-SO measurement was counted as a PSC. The mean CALIOP 532-nm particulate backscatter ($\beta_{particulate}$) in the occultation sampling volume was then calculated. For a given day, we produced 14-15 simulated CALIOP-SO profiles at 1-km vertical resolution from 14 – 30 km in altitude. We repeated this for each day in the CALIOP PSC data record, producing a CALIOP-SO database covering the

2006-2017 period.

For a quantitative comparison of the two data sets, we first converted the CALIOP-SO mean 532-nm $\beta_{particulate}$ values to 1-µm $\alpha_{particulate}$ values using the relationship from Gobbi et al. (1995). We then multiplied the archived SAM II 1-µm $\alpha_{particulate}$ values by a factor of 1.3 based on the assessment of Thomason et al. (2018) who noted that the SAM II data may be biased low by as much as 30%. Note that such a bias would have little effect on published SAM II PSC statistics that were based on relative

increases in $\alpha_{particulate}$. We then produced PDFs as a function of altitude of calculated 1-µm $\alpha_{particulate}$ for CALIOP-SO Antarctic PSCs for 2006-2017, and of adjusted (×1.3) SAM II 1-µm $\alpha_{particulate}$ for Antarctic PSCs detected during the years 1978-1989. We restricted our analysis of the optical signals to the Antarctic because of the large degree of similarity in PSCs there from

year to year, and omitted the years 1983-1986 from the SAM II composite to avoid the possible masking influence of the 1982 El Chichón eruption. From the PDFs, we determined the season-long minimum (1$^{st}$ percentile) detectable values of 1-μm $\alpha_{particulate}$ for SAM II (×1.3, solid curve) and CALIOP-SO (dotted curve) PSCs, which are plotted as a function of altitude in Fig. 31. These show a clear difference in sensitivity between the instruments, with SAM II able to detect more tenuous PSCs.

To put both datasets on equal footing for comparing PSC occurrence frequency, we then reprocessed the SAM II data for PSC detections, excluding all data points with adjusted 1-μm $\alpha_{particulate}$ that fell below the CALIOP-SO minimum detection threshold. For the CALIOP-SO profiles, we excluded data that would have been beyond the optical depth limit of SAM II, i.e. points with 1-μm $\alpha_{particulate} > 0.02$ km$^{-1}$ as well as all points at lower altitudes in those profiles. Figure 32 shows the time series of multi-year mean Antarctic PSC column occurrence frequency for CALIOP-SO (a) and SAM II (b), along with standard

errors in the means. Note that the nominal SAM II solar occultation sampling latitude tracks near the terminator, and after September there are no night-time CALIOP measurements at the SAM II sampling latitude. Overall, the magnitude and variability of the CALIOP and SAM II Antarctic PSC column occurrence frequencies are similar, suggesting there have not been any significant changes in since the SAM II era. CALIOP-SO and SAM II column PSC occurrence frequencies for the Arctic are shown in Fig. 33. The two records are similar for February, but the CALIOP-SO occurrence frequencies are

significantly higher than SAM II for December and February. This is likely a consequence of the high degree of interannual variability in Arctic PSCs rather than an indicator of a long-term trend.

**7 Summary and Discussion**

Measurements from CALIOP on the CALIPSO satellite have greatly expanded the PSC observational data record with now over 11 years of observations to date. The spaceborne lidar profiles the polar stratosphere with unprecedented spatial

(5-km horizontal × 180-m vertical) and temporal (~15 orbits/day) resolution and its dual-polarization capability allows classification of PSCs according to composition. A new v2 CALIOP PSC algorithm has been developed that corrects a number of known deficiencies in previous versions, leading to significantly improved PSC composition data products. Major v2 enhancements include dynamic adjustment of composition boundaries to account for effects of denitrification and dehydration, direct use of measurement uncertainties, addition of composition confidence indices, and retrieval of particulate backscatter,

which enables simplified estimates of particulate SAD and VD. Top-level comparisons between v1 and v2 data products indicate that the improved discrimination between ice and NAT mixtures leads to roughly twice as much ice identified in v2 relative to v1, coming primarily at the expense of enhanced NAT mixtures. Composite multi-season histograms of v2 PSC observations in each composition class versus $T$-$T_{eq}$ were shown to conform to their expected existence regimes, with narrow distributions near $T_{eq}$ for STS and ice, which are thought to be near thermodynamic equilibrium, and a broader bimodal

distribution of NAT mixtures due to the frequent non-equilibrium growth of NAT particles. These results are consistent with findings of P13 for the 2006-2009 period, underscoring the robustness of the v2 composition discrimination approach.

Utilizing the v2 algorithm, we have produced a state-of-the-art CALIOP PSC reference data record that spans the June 2006-October 2017 time period with PSC information compiled along each of the ~15 CALIPSO orbits per day. Nearly coincident Aura MLS measurements of $HNO_3$ and $H_2O$, the primary PSC condensables, along with vortex information from the Aura MLS DMPs have been mapped to the CALIOP PSC along-orbit grid and included in the PSC data products to facilitate their

use in the analyses. In combination, this data record represents the most comprehensive, high resolution PSC database in existence and establishes the foundation for the compilation of a robust climatology of PSC occurrence and particle characteristics. The CALIPSO Lidar Level 2 Polar Stratospheric Cloud Mask Version 2.0 (v2) data product is archived at the NASA Langley Science Data Center and available publically (https://eosweb.larc.nasa.gov/project/calipso/lidar_l2_polar_stratospheric_cloud_table).

From the 11+ year CALIOP PSC reference data record, we have compiled a comprehensive climatology of PSC occurrence and composition for both the Antarctic and Arctic. The seasonal evolution of Antarctic PSC areal coverage corresponds closely to the evolution of the stratospheric polar vortex which is generally similar from year to year in the Antarctic and hence is captured reasonably well by the multi-season mean depiction. However, year-to-year variability in vortex shape, size and thermal structure leads to moderate variability in PSC coverage, with about 25% relative standard deviation in PSC spatial

volume at the peak of the season in July and August. The relative breakdown of areal coverage by composition shows that STS is the predominant particle composition early in the season above 20 km where temperatures are optimal for liquid particle growth and again late in the season when efficient NAT nuclei may have been depleted. NAT mixtures are predominant in the slightly warmer ($T \cong T_{NAT}$) environment below 16 km in late May and June, and also above 17 km from July through mid-September when air parcels have long exposures to $T < T_{NAT}$, especially in the interior of the vortex, leading to the

thermodynamically-favored NAT at the expense of STS. Monthly zonal mean cross sections show the multi-season average patterns of PSC occurrence in geographic latitude/altitude and also equivalent latitude/potential temperature coordinates. The vortex-centered EqLat/$\theta$ coordinates better capture processes controlling PSC existence such as gradients in condensable abundances that are more closely aligned with the structure of the vortex. PSC occurrence is limited deep within the interior of the vortex at high equivalent latitudes due to severe denitrification and dehydration. The maximum in PSC occurrence

frequency is typically at EqLats between 65°-75° S, closer to the collar region of higher $HNO_3$ near the edge of the vortex.

Geographical patterns of Antarctic PSC occurrence were investigated through examination of polar (latitude-longitude) maps of multi-season, monthly mean PSC occurrence on constant potential temperature surfaces. Overall, there is a maximum in Antarctic PSC occurrence between 90° W and 0° longitude, consistent with the preferential region for forcing by mountain waves and upper-tropospheric anticyclones. CALIOP observations of deep cloud systems that extend from the troposphere

well into the stratosphere up to 20-25 km are indicative of the important role of large-scale upper tropospheric forcing in PSC formation. The particle characteristics within these deep cloud systems, particularly in the transition region near the tropopause are not well understood and warrant further investigation.

Specific compositions also exhibit preferred geographical patterns of occurrence. STS occurrence is typically limited in the interior of the vortex, while NAT mixtures are abundant throughout the vortex. The ubiquitous NAT mixtures and concomitant absence of STS-only observations is likely an indication that air parcels well inside the vortex have been exposed to temperatures below $T_{NAT}$ for sufficiently long periods of time to allow the condensed $HNO_3$ to migrate from STS droplets to the more thermodynamically-favored NAT particles. A NAT mixture belt is also seen in the multi-year means over East Antarctica, consistent with MIPAS observations (Höpfner et al., 2006). The mean pattern of ice PSC occurrence is dominated by mountain wave forcing, with a maximum in the 90° W to 0° longitude quadrant near the Antarctic Peninsula.

In contrast to the Antarctic, Arctic PSC occurrence is highly variable from year-to-year due to the more disturbed Arctic vortex that is prone to sudden stratospheric warmings. As such, the evolution of an Arctic PSC season doesn't follow a climatological mean pattern and instead each PSC season is distinctly different. For instance, PSC areas during the 2010-11 and 2015-16 Arctic seasons were the highest observed during the CALIOP lifetime to date, while the 2014-15 season was almost devoid of PSCs. As a result, the relative standard deviation in Arctic PSC spatial volume is greater than 100% throughout most of the season. In spite of the high variability in Arctic PSC occurrence, when PSCs occur they are typically found between 60° W and 90° E longitude, consistent with the preferential location of the Arctic vortex during the last decade. The larger, colder, and more stable Antarctic vortex is much more conducive for PSC formation than the Arctic vortex, leading to about a factor of 14 more PSC observations on average in the Antarctic than Arctic during the CALIOP era. The most compelling difference between the hemispheres in observed composition is in ice, which comprises 24% of PSC observations in the Antarctic on average, but only 4% in the Arctic due to the inherently warmer conditions.

Estimates of the bulk particle microphysical quantities SAD and VD are included in the new CALIOP v2 PSC data record. The estimates assume liquid particles (binary $H_2SO_4$-$H_2O$ or STS) only and thus have large uncertainties when NAT mixtures or ice are present. Our estimated SAD is likely an upper (lower) limit for the actual SAD in NAT mixture (ice) PSCs, while our estimated VD is likely a lower limit for the actual VD in ice PSCs and in most NAT mixtures. Nonetheless, they represent the first long-term, vortex-wide observational-based record of SAD and VD and can be used to compare CALIOP stratospheric data with in situ particle measurements and to test parameterizations of the chemical and radiative effects of particles in current and future theoretical models. A climatology of the seasonal evolution of vortex-averaged SAD was presented, showing an initial increase in May associated with particle growth as the vortex cools, possibly from deliquescence of binary aerosol, and then a more substantial increase as PSCs become widespread in June. Maximum SAD occurs in July and August below 20 km when ice PSCs are most prevalent. Multi-season average, monthly mean polar maps of SAD exhibit a zonally asymmetric pattern that mimics ice PSC occurrence, with maxima occurring near the mountainous Antarctic Peninsula where orography leads to enhanced ice cloud formation.

Comparisons of CALIOP v2 and MIPAS data showed excellent agreement in the overall spatial and temporal evolution of Antarctic PSCs as well as that for different PSC composition classes. CALIOP v2 PSC occurrence frequency patterns in the

vicinity of ground-based lidars at McMurdo Station, Antarctica, and Ny-Ålesund, Spitsbergen, are similar in nature to the climatological patterns derived from the ground-based measurements. Finally, to investigate potential longer term trends in PSC occurrence, appropriately subsampled and averaged CALIOP v2 PSC observations from 2006-2017 were compared with PSC data from the 1979-1989 period collected by the spaceborne solar occultation instrument SAM II (Stratospheric Aerosol

Measurement II). The two instruments showed similar magnitude and variability in Antarctic PSC column occurrence frequency, suggesting that there has been no long-term trend. For the Arctic, the two instruments showed similar results for February, but CALIOP column occurrence frequencies were substantially higher than SAM II for December and January. This finding is likely a reflection of the high degree of interannual variability in Arctic PSCs rather than an indicator of a long-term trend.

**8. Data availability**

CALIPSO/CALIOP L1B: Winker, D. (2016), CALIPSO LID L1 Standard HDF File - Version 4.10, NASA Langley Research Center Atmospheric Science Data Center DAAC, Last access December 2017, https://doi.org/10.5067/caliop/calipso/lid_l1-standard-v4-10.

CALIPSO/CALIOP L2 PSC Mask: CALIPSO Science Team (2015), CALIPSO/CALIOP Level 2, Polar Stratospheric Cloud Data, version 1.00, Hampton, VA, USA: NASA Atmospheric Science Data Center (ASDC), Last access October 2017, https://doi.org/10.5067/CALIOP/CALIPSO/CAL_LID_L2_PSCMask-Prov-V1-00_L2-001.00.

Aura MLS $HNO_3$ data: EOS MLS Science Team (2017), MLS/Aura Near-Real-Time L2 Nitric Acid (HNO3) Mixing Ratio

V004, Greenbelt, MD, USA, Goddard Earth Sciences Data and Information Services Center (GES DISC), Last access October 2017, https://disc.gsfc.nasa.gov/datacollection/ML2HNO3_NRT_004.html.

Aura MLS $H_2O$ data: EOS MLS Science Team (2017);, MLS/Aura Near-Real-Time L2 Water Vapor (H2O) Mixing Ratio V004, Greenbelt, MD, USA, Goddard Earth Sciences Data and Information Services Center (GES DISC), Last access October

2017, https://disc.gsfc.nasa.gov/datacollection/ML2H2O_NRT_004.html.

Aura MLS Derived Meteorological Products: Manney et al. (2007); Manney et al. (2011a), Last access December 2017 at https://mls.jpl.nasa.gov/dmp/.

SAM II Aerosol Extinction data: SAM II Science Team (1999), SAM II Level 2 Data, Hampton, VA, USA: NASA Atmospheric Science Data Center (ASDC), Last access October 2017 at doi: 10.5067/NIMBUS7/SAMII/SOLAR_ASCII_L2-AV

**Appendix A**

**A.1 Crosstalk Correction**

The CALIOP backscatter signal is separated into parallel (∥) and perpendicular (⊥) components by a polarization beam splitter in the receiver subsystem (Hunt et al., 2009). With an ideal beam splitter, the measured molecular depolarization ratio ($\delta_{mol,meas}$) would equal the theoretical value of 0.00366 at the ~40-pm bandwidth of the etalon in the CALIOP receiver (Cairo et al. 1999; Hostetler et al., 2006). The difference between the measured and theoretical molecular depolarization ratios indicates the level of crosstalk (*CT*) between the two polarization channels. We assume for simplicity that a fraction *CT* of the received parallel signal is reflected into the perpendicular channel and that the remainder (1-*CT*) of the received parallel signal is transmitted into the parallel detector. With this assumption and some algebraic manipulation, it can be shown that

$$CT = (\delta_{mol,meas} - 0.00366) / (1 + \delta_{mol,meas}) \tag{A.1}$$

The crosstalk-corrected attenuated backscatter signals can then be derived from the measured signals as follows:

$$\beta'_{\parallel} = \beta'_{\parallel,meas} / (1\text{-}CT) \tag{A.2}$$

$$\beta'_{\perp} = \beta'_{\perp,meas} - \beta'_{\parallel}(CT) \tag{A.3}$$

Figure A1 shows a time series of *CT* calculated from daily values of $\delta_{mol,meas}$ over the course of the CALIPSO mission, as well as the PSC season averages used for simplicity in our algorithm. The abrupt jumps in *CT* are all associated with events in which the etalon temperature was changed, suggesting that they are real changes due to hysteresis associated with temperature cycling of the etalon and its mount. *CT* has been less than 0.5% over the entire mission except for the 2008 Antarctic and 2008-09 Arctic winters, when it was 0.6% – 0.65%. $\delta_{mol}$ has not been measured regularly since March 2015, so a constant value of *CT* has been assumed after that point.

**A.2 Random Measurement Uncertainties**

Random uncertainties in $\beta'_{\parallel}$ [$u(\beta'_{\parallel})$] and $\beta'_{\perp}$ [$u(\beta'_{\perp})$] due to shot noise are computed using the noise scale factor (NSF) approach introduced by Liu et al. (2006) and described in detail for the CALIOP system by Hostetler et al. (2006). The uncertainties are scaled by the inverse square root of the product of: the number of 15-m vertical bins being averaged, which is 12 in the case of our fixed 180-m vertical resolution, and the number of 1/3-km horizontal resolution laser shots being averaged, which ranges from 15 to 405 in our successive horizontal averaging scheme. Relative random uncertainty in attenuated scattering ratio $R'_{532}$ [$u(R'_{532})/R'_{532}$] is calculated as the square root of the sum of squares of the relative random uncertainties in $\beta'_{\parallel}$ [$u(\beta'_{\parallel})/\beta'_{\parallel}$] and $\beta'_{\perp}$ [$u(\beta'_{\perp})/\beta'_{\perp}$] plus an assumed 3% relative uncertainty in $\beta_{mol}$ (Hostetler et al., 2006). The basic random uncertainties are propagated through the calculation of other optical quantities to estimate their uncertainties as well.

**Competing Interests**

The authors declare that they have no conflict of interest.

**Acknowledgements**

The authors would like to thank David Considine, Program Scientist for the CALIPSO/CloudSat Missions for continued
support of this research.  The authors also acknowledge the Stratosphere-troposphere Processes And their Role in Climate
(SPARC) project and the International Space Science Institute (ISSI) for its support of the SPARC Polar Stratospheric Cloud
Initiative (PSCi).  Support for L. Poole is provided under NASA contract NNL16AA05C.  MCP and LRP would like to pay
special tribute to our late colleague William H. (Bill) Hunt, a senior lidar engineer who made many significant contributions
to the success of atmospheric lidar programs at NASA Langley over his 40-year career.  A testament to his thoroughness and
dedication is the fact that CALIOP has exceeded its expected lifetime many times over.

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

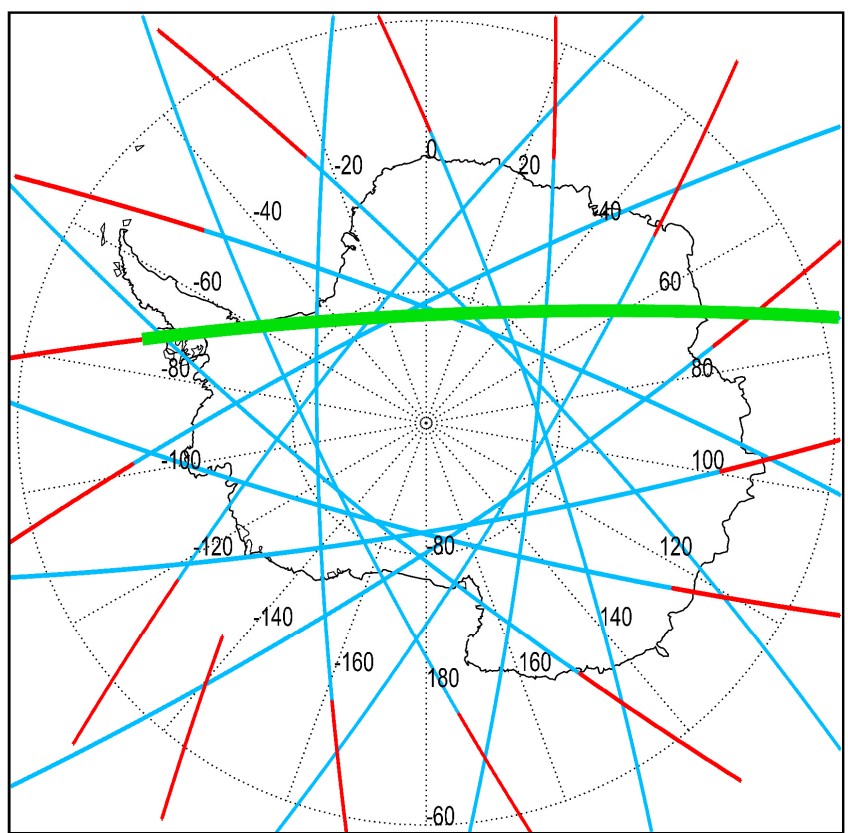

**Figure 1: CALIPSO orbital coverage over the polar region of the Southern Hemisphere on 17 July 2008. Blue (red) lines depict nighttime (daytime) orbit segments. The CALIOP curtain along the orbit highlighted in green is shown in Fig. 2.**

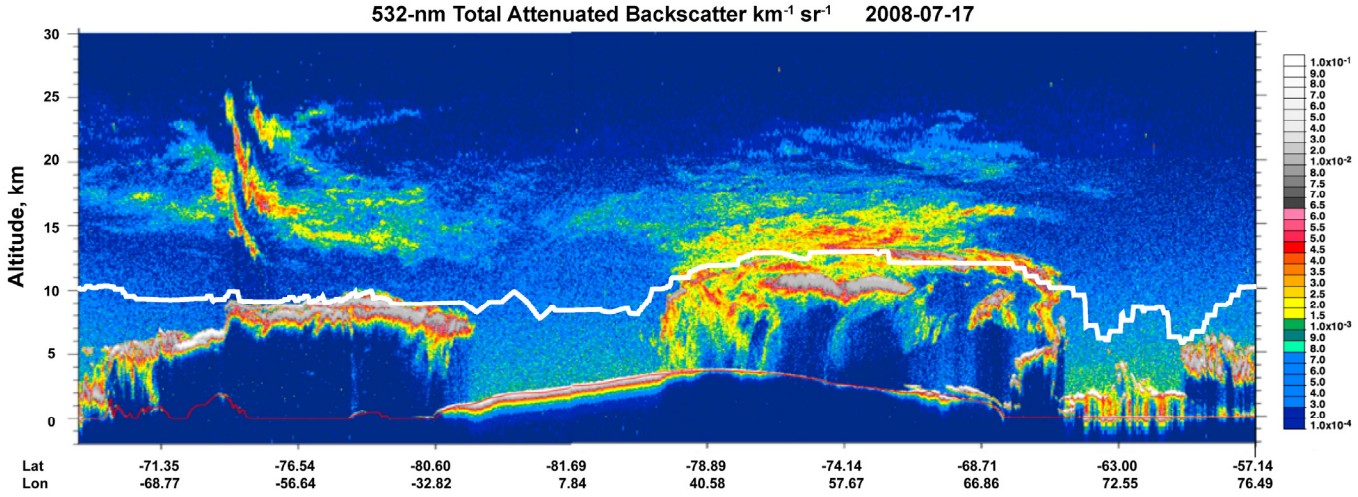

**Figure 2: Orbital curtain of CALIOP 532-nm total attenuated backscatter coefficient (km⁻¹sr⁻¹) along the single orbit highlighted in green in Fig. 1.  The MERRA-2 tropopause height is indicated by the solid white line.**

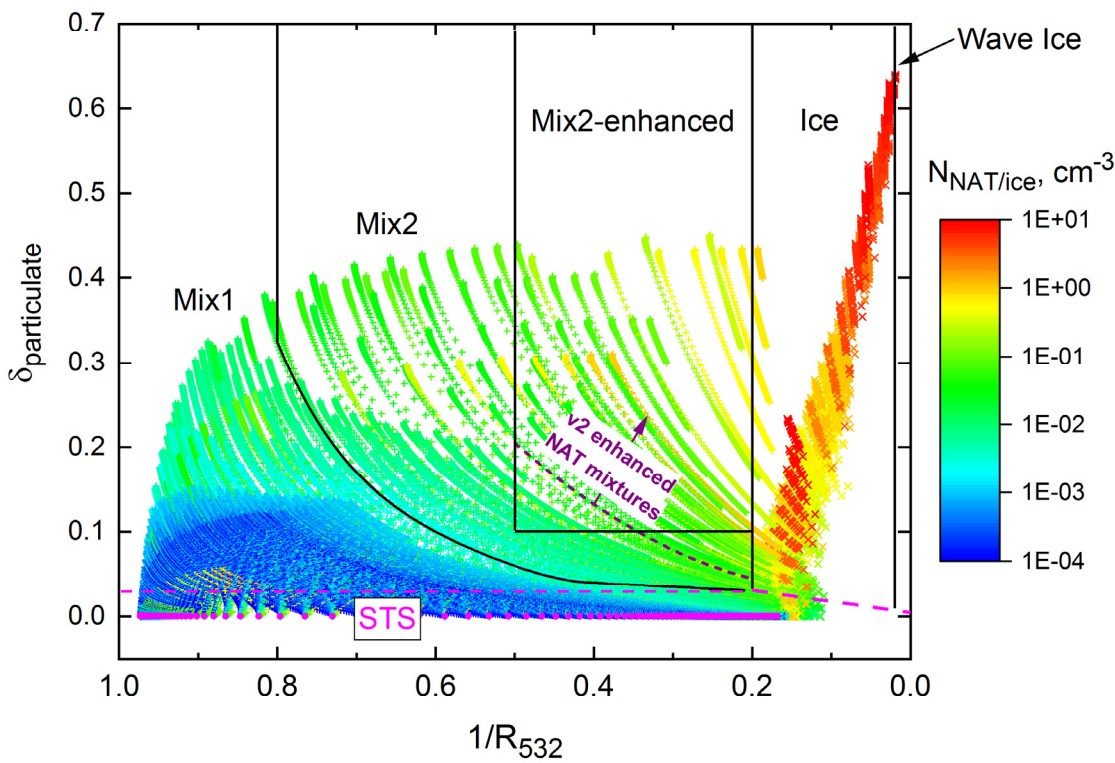

**Figure 3: Theoretical optical calculations for non-equilibrium liquid-NAT and liquid-ice mixtures, illustrating the CALIOP v1 PSC composition classification scheme. The dashed purple curve represents the lower boundary of the v2 enhanced NAT mixtures composition class, as discussed in the text and in Fig. 4.**

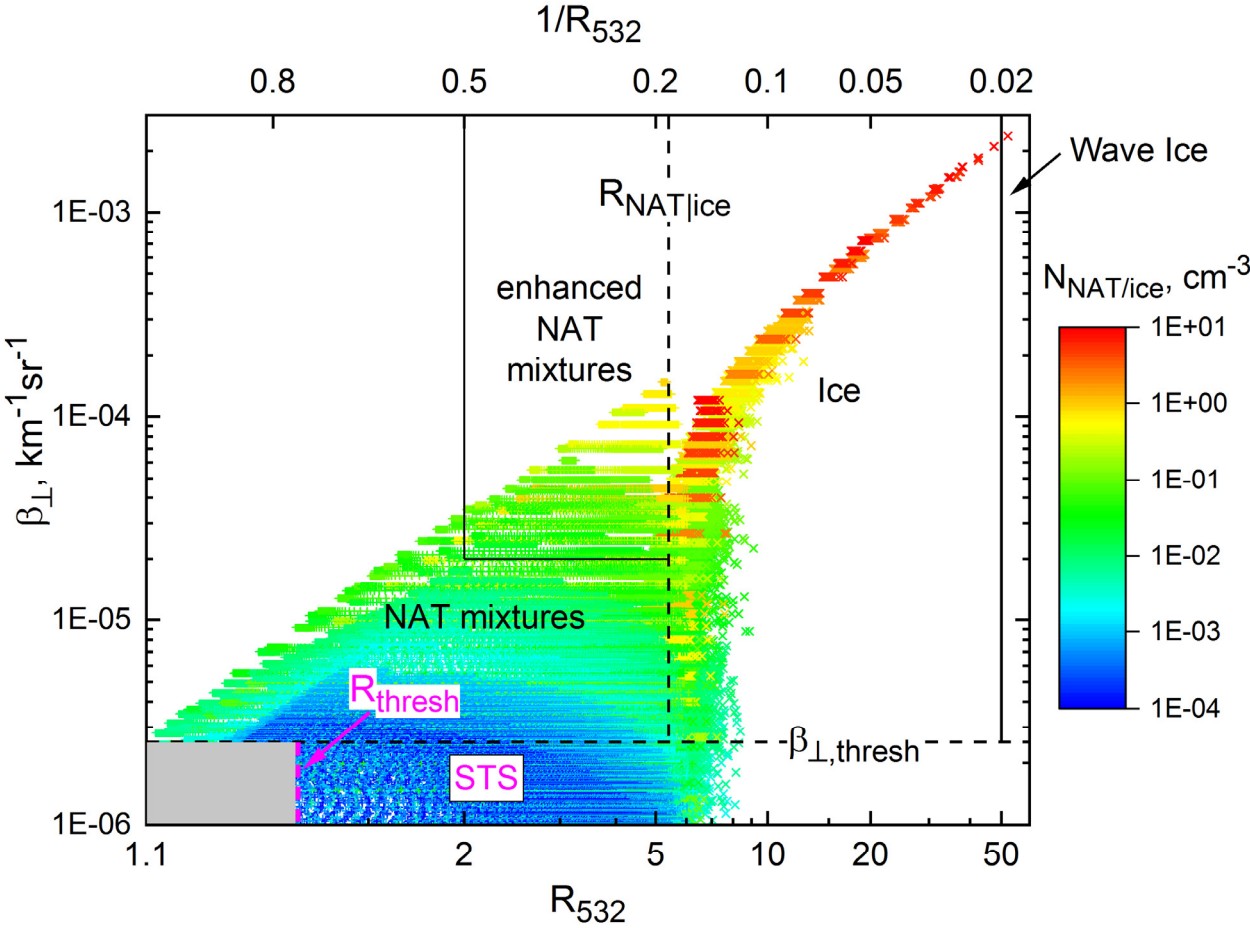

**Figure 4: Theoretical optical calculations for non-equilibrium liquid-NAT and liquid-ice mixtures, illustrating the v2 PSC composition classification scheme. The grey box at the lower left represents points that fall below both CALIOP v2 PSC detection thresholds and are classified as non-features.**

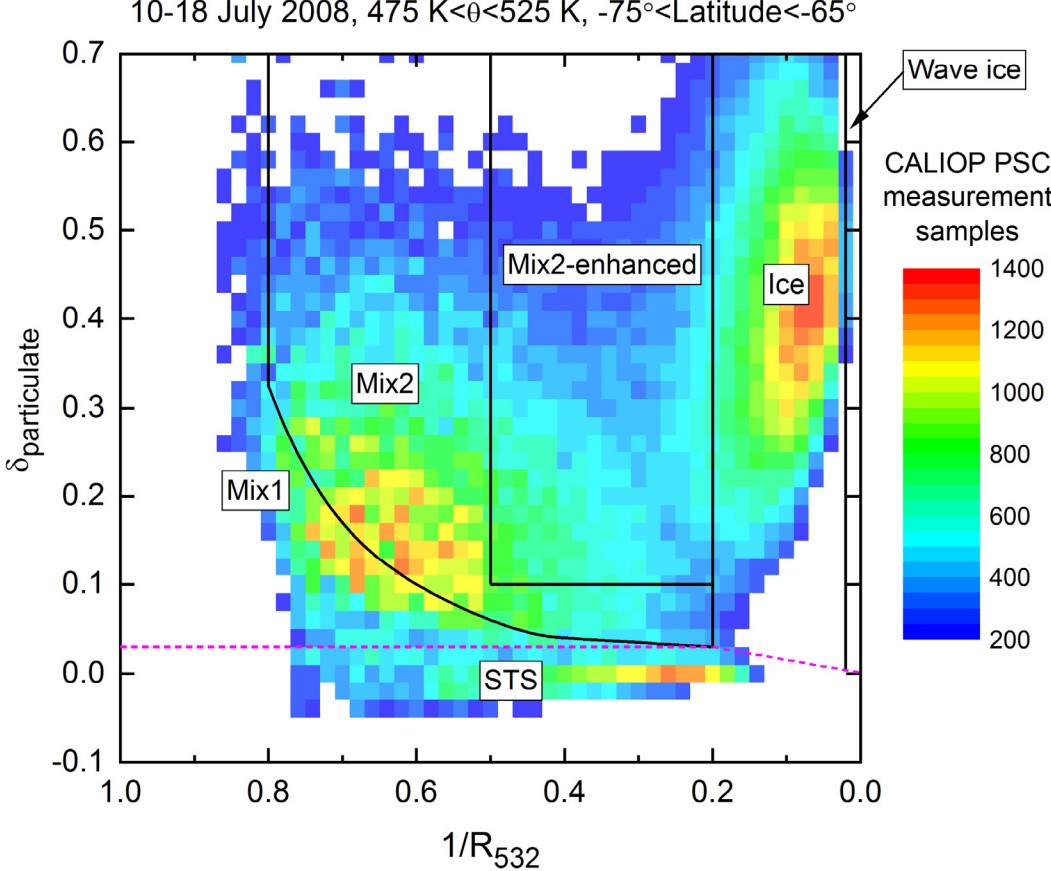

**Figure 5: 2-D histogram of CALIOP PSC data for the period 10-18 July 2008, latitudes 65°S-75°S, and potential temperatures (θ) = 475 K-525 K plotted in the CALIOP v1 composition classification coordinate system.**

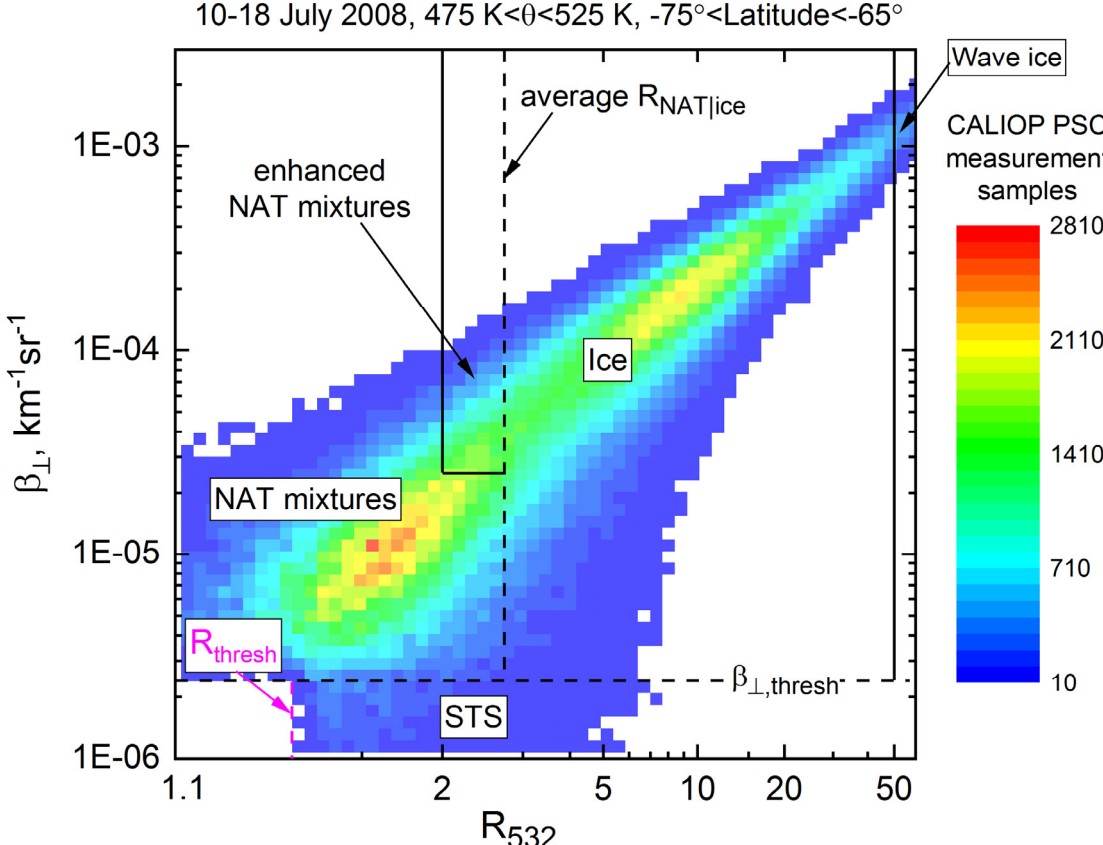

**Figure 6: As in Fig. 5, but plotted in the CALIOP v2 composition classification coordinate system.**

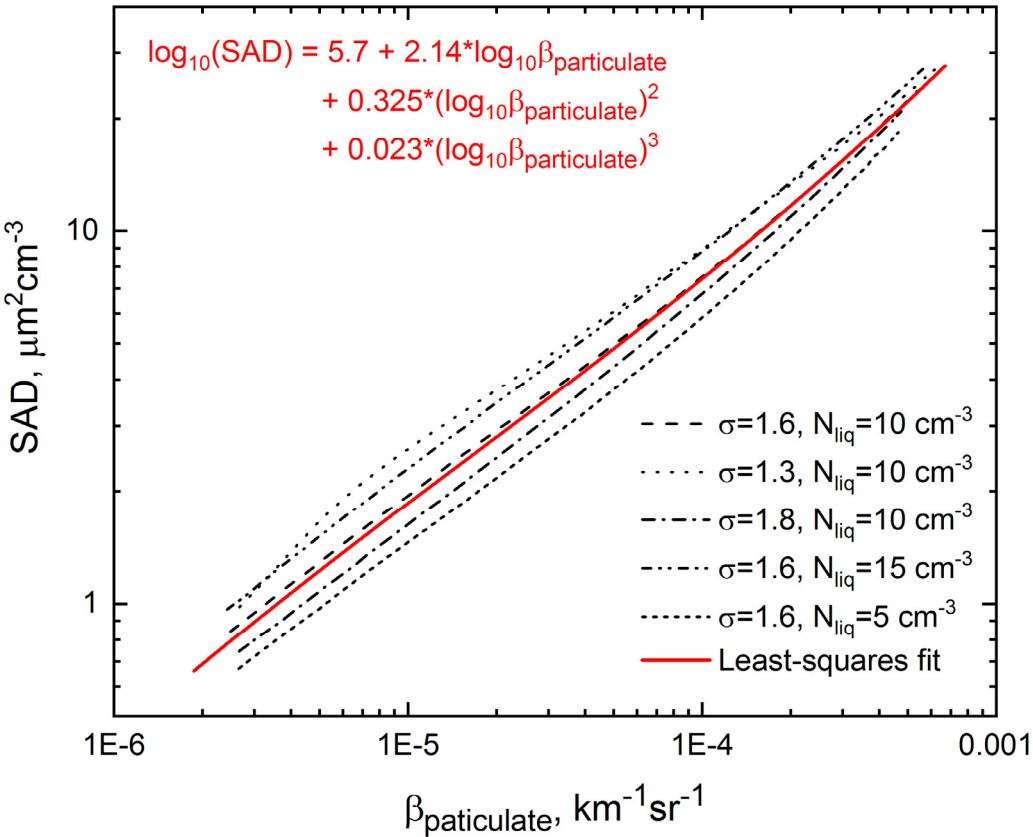

**Figure 7: Theoretical particulate surface area density (SAD) vs. $\beta_{particulate}$ for various combinations of liquid particle number density $N_{liq}$ and lognormal geometric standard deviation $\sigma$, along with the 3rd order polynomial least-squares fit.**

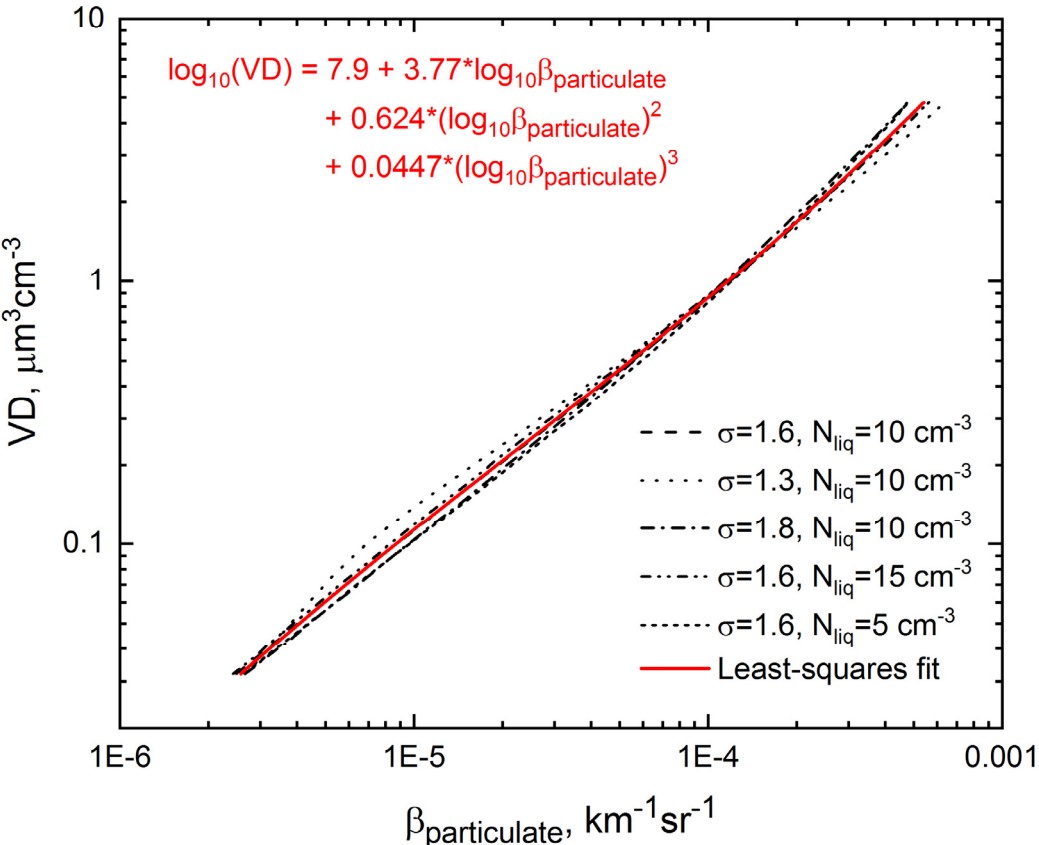

**Figure 8: Theoretical particulate volume density (VD) vs. $\beta_{particulate}$ for various combinations of liquid particle number density $N_{liq}$ and lognormal geometric standard deviation $\sigma$, along with the 3rd order polynomial least-squares fit.**

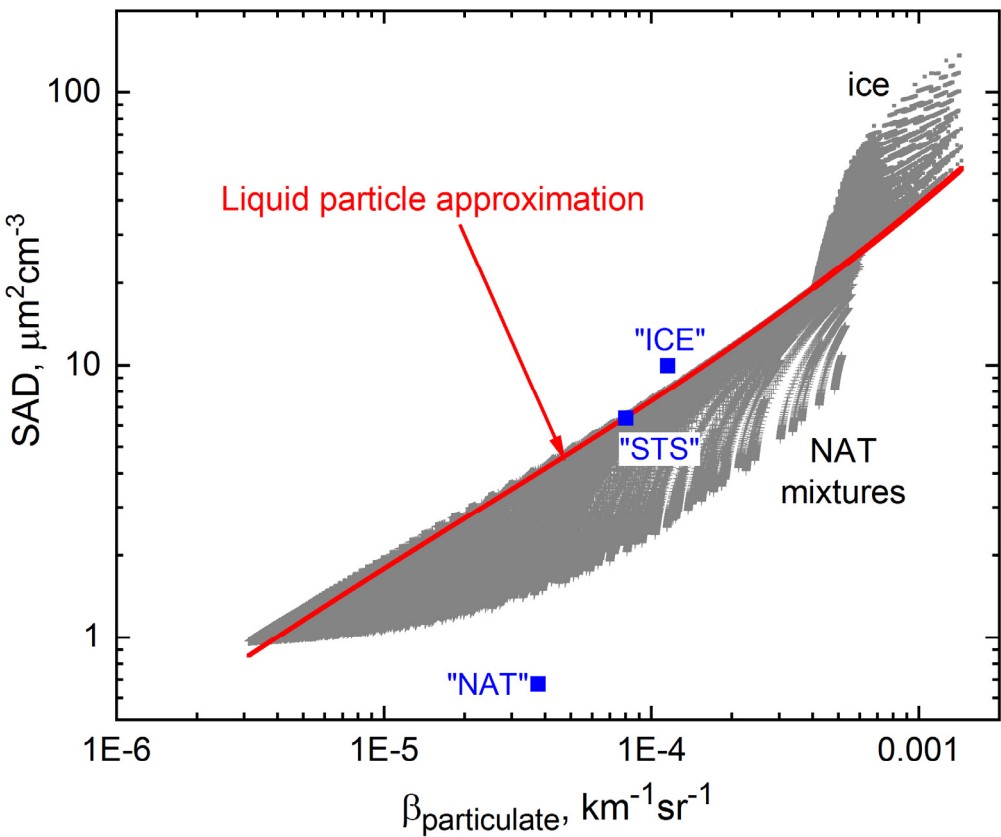

**Figure 9: Theoretical particulate surface area density (SAD) vs. $\beta_{particulate}$ from the full suite of results for NAT mixtures and ice, compared with the liquid particle approximation. Blue symbols are computed values based on bimodal lognormal size distribution fits to in situ optical particle counter measurements within STS, NAT, and ice PSC layers (Deshler et al., 2003).**

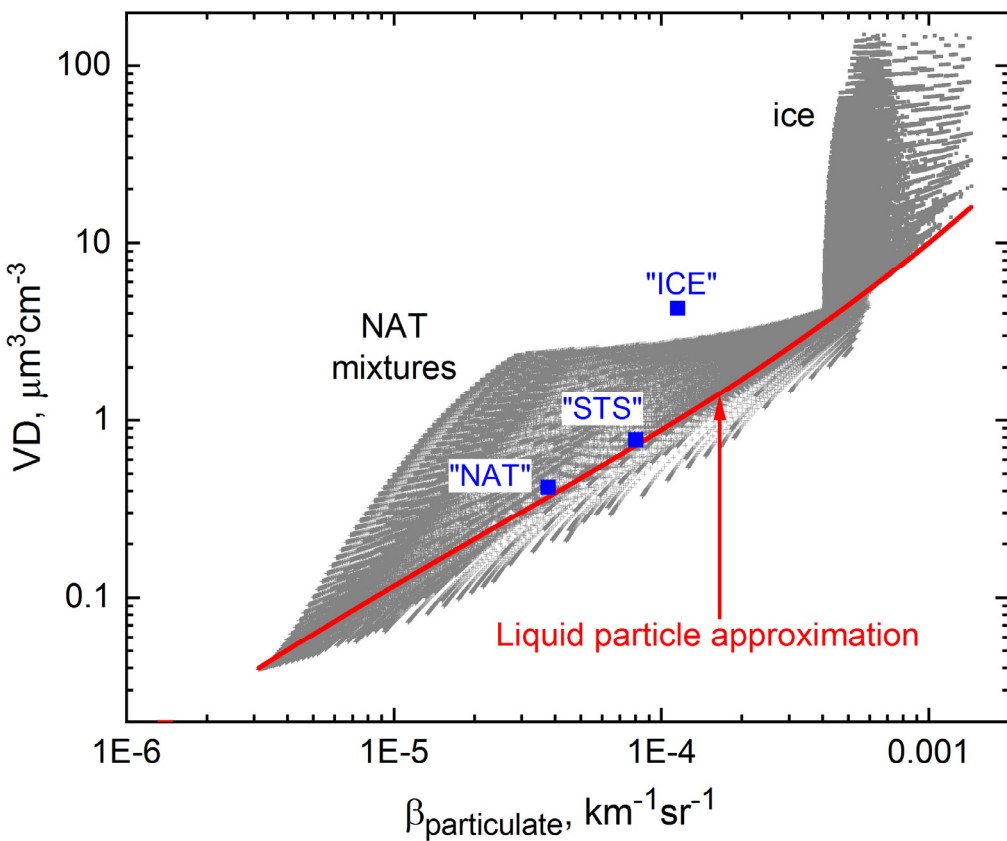

**Figure 10: Theoretical particulate volume density (VD) vs. $\beta_{particulate}$ from the full suite of results for NAT mixtures and ice, compared with the liquid particle approximation. Blue symbols are computed values based on bimodal lognormal size distribution fits to in situ optical particle counter measurements within STS, NAT, and ice PSC layers (Deshler et al., 2003).**

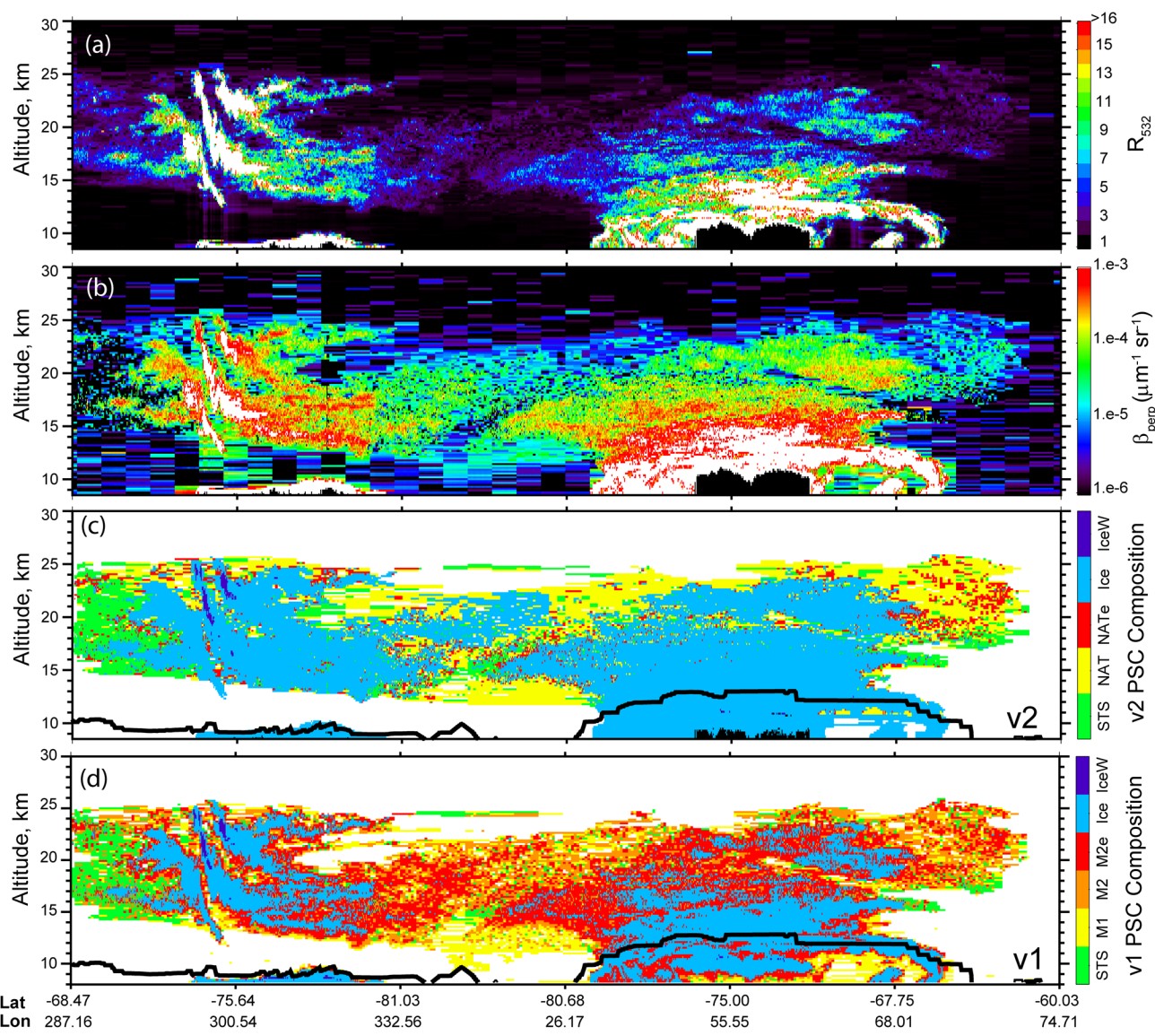

**Figure 11:** Curtains of CALIOP (a) $R_{532}$, (b) $\beta_\perp$, (c) v2 PSC composition mask, and (d) v1 PSC composition mask along the orbit track on 17 July 2008 highlighted in green in Fig.1.  The location of the MERRA-2 "blended" tropopause is shown in panels (c) and (d) by the solid black line.

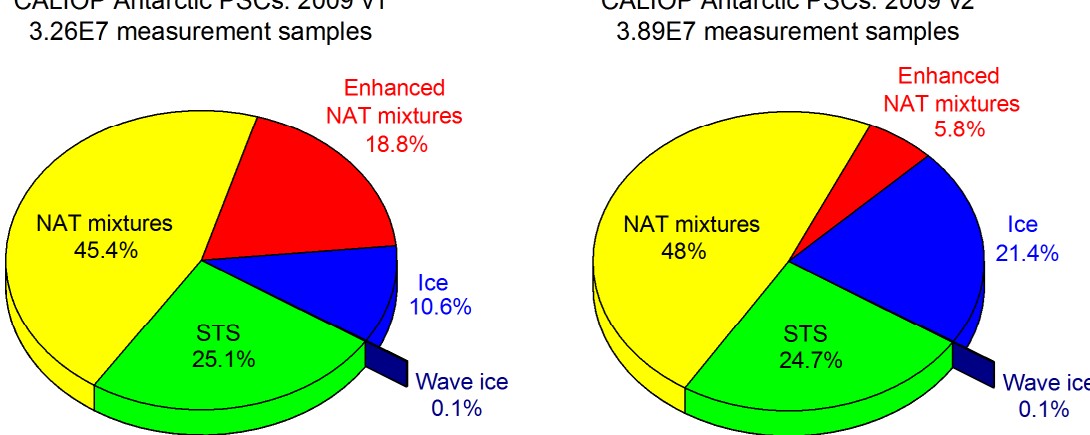

**Figure 12: Comparison between v1 and v2 CALIOP PSC measurements during the 2009 Antarctic winter and their breakdown by composition classification. Data are restricted to altitudes 4 km or more above the tropopause to avoid distortion of statistics by ubiquitous underlying cirrus.**

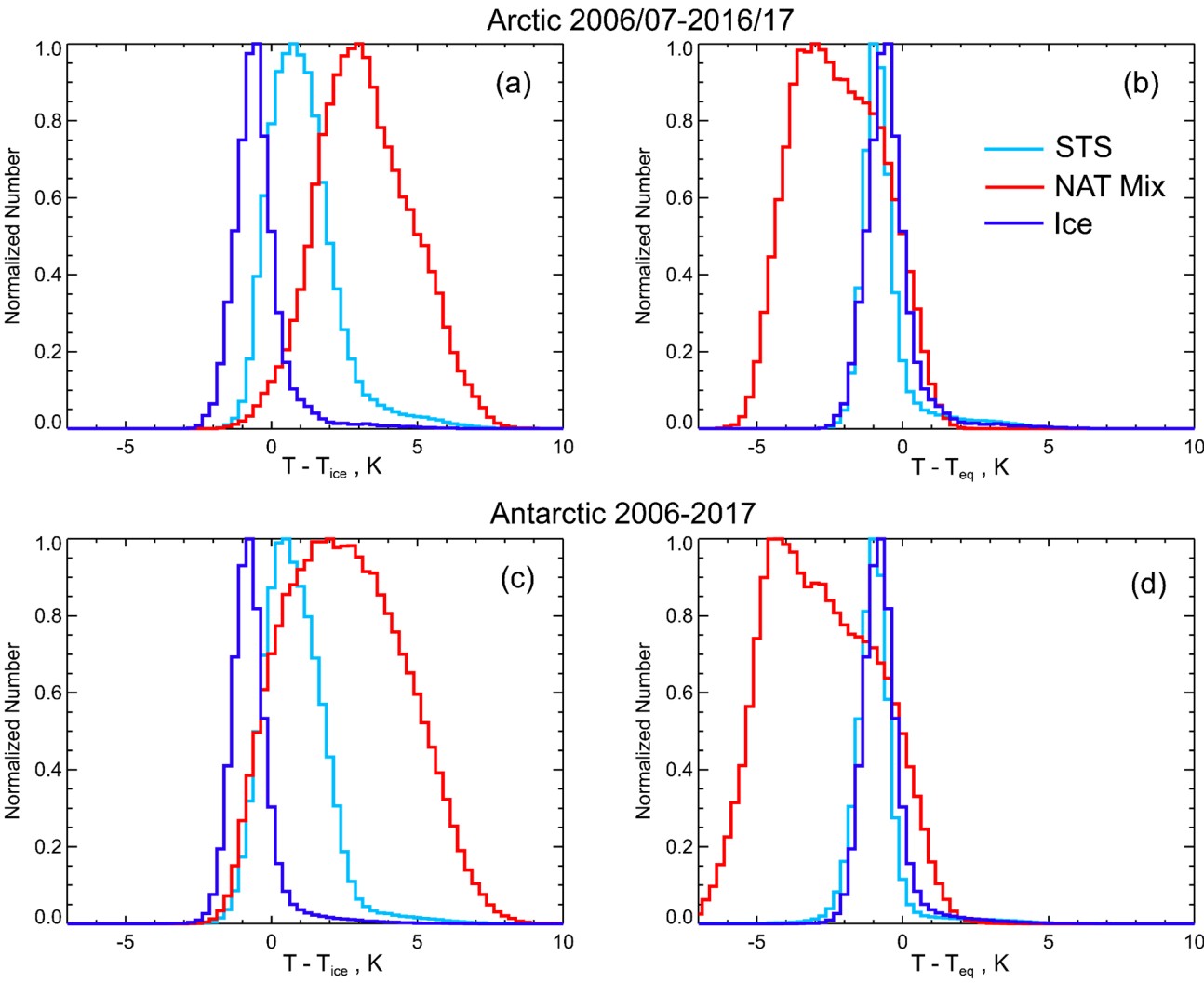

**Figure 13: Histograms of CALIOP v2 PSC measurements at 21 km from 11 Arctic and 12 Antarctic winters as a function of (a, c)** $T$-$T_{ice}$ **and (b, d)** $T$-$T_{eq}$ **by composition: STS (light blue); NAT mixtures, including enhanced NAT mixtures (red); and ice, including wave ice (dark blue).**

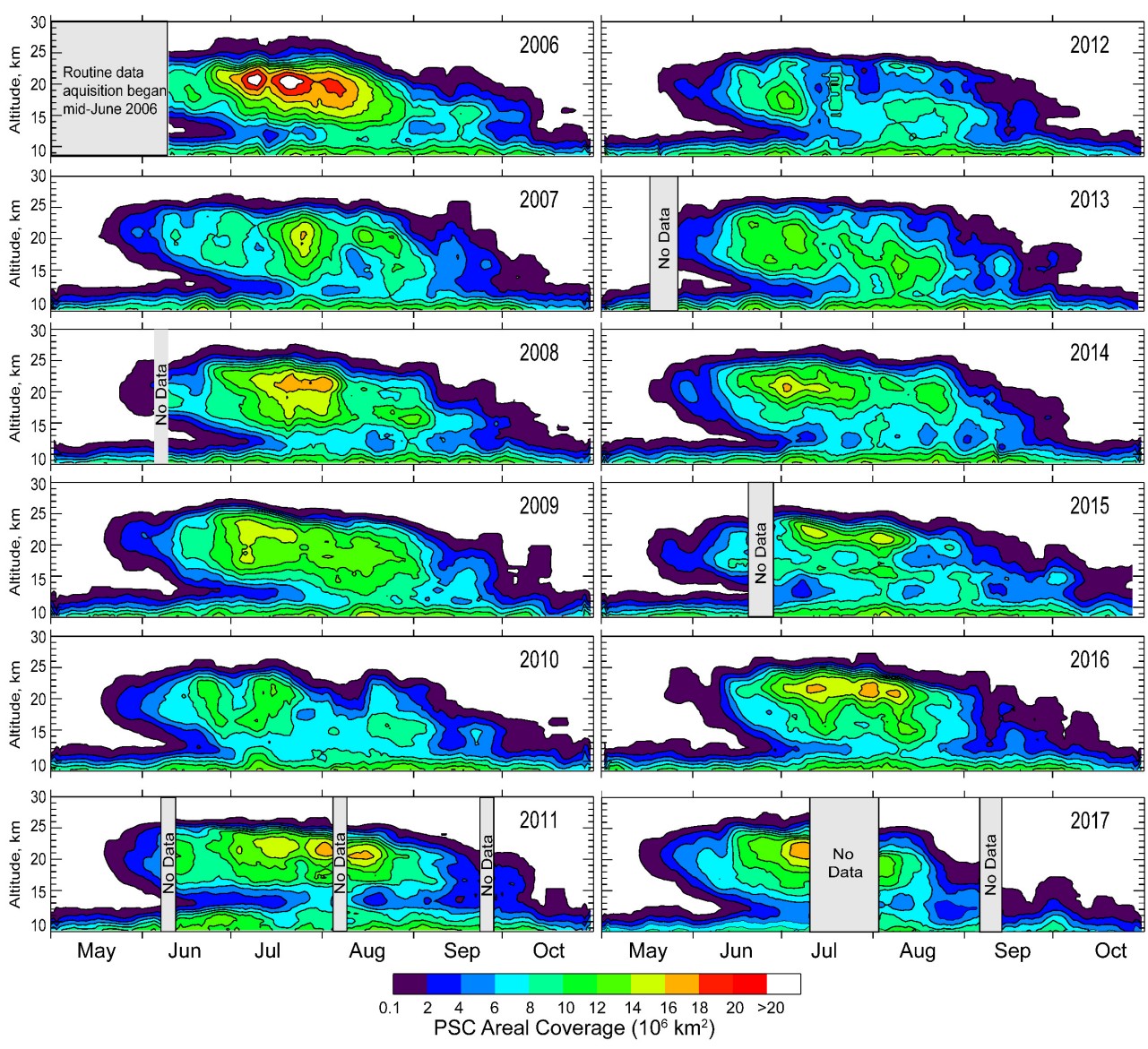

**Figure 14: Time series of total PSC areal coverage over the Antarctic region as a function of altitude for each of the 12 Antarctic winters in the CALIOP v2 data record.**

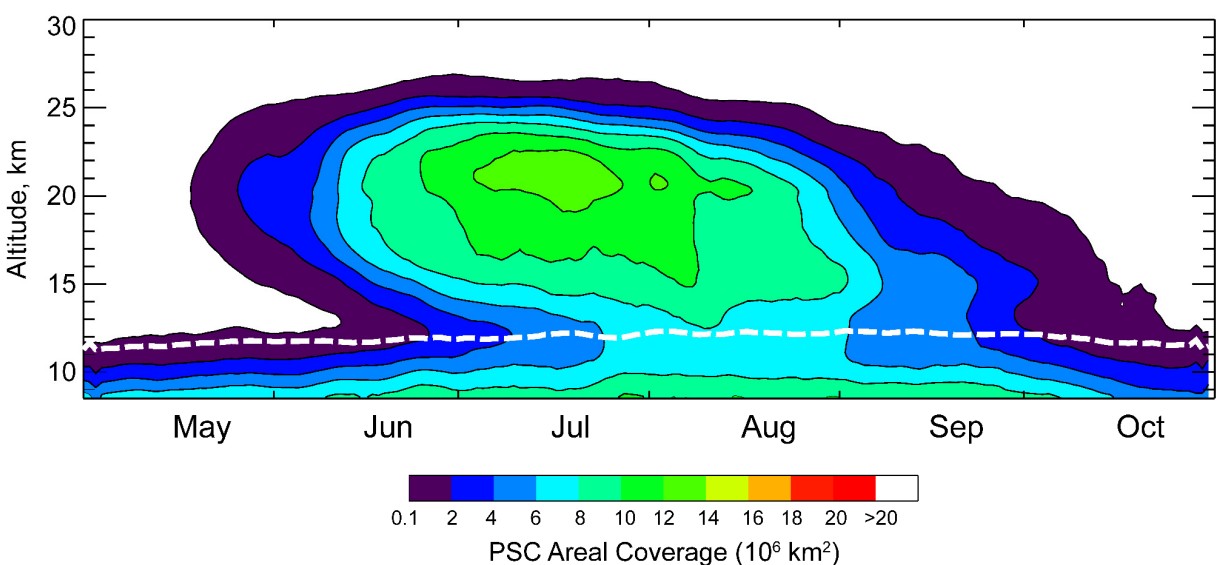

**Figure 15: Twelve-year mean daily PSC areal coverage over the Antarctic. The climatological daily maximum MERRA-2 tropopause height is indicated by the dashed white line.**

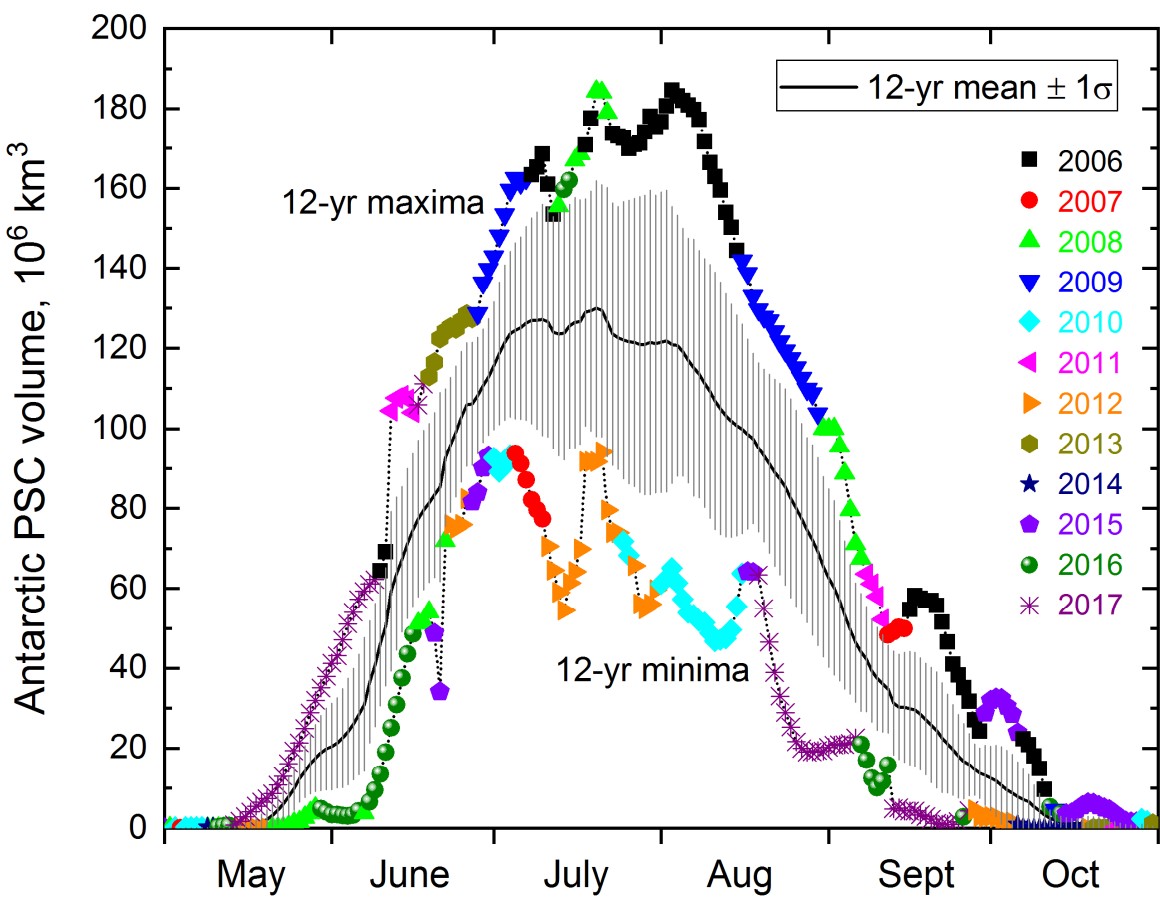

**Figure 16: Time series of the 12-year mean, standard deviation, and range of daily values of Antarctic PSC spatial volume (daily areal coverage integrated over altitude). The daily maximum and minimum values are color-coded according to the year in which they occurred.**

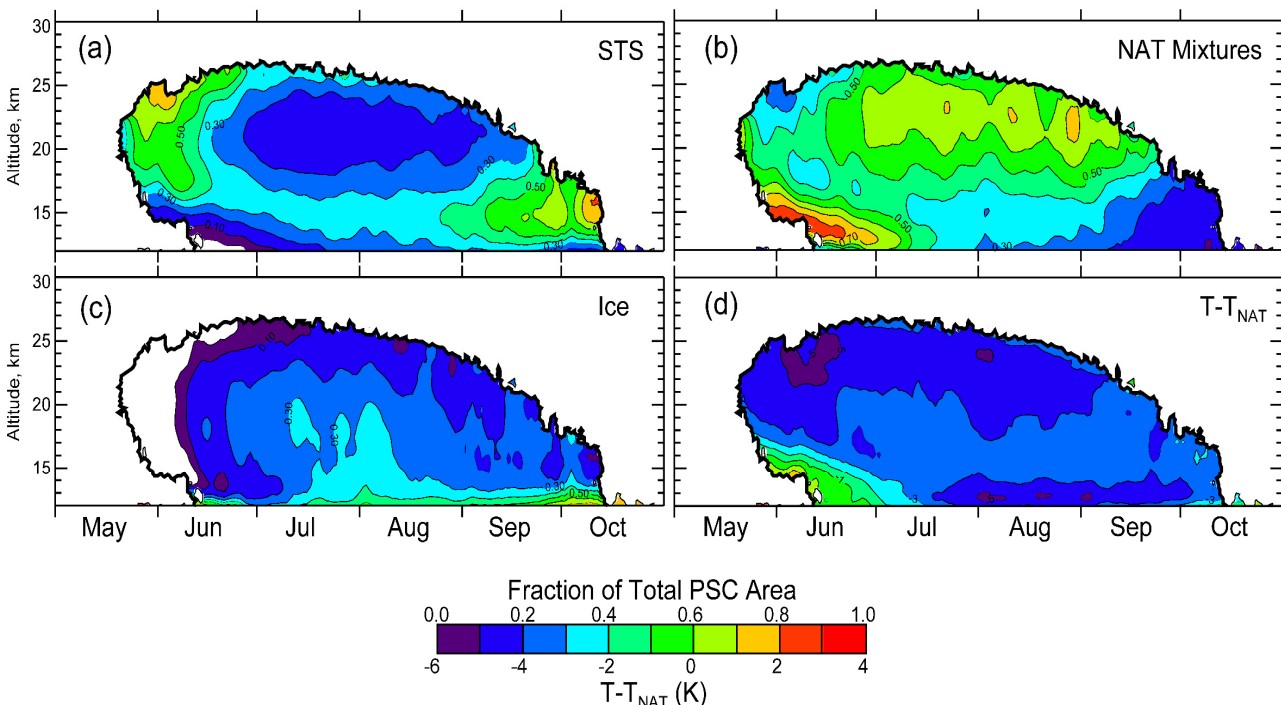

**Figure 17: Twelve-year mean relative spatial coverage (composition-specific area normalized by total PSC area) for (a) STS; (b) NAT mixtures, including enhanced NAT mixtures; and (c) ice, including wave ice. For additional perspective, (d) shows 12-year mean distribution of $T$-$T_{NAT}$. The thick black contour line on each of the color panels corresponds to the range of days and altitudes where PSCs (of any composition) were observed in at least six of the twelve Antarctic seasons.**

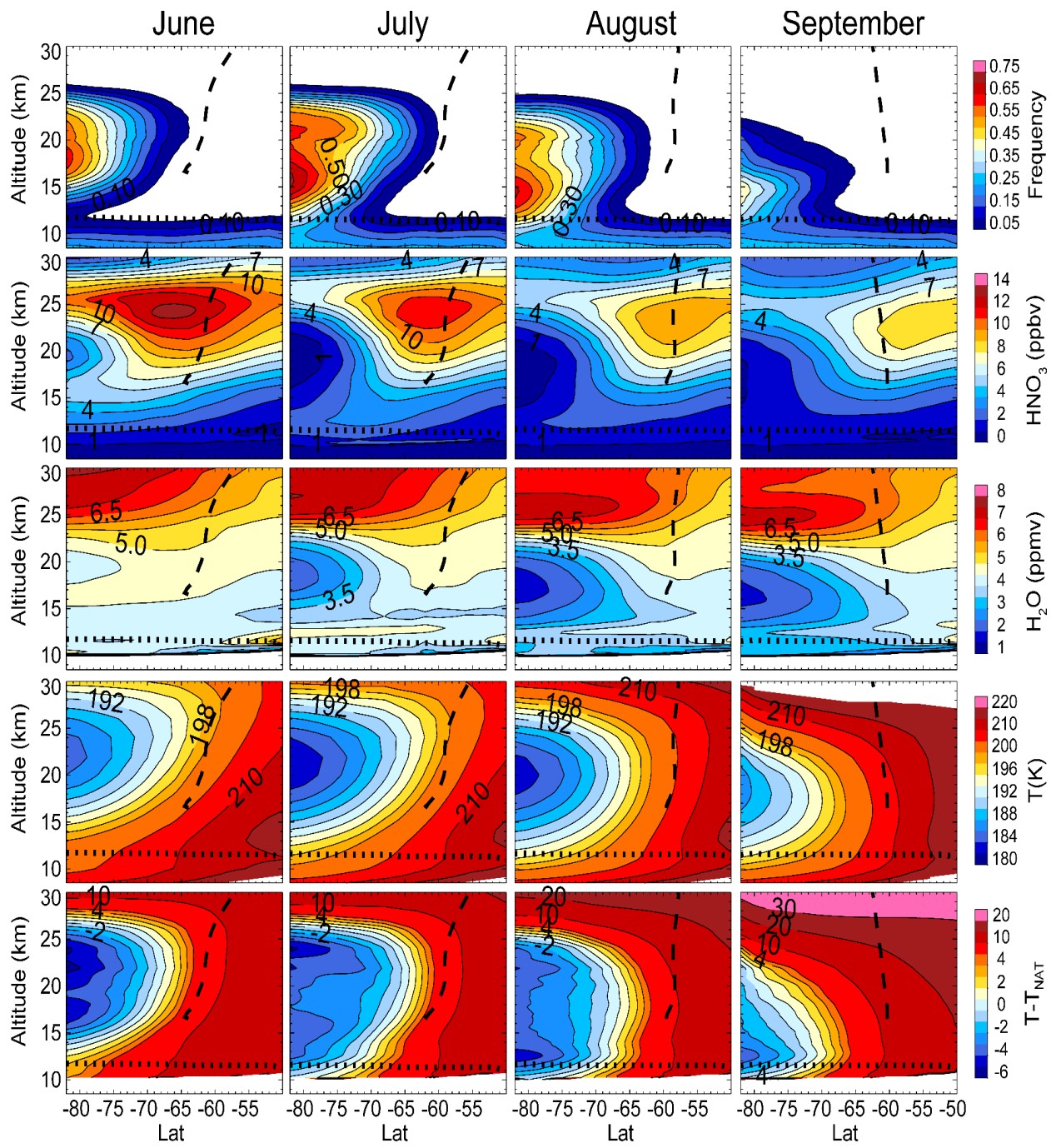

**Figure 18: Latitude/altitude cross sections of 12-year average, monthly zonal mean (top row) Antarctic PSC occurrence frequency, (second row) cloud-free MLS HNO₃, (third row) cloud-free MLS H₂O, (fourth row) MERRA-2 temperature, and (fifth row) *T-T*ₙₐₜ. For reference, the mean location of the vortex edge (heavy dashed line) and MERRA-2 tropopause height (dotted line) are indicated in the panels.**

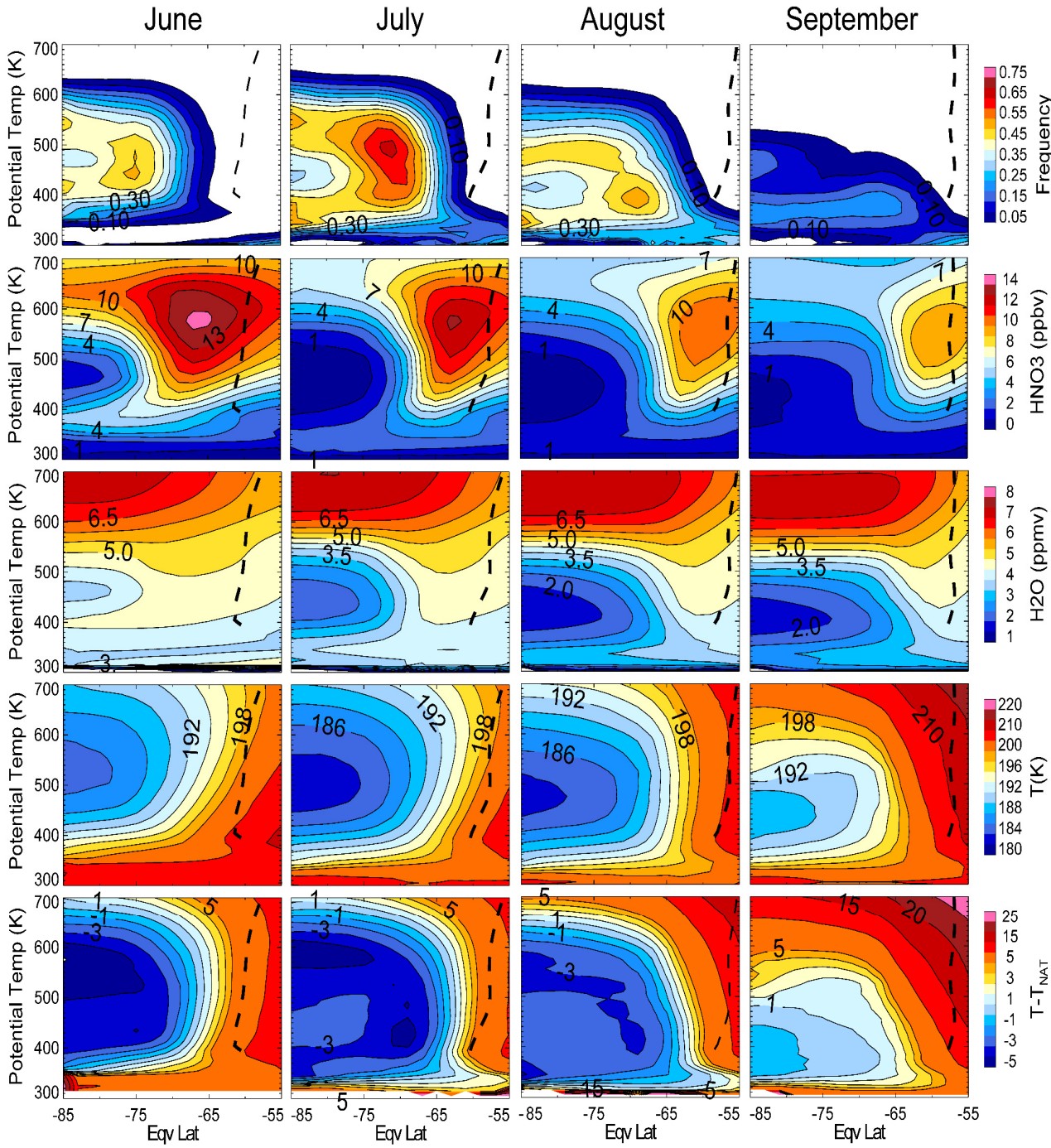

**Figure 19: Equivalent latitude/potential temperature cross sections of 12-year average, monthly zonal mean (top row) Antarctic PSC occurrence frequency, (second row) cloud-free MLS HNO₃, (third row) cloud-free MLS H₂O, (fourth row) MERRA-2 temperature, and (fifth row) *T-T*ₙₐₜ. For reference, the mean location of the vortex edge (heavy dashed line) is indicated in the panels.**

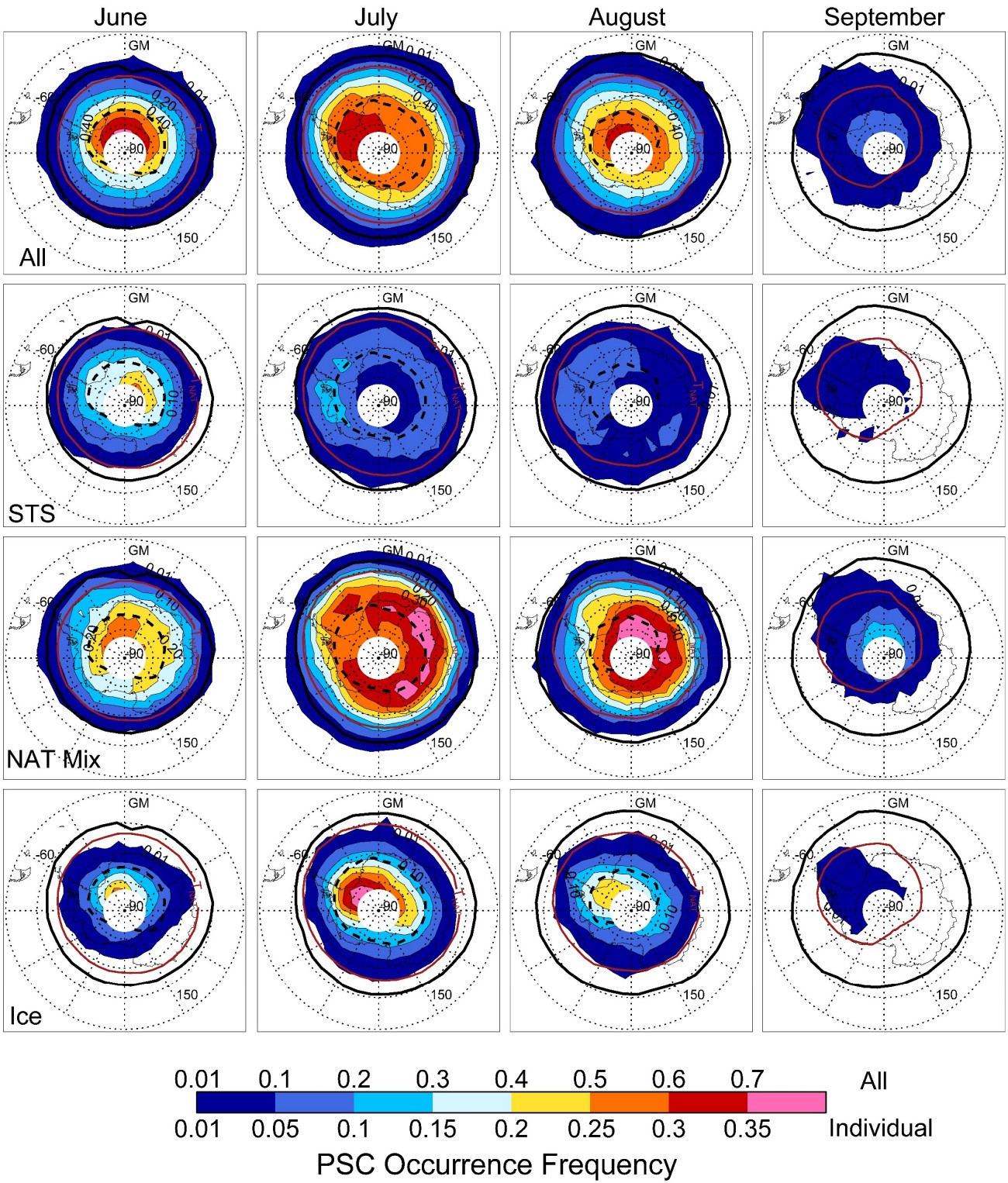

**Figure 20: Twelve-year average, monthly mean polar maps of Antarctic PSC occurrence frequency at θ=500 K (~20 km). The top row shows the occurrence frequency for all PSCs, while the subsequent rows display the occurrence frequencies of STS, NAT mixtures (including enhanced NAT), and ice (including wave ice), respectively. Overlaid in the panels are the mean location of the edge of the vortex (black line) and the boundaries of the regions where the mean temperature is below $T_{NAT}$ (solid red line) and below $T_{ice}$ (dashed black line). Light gray regions indicate latitudes not sampled by CALIOP. Note the different color scales for "all" PSCs and "individual" compositions.**

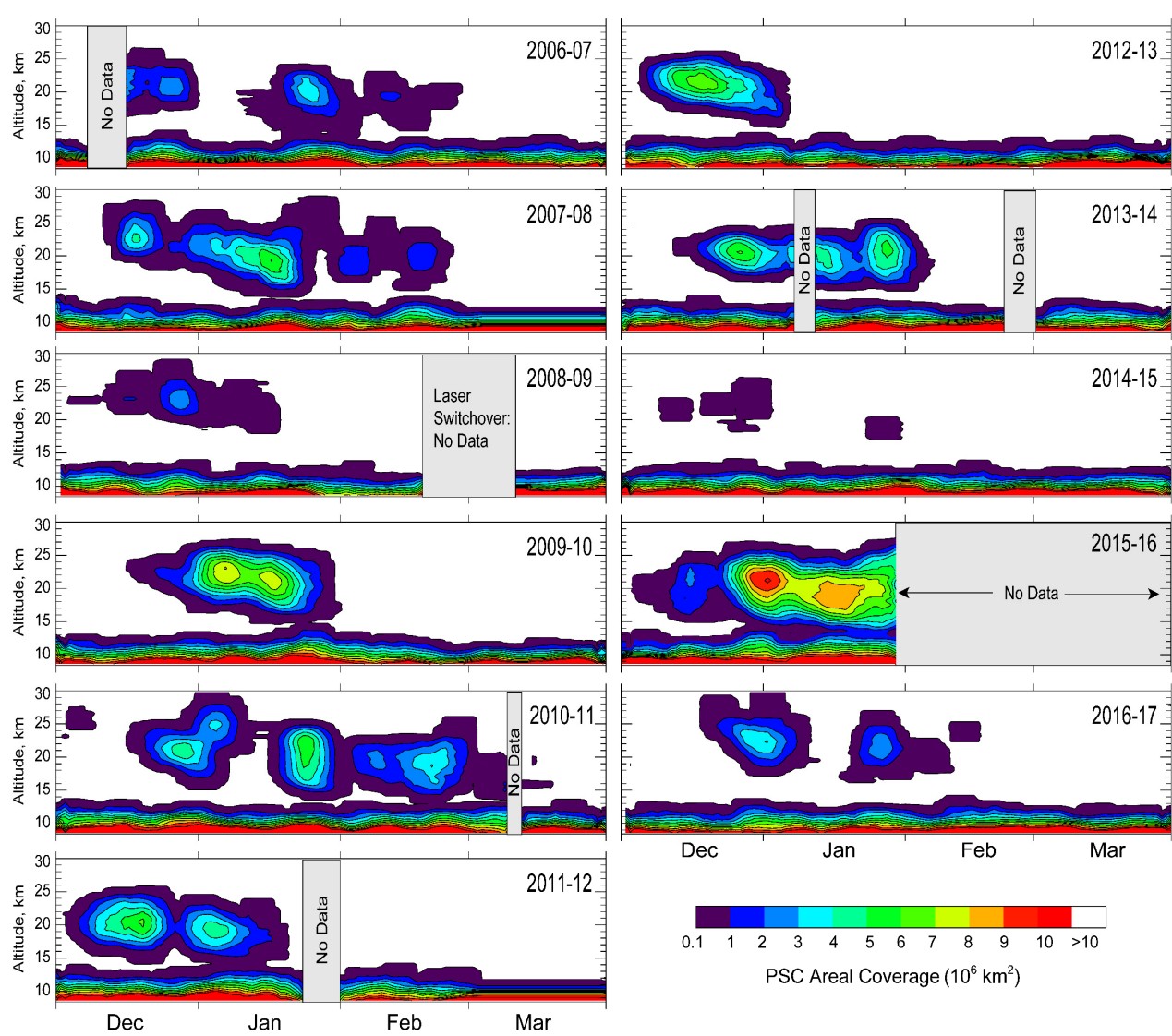

**Figure 21: Daily PSC areal coverage as a function of altitude for each of the 11 Arctic winters in the CALIOP v2 data record. Note the change in color scale compared with Fig. 14.**

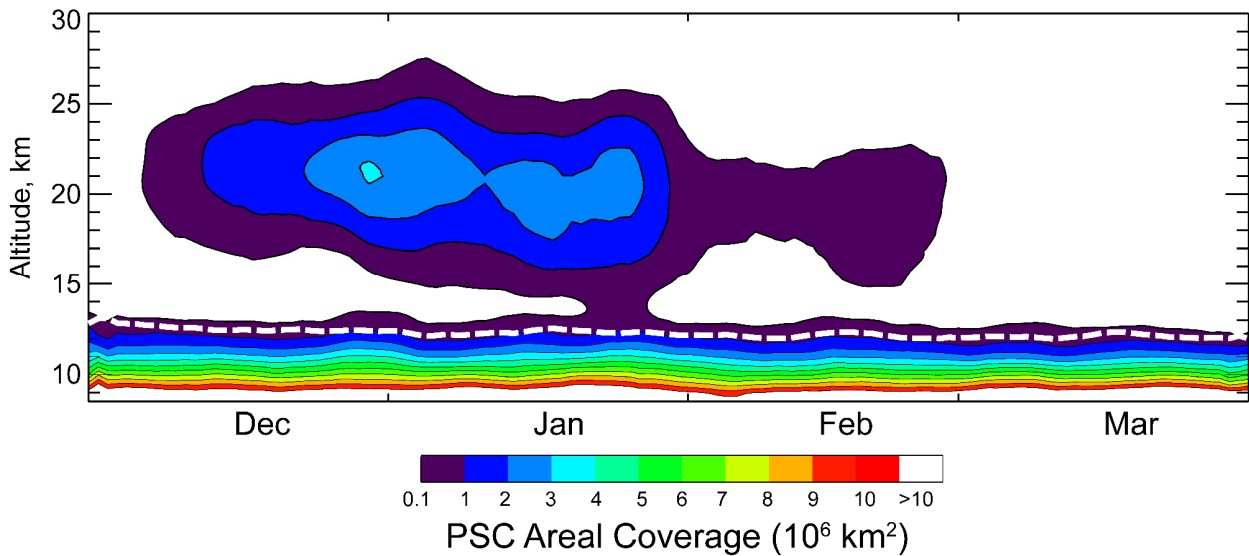

**Figure 22: Eleven-year mean daily PSC areal coverage over the Arctic. The climatological daily maximum MERRA-2 tropopause height is indicated by the dashed white line.**

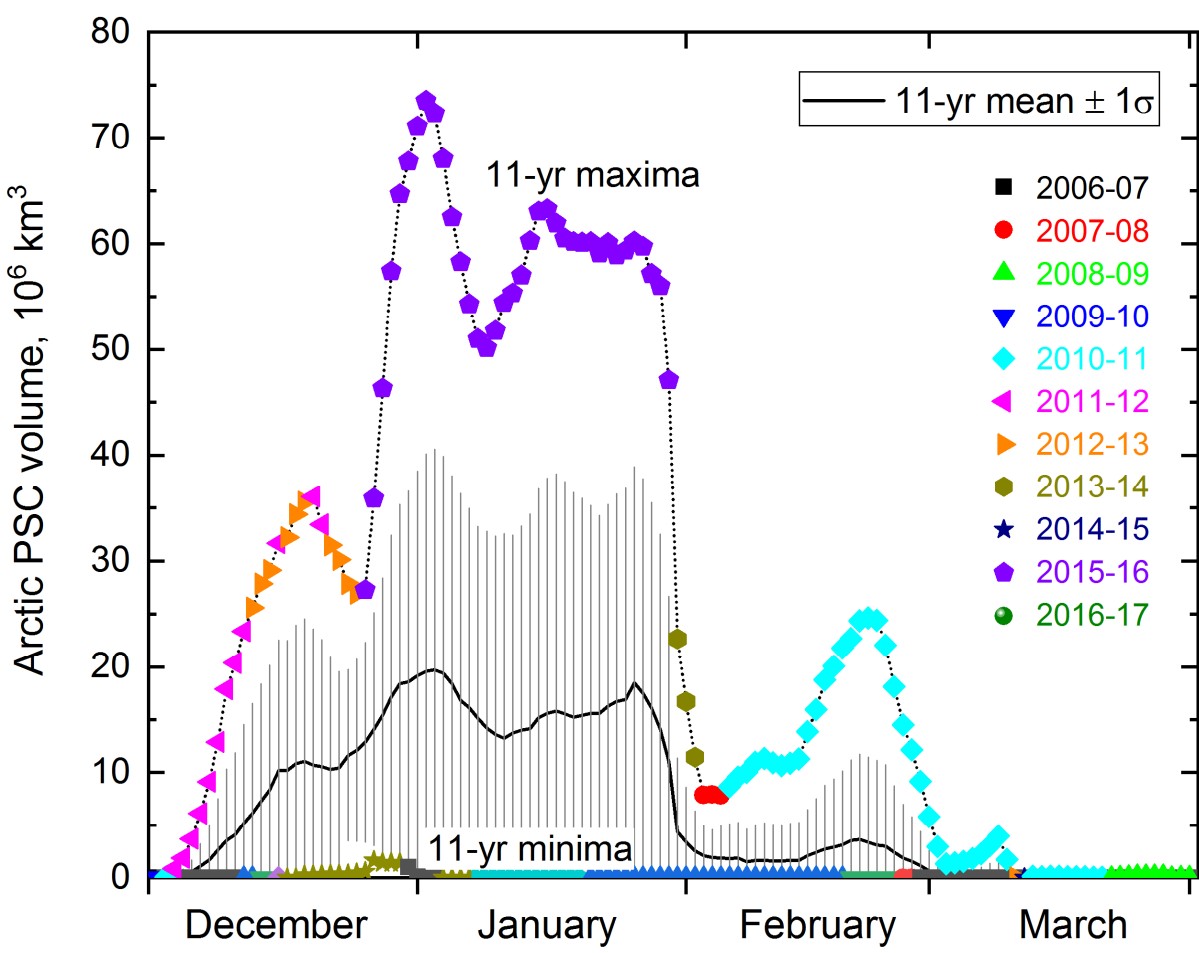

**Figure 23: Time series of the 11-year mean, standard deviation, and range of daily values of Arctic PSC spatial volume (daily areal coverage integrated over altitude). The daily maximum and minimum values are color-coded according to the year in which they occurred.**

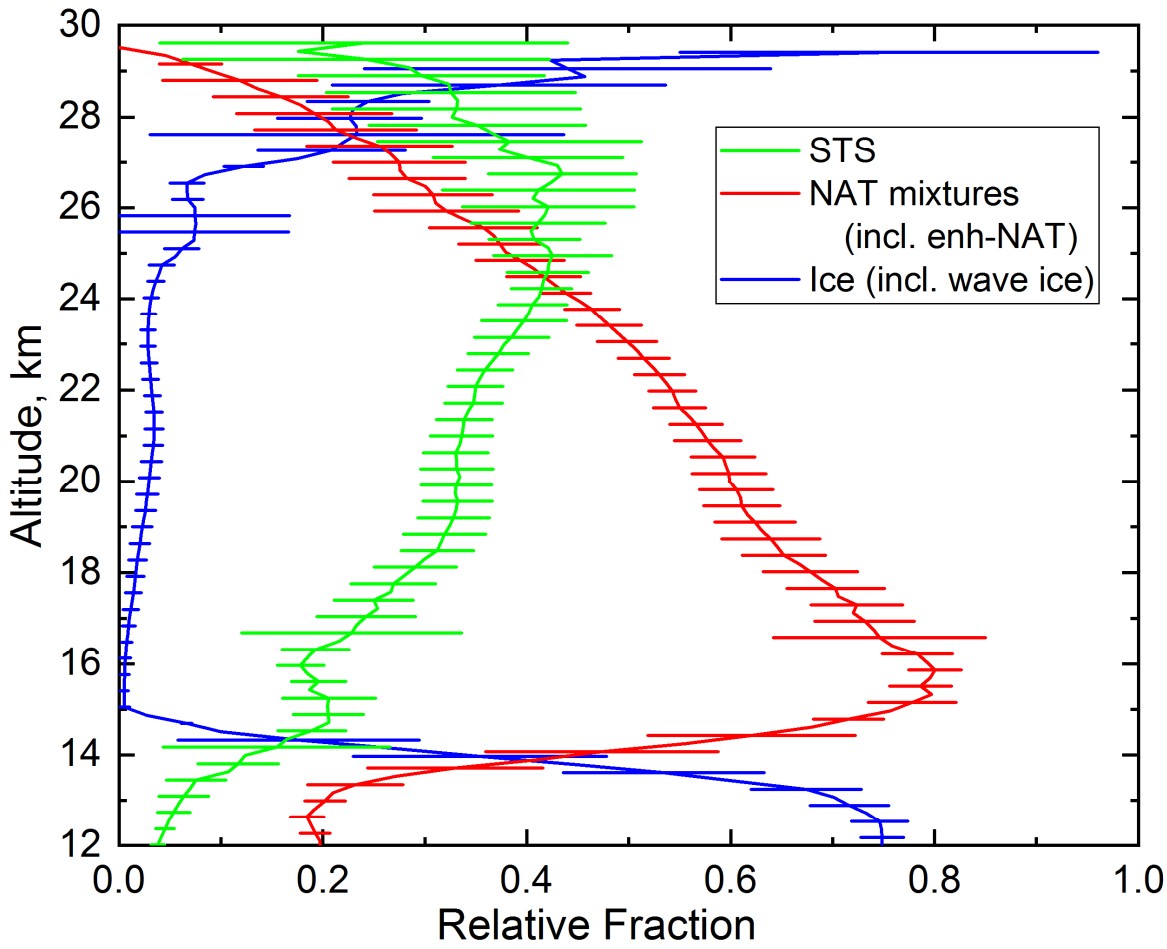

**Figure 24: Vertical profile of 11-year mean season-long relative spatial coverage (composition-specific area normalized by total PSC area) of Arctic PSCs by composition. Horizontal bars show the standard error in the mean values and are offset by 0.1 km to avoid overlap.**

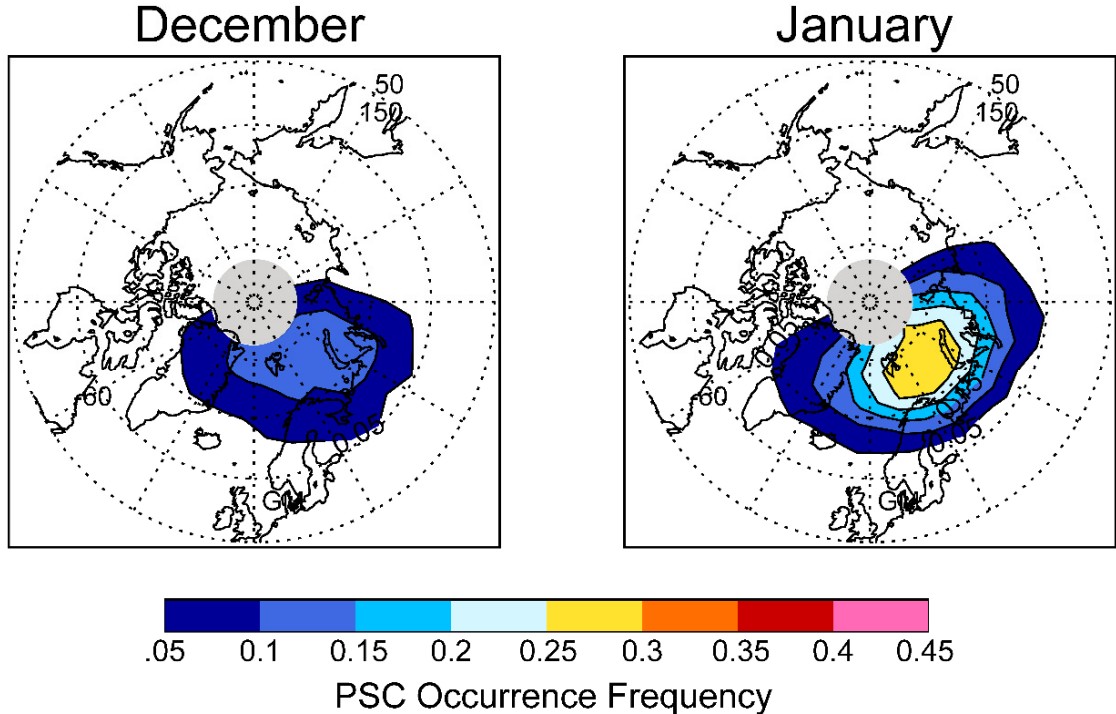

**Figure 25: Eleven-year average, monthly mean polar maps of Arctic PSC occurrence frequency at θ=500 K (~20 km). Light gray regions indicate latitudes not sampled by CALIOP.**

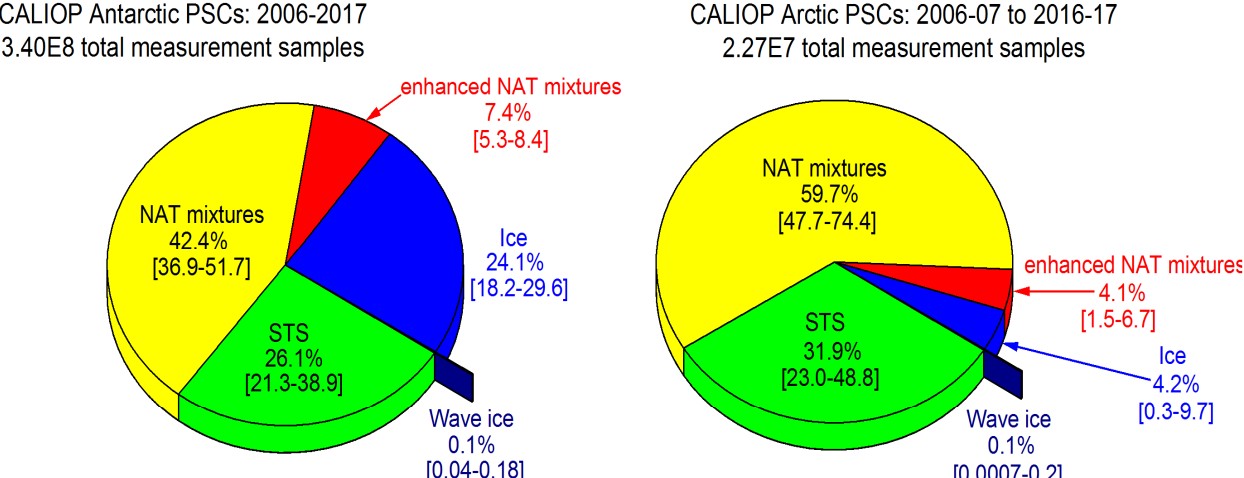

**Figure 26: Composition breakdown of CALIOP PSC measurements in the Antarctic and Arctic during 2006–2017. The percentages are averages over the 12 Antarctic and 11 Arctic seasons in the data record, and the minimum and maximum percentages in any one season are indicated by the numbers in brackets. Data are restricted to altitudes 4 km or more above the tropopause to avoid distortion of the statistics by ubiquitous underlying cirrus.**

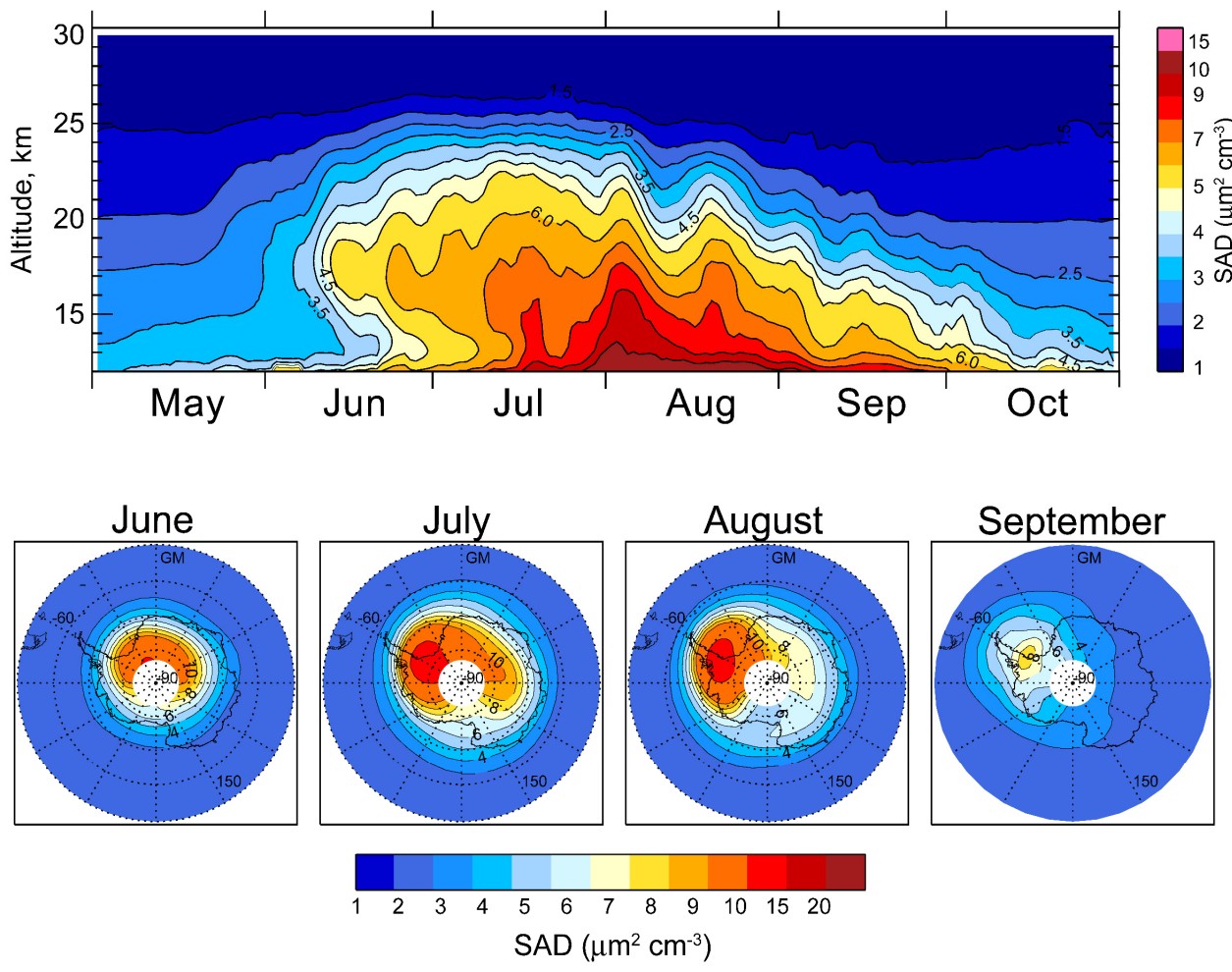

**Figure 27: (Top):** Twelve-year mean seasonal evolution of Antarctic vortex-averaged SAD. **(Bottom):** Twelve-year average, monthly mean polar maps of SAD over the Antarctic at 18 km for June-September.

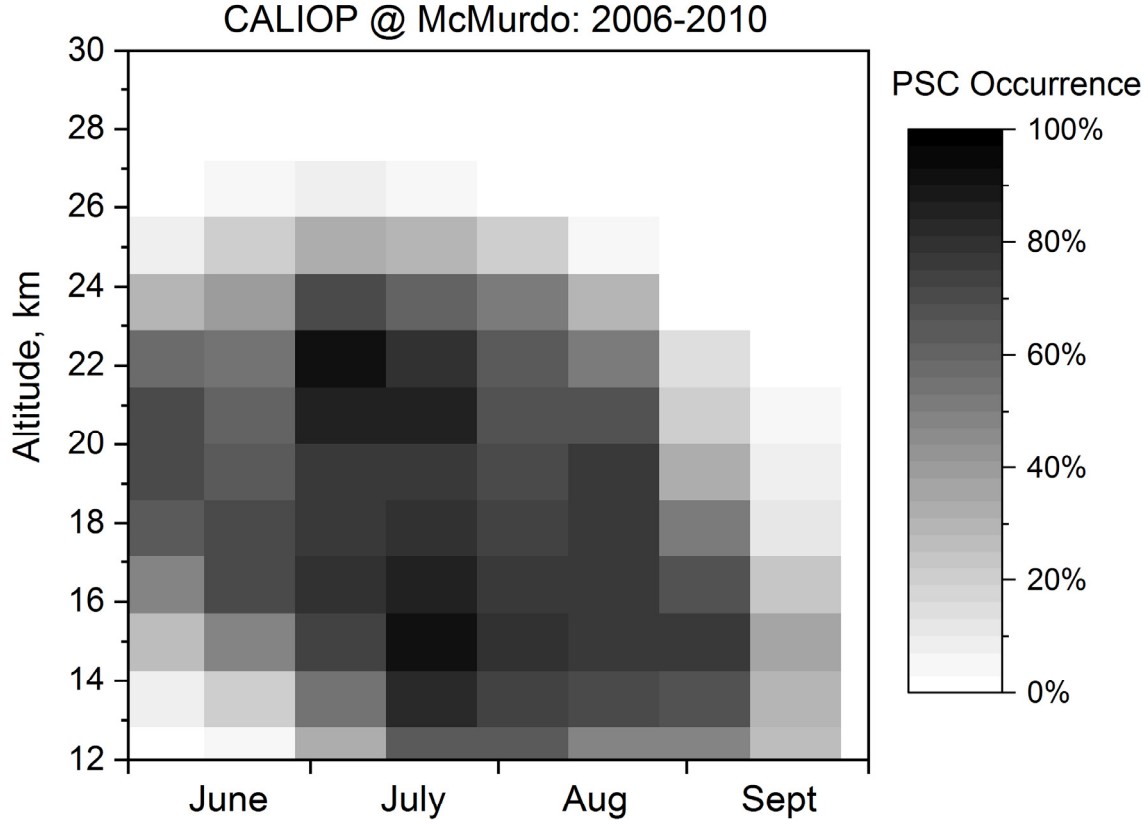

**Figure 28: CALIOP v2 PSC occurrence frequencies near McMurdo Station, Antarctica, for June-September, 2006-2010 (in 15-d×1.5-km bins) for comparison with ground-based lidar data for the same period shown in Di Liberto et al. (2014).**

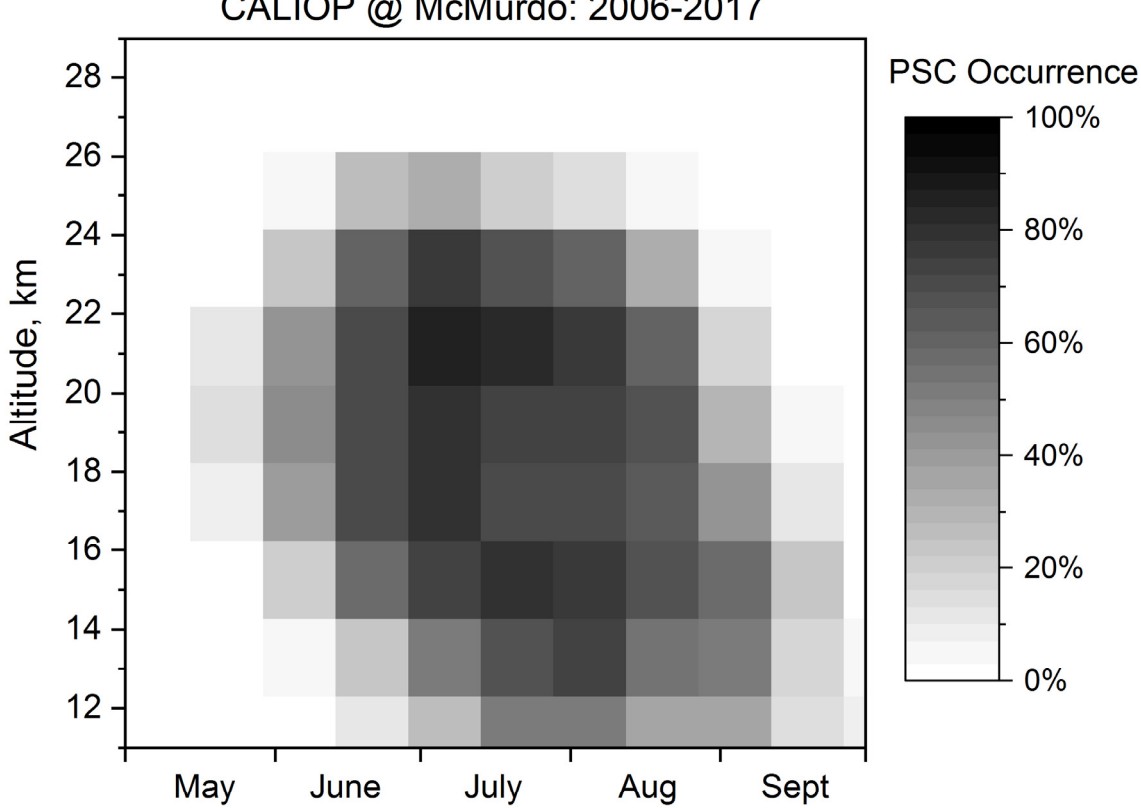

**Figure 29: CALIOP v2 PSC occurrence frequencies near McMurdo Station, Antarctica, for May-September, 2006-2017 (in 15-d×2-km bins) for comparison with ground-based lidar data from 1995-2001 shown in Massoli et al. (2006).**

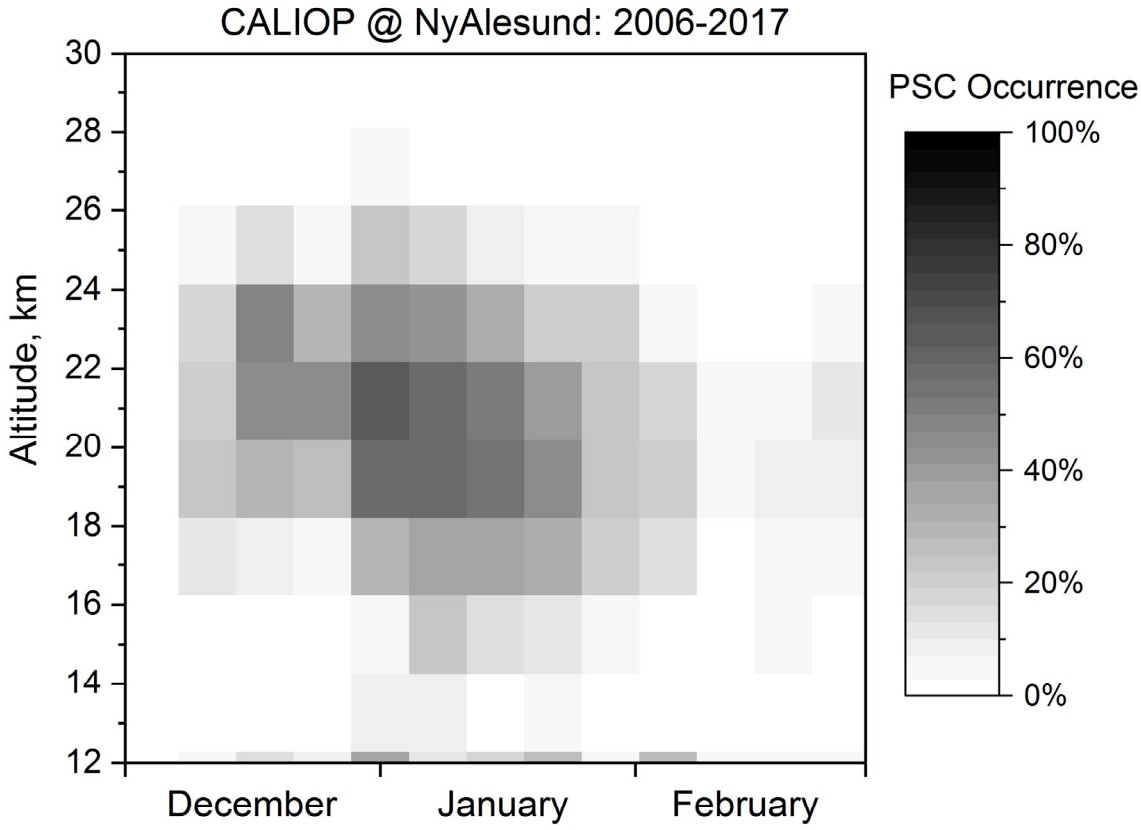

**Figure 30: CALIOP v2 PSC occurrence frequencies near Ny-Ålesund, Spitsbergen, for December-February, 2006-2017 (in 7-d×2-km bins) for comparison with ground-based lidar data from 1995-2003 shown in Massoli et al. (2006).**

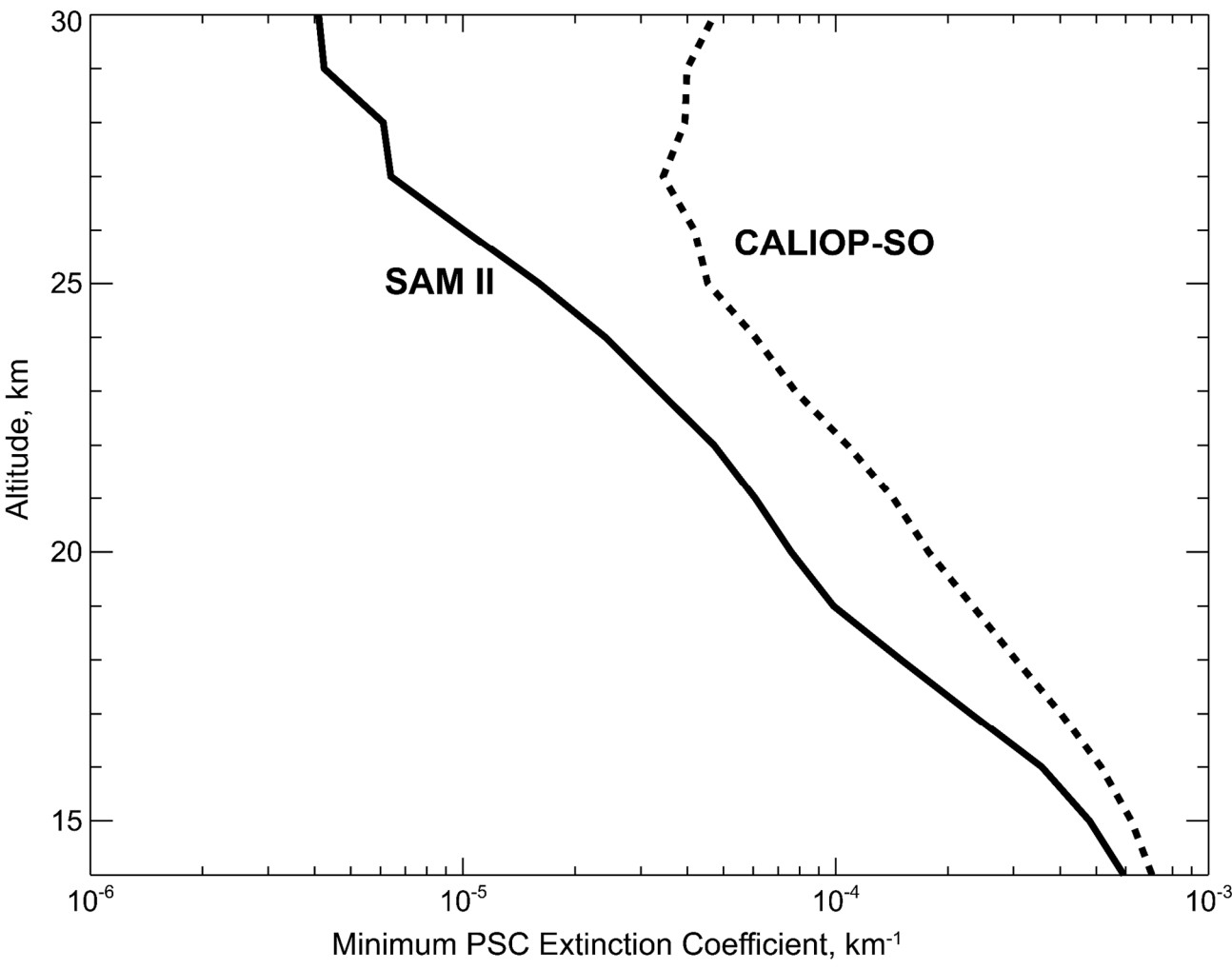

**Figure 31: Vertical profile of season-long minimum (1<sup>st</sup> percentile) values of adjusted (×1.3) 1-µm $\alpha_{particulate}$ for SAM II Antarctic PSCs (solid curve) and calculated 1-µm $\alpha_{particulate}$ for CALIOP-SO Antarctic PSCs (dotted curve).**

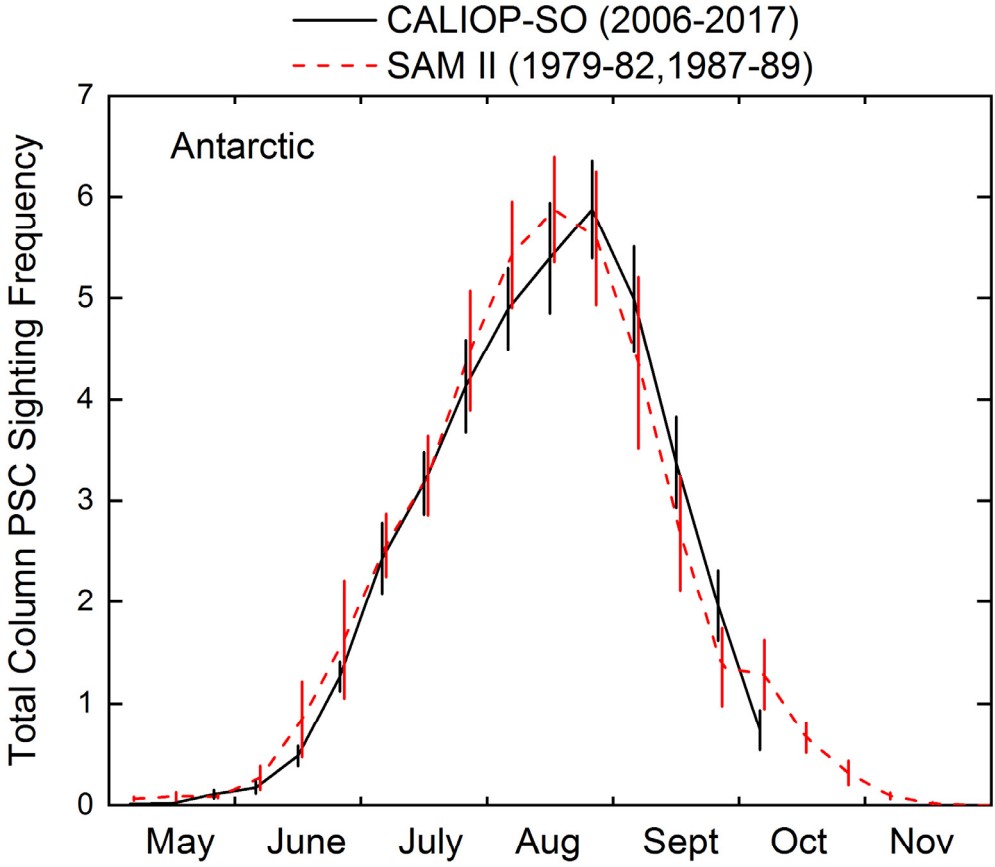

**Figure 32: Multi-year mean Antarctic PSC column occurrence frequency from May-November for CALIOP-SO (black solid) and SAM II (red dashed), along with standard errors in the means.**

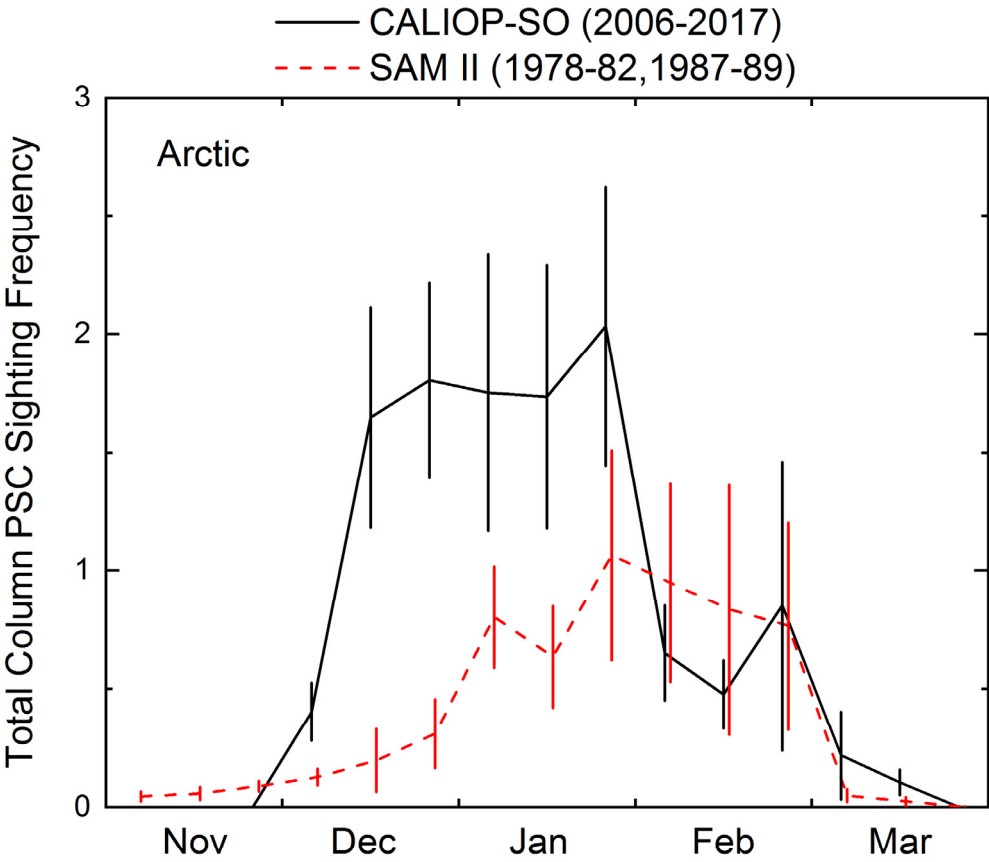

**Figure 33: Multi-year mean Arctic PSC column occurrence frequency from November-March for CALIOP-SO (black solid) and SAM II (red dashed), along with standard errors in the means.**

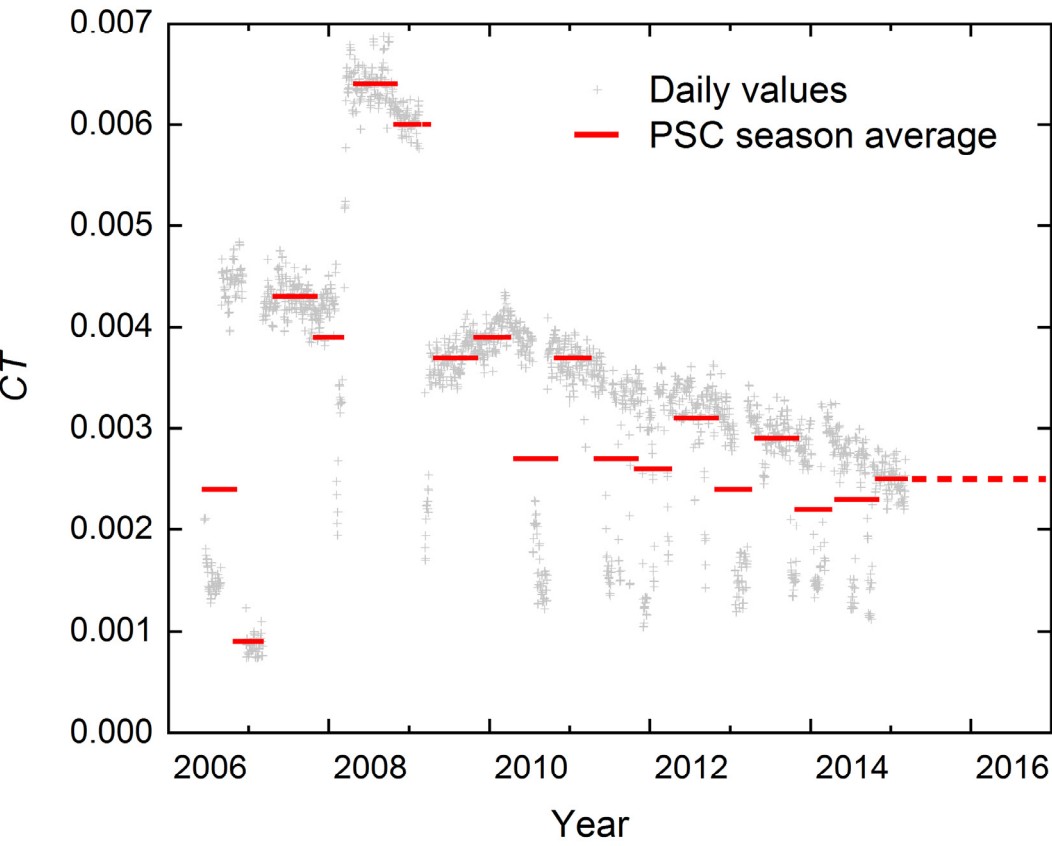

**Figure A1: Time series of *CT* calculated from daily values of δ$_{mol,meas}$ over the course of the CALIPSO mission, as well as the averages over PSC seasons.**