# Peer review of "Polar stratospheric cloud climatology based on CALIPSO spaceborne lidar measurements from 2006-2017"

_Atmospheric Chemistry and Physics, 2018_

## Referee Comment (RC1) · Anonymous Referee #1 · 23 Mar 2018

On page 5 the authosr state that

$$R_{532} = \frac{\beta^{\parallel}_{\text{par}} + \beta^{\perp}_{\text{par}}}{\beta_{\text{mol}}} \, . \tag{1}$$

However the general definition of the backscatter ratio is

$$R = \frac{\beta}{\beta_{\text{mol}}} = \frac{\beta_{\text{par}} + \beta_{\text{mol}}}{\beta_{\text{mol}}} \, , \tag{2}$$

If backscattered light is measured in separate channels for the detection of light that is polarized parallel and perpendicular with respect to the plane of polarization of the

emitted linearly polarized laser light, the definition of $\beta$ changes to

$$\beta_{\mathsf{T}} = \beta_{\mathsf{par}}^{\parallel} + \beta_{\mathsf{par}}^{\perp} + \beta_{\mathsf{mol}}^{\parallel} + \beta_{\mathsf{mol}}^{\perp}. \tag{3}$$

Now $\beta_{\mathsf{par}}^{\parallel}$ and $\beta_{\mathsf{par}}^{\perp}$ represent the co- and cross-polarized backscatter coefficient, respectively.

Combining eqs. (2) and (3) leads to the total backscatter ratio for polarization-sensitive measurements

$$R_{\mathsf{T}} = \frac{\beta_{\mathsf{par}}^{\parallel} + \beta_{\mathsf{par}}^{\perp} + \beta_{\mathsf{mol}}^{\parallel} + \beta_{\mathsf{mol}}^{\perp}}{\beta_{\mathsf{mol}}^{\parallel} + \beta_{\mathsf{mol}}^{\perp}}. \tag{4}$$

The backscatter ratio can also be calculated individually from the measurements in the polarized channels as

$$R^{\parallel} = \frac{\beta_{\mathsf{par}}^{\parallel} + \beta_{\mathsf{mol}}^{\parallel}}{\beta_{\mathsf{mol}}^{\parallel}} \tag{5}$$

and

$$R^{\perp} = \frac{\beta_{\mathsf{par}}^{\perp} + \beta_{\mathsf{mol}}^{\perp}}{\beta_{\mathsf{mol}}^{\perp}}. \tag{6}$$

It is not clear if this is an error in writing the manuscript (i.e. notation) or if the calculations have been performed with an incorrectly calculated backscatter ratio. Your use of the attenuated backscatter coefficient, which is defined as

$$\beta_{532}'^{\parallel} = \beta_{532}'^{\parallel} + T_{532}^2, \beta_{532}'^{\perp} = \beta_{532}'^{\perp} + T_{532}^2 \tag{7}$$

in the CALIPSO ATBDs cannot explain the difference.
* * *

---

## Referee Comment (RC2) · Anonymous Referee #1 · 29 Mar 2018

From the authors' reply I take that the attenuated parallel (perpendicular) backscatter coefficients as used in the paper actually denote the sum of the parallel (perpendicular) backscatter coefficients related to molecules and particles. This should be clearly stated in the manuscript. Generally, the term total attenuated backscatter coefficient is used to avoid misunderstanding.

Having said this, I would like to urge the authors to use less ambiguous names for their parameters and provide clear definitions/equations of how they are defined.

---

## Author Comment (AC1) · 29 Mar 2018

Anonymous Referee #1 seems to have misinterpreted our definition of attenuated scattering ratio, $R'_{532}$, as defined on page 5 (lines 30–31):

$$R'_{532} = \frac{(\beta'_{par} + \beta'_{perp})}{\beta_{mol}}, \tag{1}$$

where as stated on page 5 (lines 24–25), $\beta'_{par}$ is the attenuated parallel backscatter and $\beta'_{perp}$ is the attenuated perpendicular backscatter. We use the subscript "par" to represent the parallel component of backscatter, NOT the particulate component as

we think the Referee has interpreted. Therefore, our definition of $R'_{532}$ (Eq. 1 above) is equivalent to Referee #1's Eq. 2, with the exception of the prime symbols denoting attenuated backscatter quantities.
* * *

---

## Referee Comment (RC3) · Anonymous Referee #2 · 13 Apr 2018

The manuscript presents a climatology of PSC observations acquired by the spaceborne CALIOP lidar instrument over more than ten years. In this new version of the dataset, the authors have significantly improved their PSC detection and classification schemes, especially with respect to the detection threshold and PSC composition boundaries. They have included confidence indices on the composition and an estimation of the random uncertainties of the optical quantities. As new parameters, derived PSC surface and volume densities have been added to the dataset.

The multi-year climatology is analysed with respect to the thermodynamic equilibrium state of the different PSC compositions. Further, the spatial and temporal evolution

of PSCs in both hemispheres for different years has been described in detail. At the end, the CALIPSO observations are compared to satellite measurements which have been acquired during a ten-year period between 1979 and 1989 by the SAM II solar occultation instrument. It is concluded that within the limitations of this dataset no long-term changes can be observed.

This is a very important new dataset which will help to further progress our knowledge about polar stratospheric clouds and their involvement in stratospheric chemistry. The manuscript is well organized, clearly written and all Figures support the statements and conclusions drawn in the text. Only the enhanced classification of ice in the new dataset could be more clearly justified and the comparison with SAMII a bit expanded (see below). I strongly support publication of the manuscript after taking the specific comments below into consideration.

Specific comments:

P2, L20 'sedimentation of large NAT particles (Molleker et al., 2014)'

Please cite also the original paper(s) referring to the large NAT particles as reason for the denitrification.

P5, L16 'interpolated to the CALIOP PSC orbit grid using a weighted average of the two nearest MLS profiles'

How has this weighting been performed?

P5, L28 'The data are then corrected for molecular and ozone attenuation using the MERRA-2 molecular and ozone number density profiles'

Are the ozon profiles from MERRA-2 or those from the MLS retrievals? If MERRA, how strong does it affect the preprocessing if the 'real' (MLS) profiles would be used?

P6, L11:

Is the second equation really correct, or should it read: beta'_perp = beta'_perp,meas

– beta'_par x CT ?

P6, L17 '[unc_par] . . . [unc_perp]':

1. 'par' should everywhere be exchanged by 'para' not to confuse it with 'particles'

2. 'unc' is a strange variable. Could it not be called delta_beta', so that the direct connection with beta' becomes clear?

P7, L26 'is fixed at 10 cm−3':

Could you provide a reason for this value (e.g. a citation) and how large its variation could be?

P9, L21 'Points with CI_NS > 1 are presumed to be PSCs containing non-spherical particles.':

This seems to be a 2-sigma limit – could you describe more clearly what it means in percentage of the whole data points: how many of the 'non-spherical' particles might be spherical and vice-versa?

P9, L28 'detected through gas-phase uptake of HNO3 as observed by MLS'

Regarding your analysis of volume-density later in the manuscript: can you confirm, that the amount of HNO3 uptake seen by MLS is in accordance with the detection limit of CALIOP?

P12, L18 '8.3%'

In Fig. 10 the number seems 5.8% (?)

P12, L17..:

Apart from the referenced modelling work by Zhu et al., 2018, which additional arguments are there to support the strong increase in the detected ice-PSCs in v2 compared to v1?

[Figure]

P13, L17..:

Is the explanation for the bimodal distribution of NAT-mixtures also confirmed by any modelling work which could be referenced here?

P14, L22 'persisting until early October':

From your Fig. 12, PSC are also often seen until mid and even end of October.

P14, L30..:

Is there any possibility to distinguish the PSCs from upper tropospheric cirrus (e.g. in terms of volume/surface density? Do they 'separate' when plotting against potential temperature, if it is mainly caused by an upward displacement of isentropic surfaces, a separation might be possible.

P20, L10 'Then, we integrated the occurrence frequencies':

Does 'integrated' just mean 'summed up' ?

P20, L16: 'However, note that the SAM II occurrence frequencies are higher than those of CALIOP early in the PSC season.'

May this difference in the early PSC period also be caused by how the sightings in case of SAM II are counted? During this time, there is a clear maximum at around 18 km and a minimum below. As SAM II is a limb-sounder, the sightings below might be influenced by the higher PSCs through which the SAM II line-of-sight passes. Has this been taken into account in the comparison?

P20, L17: 'This may be a reflection of the greater sensitivity of the limb-viewing occultation measurements to the onset of PSCs when liquid droplets first began to deliquesce and/or when low number density NAT particles form that are below the CALIOP detection thresholds'

How far south are the soundings by SAM II in mid-May? Aren't the 'sub-visible' PSCs

identified by MLS more in the centre of the vortex where SAM II has not been observing?

P22, L24 'The estimates assume liquid particles (binary H2SO4-H2O or STS) only and thus have large uncertainties when NAT mixtures or ice are present'

It would be helpful for the reader to repeat at this point that in case of NAT or ice the values are very probably lower limits of surface and volume density.

P33, Fig. 3:

To better understand the differences between the new and the old composition classification, either the separation lines of v2 should be included in Fig. 3 or those of v1 in Fig. 4.

P33, Fig. 3, caption 'and symbol sizes are proportional to NAT'

Does 'size' mean the symbol area or the diameter?

P44, Fig. 14:

The colours are partly difficult to distinguish, could different symbols be used, at least for the case of similar colours?

P54, Fig. 24:

Please plot the results from both instruments in one graph. As it is now, the comparison in relation to the text is difficult.

Technical:

P2, L5: 'of season' -> 'of the season'

P6, L9,11: numbering of equations missing

P14, L20: 'Hence, it not' -> 'Hence, it is not'

P16, L17: 'from the near' -> 'from near'

P17, L24: 'about the pole' -> 'around the pole'

P20, L6: delete blank between 'degree' and 'N' or 'S'

P21, L26: 'that more are more' -> 'that are more'

P32, Fig. 2 caption: 'in yellow': in Fig. 1 this appears more light green than yellow

P41, Fig. 11 caption: '1 Antarctic' -> '12 Antarctic' (?)

P43 and P49: color bar legend: '(x 10ˆ6 kmˆ2)' should read either '(10ˆ6 kmˆ2)' or '/(10ˆ6 kmˆ2)'

---

## Referee Comment (RC4) · Anonymous Referee #3 · 24 Apr 2018

**General remarks:**

This is an excellent overview paper on unique measurements of polar stratospheric clouds (PSCs) and their composition of now more than 11 years by the CALIOP lidar on-board the CALIPSO satellite. The paper discusses new improvements in the classification approach in details, introduces a new retrieval approach of particle surface and volume density information, and gives a detailed introduction in the corresponding climatology of PSC occurrence and composition. The paper is well organised and written, and the scientific objectives fit perfectly to the scope of ACP. I strongly recommend the publication of the manuscript after some minor corrections and improvements.

[Figure]

**Minor comments:**

To my mind the manuscript is missing a short section/paragraph on comparisons with PSC measurements of other sensors. Some references on comparisons are given at various places of the paper, but I recommend to summarise the comparisons results at one specific place of the manuscript. This would better highlight the quality and reliability of the PSC detection and classification methods of the CALIOP instrument.

I would recommend to move 3.1.1. and 3.1.2 into an appendix. A reduction on technical details in section 3 would be desirable for none-expert readers.

Page 9, line 5: I am wondering that the MIPAS observations show a NAT belt on 2008-05-29 and 2008/06-01/02 but no indication on May 30. Usually the NAT belt is devolving slowly over a couple of days starting with a small area of NAT/ice activity followed by a downstream formation of a belt-like structure in the next days (e.g. Höpfner et al., 2006). Please clarify, if May 30 is really NAT-free (maybe a typo?) in the MIPAS observations. Is it possible that MIPAS just misses the small NAT area from the day before due to a sampling issue? This potential mismatch may bias your definition of the empirical sub-class of 'enhanced NAT mixtures'.

Section 3.6: The percentages of Figure 10 suggests that mainly enhanced NAT mixtures of Version 1 are classified in Version 2 as ice. Is this correct, or is the effect caused by misclassification of the former Mix-2 and Mix-2enhanced classes. Can you quantify the partitioning between the two V1 classes into the V2 ice class?

Section 3.7: The temperature difference between STS and ice in the $T - T_{ice}$ histogram for the maximum position ($\Delta T$ 1-1.5K) looks unexpectedly small to me. I would expect from the equilibrium curves, for example presented in Fig. 5 of Pitts et al. (2013), higher temperatures for STS. Can you please clarify and/or explain in more detail how you defined $T_{STS}$ based on Carslaw et al. (1995).

Page 13, line 8: Is the 'strong' statement regarding the positive tail in the PSC distribution (that this is due to warm biased temperatures associated with wave ice events not fully resolved in MERRA-2 fields) based on a detailed analysis or 'only' one plausible explanation. Uncertainties of the threshold lines between Ice and NAT may cause a similar tail in the distribution. Please commend and clarify.

The authors may think about to skip Figure 16, which is partly redundant to Fig. 17. For example, Fig. 17 includes by far more quantitative information than Fig. 16 due to the choice of the vertical and horizontal coordinates.

Section 6: To my mind the SAM II - CALIPSO comparison would profit by some more detailed descriptions and analyses. The information on SAM II measurements are very limited. For a profound comparison of the PSC occurrence frequencies it would be necessary to discuss the detection limits of both instruments (I guess based on extinction or volume density thresholds). The authors should discuss similarities and differences between the two sensors as well.

**Technical corrections:**

page 14, line 20: 'Hence, it is not ...'

p21, L7: please explain 'DMPs'

p23, L26: For completeness the authors may like/need to add a reference to the SAM II dataset as well.

page 31/32; Fig. 1/2 caption: CALIOP curtain of Fig. 2 looks on my printout and screen greenish and not yellow. Please check.

Figure 3: 'The symbols size are proportional to volume-equivalent radii of NAT and ice'. This fact is hard to see in Figure 3 and may cause the effect, that the Mix1 calculations are hiding all STS results. Is the particle size an important topic for this figure? If not, keeping the figure more simple the interpretation of the figure might be easier for the reader. Is the the particle radius also an issue in Figure 4? If yes, this is not obvious from the caption and the text passages in the corresponding section.

Figure 4: The authors may explain the grey box in the figure caption (S/N issue) or reference to the details in the corresponding section.

Figure 18: Starting at 5% occ. freq. with the colour bar looks a bit extreme. Especially, if this is leading to the strong statement of section 4.1.2 'with essentially no STS' occurrence in the the deep vortex during September. Please clarify, if this statement is an 'artefact'.

**References:**

Please abbreviate the First Name of the authors (copernicus style) for the references of Wegner et al., Young et al. and Prata et al. .

———————————————

---

## Referee Comment (RC5) · Anonymous Referee #1 · 30 Apr 2018

The authors present a comprehensive analysis of 11 years of PSC observation based on spaceborne CALIPSO lidar measurements. The authors introduce and apply the new version 2 algorithm (v2) and compare its findings to those of the previous v1 used in earlier studies. The paper is important for the community and should be published in ACP after the fowling comments have been addressed.

Major comments:

- The paper lacks a comparison/validation of the CALIPSO-derived PSC occurrence statistics to long-term PSC statistic from ground-based measurements in the Antarctic and Arctic.

[Figure]

- The authors compare their v1 and v2 composition schemes only for theoretical calculations. The paper would benefit from an example case for both the v1 and v2 classification, i.e. plots such as Figures 3 and 4 with real data, for instance the PSC observation presented in Figure 2.

- Page 11: There is an abundance of balloon-borne in-situ measurements of particle size distributions in PSCs, e.g. Schreiner et al. (2003), Voigt et al. (2003), Deshler et al. (2003). A discussion of these measurements should be provided. Also, why have those data not been considered in the estimation of SAD and VD?

- Where does the information on the height of the tropopause used in the retrieval come from? How reliable are the values? It seems excessive to dismiss any data within 4 km of the tropopause height.

- Figure 15c shows the highest occurrence rate of ice PSCs below 15 km height. I would expect to see those higher up, particularly as Figure 16 shows that the frost point temperature is only reached above 15 km height. Can cirrus still be misclassified as ice PSC?

- Please add a figure on the occurrence rate of different PSC types with altitude for the Arctic.

Minor comments:

- Please be precise in the naming of your parameter to avoid misunderstandings as in my initial comment.

- What is the explanation for the change of the crosstalk value in the 2008 Antarctic and 2008-09 Arctic winters, and then again in 2015?

- Please provide the equation used to calculate the particle linear depolarisation ratio.

- What is a typical total backscatter ratio for the stratospheric background aerosol layer from CALIPSO? Is it comparable to ground-based measurements?

- Page 8, line 7: Spheroids with aspect ratios smaller than 1 should be oblates. Why not use a mixture of prolates and oblates as done by Reichardt et al. (2002, 2014)?

- Page 8, line 8: What is the maximum value of the particle linear depolarisation ratio and how often have they been observed? I would like to know that these values are not outliers.

- Page 10, line 21: Note that there is literature of lidar ratios in PSCs from ground-based observations, e.g. Reichardt et al. (2004).

- Page 13, line 24: Are you referring to millions of PSC profiles or indeed millions of PSC observations (i.e. individual clouds)? Please note that your profiles in the same don't provide independent measurements and that a single cloud might be present over several orbits. This means that your statistics are biased towards oversampled clouds. Please comment.

- Page 19, line 7: Again, are you referring to profiles or individual clouds? Please carefully revise your use of numbers of observations/profiles.

- Page 20: Why has the comparison to SAM II not been performed for Arctic PSCs?

- Figure 2: There is no gap between tropospheric and stratospheric clouds. Are they separated, e.g. through the feature classification or the height of the tropopause? If it's the height of the tropopause, please add the respective data to the figure. If it's through the feature mask, please add a subplot with the feature mask.

- Figure 9: please provide information on the height of the tropopause.

- Figure 13: please remove all data points below the tropopause

- Figure 19: This figure reveals the same effect as shown in Figure 12 and described on page 14, line 27, though not for all years. This should be discussed. Note that it has been reported previously by Fromm et al. (2003) and Achtert et al. (2012).

- Figure 22: I would have expected a larger fraction of wave ice in the Arctic due to the

stronger wave activity, e.g. triggered by Greenland and the Scandinavian mountains. With generally lower temperatures in the Arctic, does the threshold for wave ice require adjustment?

Achtert, P. et al., 2012. On the linkage between tropospheric and Polar Stratospheric clouds in the Arctic as observed by space–borne lidar. Atmospheric Chemistry and Physics, 12(8), pp.3791–3798. Available at: http://www.atmos-chem-phys.net/12/3791/2012/

Deshler, T., 2003. Large nitric acid particles at the top of an Arctic stratospheric cloud. Journal of Geophysical Research, 108(D16), p.4517. Available at: http://doi.wiley.com/10.1029/2003JD003479

Fromm, M., 2003. A unified, long-term, high-latitude stratospheric aerosol and cloud database using SAM II, SAGE II, and POAM II/III data: Algorithm description, database definition, and climatology. Journal of Geophysical Research, 108(D12), p.4366. Available at: http://doi.wiley.com/10.1029/2002JD002772

Reichardt, J., 2002. Retrieval of polar stratospheric cloud microphysical properties from lidar measurements: Dependence on particle shape assumptions. Journal of Geophysical Research, 107(D20), p.8282. Available at: http://doi.wiley.com/10.1029/2001JD001021

Reichardt, J. & Reichardt, S., 2004. Mountain wave PSC dynamics and microphysics from ground-based lidar measurements and meteorological modeling. Atmos Chemistry and Physics, (January 1997), pp.1149–1165. Available at: http://www.atmos-chem-phys.net/4/1149/2004/acp-4-1149-2004.pdf

Reichardt, J. et al., 2014. Mother-of-Pearl cloud particle size and composition from aircraft-based photography of coloration and lidar measurements. Applied Optics, (October).

Schreiner, J. et al., 2002. Chemical, microphysical, and optical properties of polar

stratospheric clouds. Journal of Geophysical Research, 108(D5), p.8313. Available at: http://doi.wiley.com/10.1029/2001JD000825

Voigt, C., 2003. In situ mountain-wave polar stratospheric cloud measurements: Implications for nitric acid trihydrate formation. Journal of Geophysical Research, 108(D5), p.8331. Available at: http://doi.wiley.com/10.1029/2001JD001185

---

## Referee Comment (RC6) · Anonymous Referee #4 · 7 May 2018

Summary

Pitts et al. develop a new version of their PSC detection algorithm (called V2) which is a comprehensive advance on their PSC algorithms developed in a series of papers by Pitts et al. stretching back to 2007. Importantly, this new V2 algorithm incorporates the effects of denitrification and dehydration (especially important in the Antarctic) during late winter with the use of near-coincident MLS gas phase data as well as incorporating the effects of uncertainties. Pitts et al. also refine the boundaries separating the different PSC classes. It is pleasing to note a much greater spatial consistency of the different classes (e.g. their Figure 9c) compared with V1 (Figure 9d) where

more speckled nature is observed. Using this new PSC detection algorithm, Pitts et al. provide the Arctic and Antarctic climatologies of the PSC classes from 2006 till 2017. Finally, comparisons are made with historical SAM occultation data.

This is a well organised and well written paper and represents an important advance in the field. Consequently it should be published in ACP subject to the following minor revisions.

Minor Comments

1) In the text in Section 3.6 regarding Figure 9, some comment on the fact that v2 only detects ice in the lowest altitude region in the cloud at 26E is also warranted – i.e. ice in the upper troposphere merges seamlessly into ice in the lower stratosphere, rather than having low-level mixed NAT PSCs in (assumedly) the upper troposphere as in v1.

2) p13, line 9. It would be worth providing a citation for this comment regarding the tail of ice PSC distributions being due to wave ice events.

3) Section 5 p 19 line 24. There is large variability in climatological SAD in Figure 23 over ∼ fortnightly intervals in July or August. Does this follow directly from the variability of the climatology of PSC ice at this time (Fig 15c, i.e. a large fraction of total PSC area is ice at the start of August)? Please clarify.

4) Do other Antarctic years show similar changes between v1 and v2, as shown in Figure 10 for 2009?

5) Figure 15c. There is a white contour enclosed in the ice PSC in May – June but your colorbar indicates purple as 0.0. Please reconcile or explain in the figure caption.

---

## Author Response (AR1)

**Response to Anonymous Referee #1**

Included below are our responses to the comments from Anonymous Referee #1 on our ACPD paper. The specific referee comments are given in bold italics and our responses in plain text. We thank the referee for their insightful comments which have resulted in a much improved manuscript.

*Major Comments:*

***-The paper lacks a comparison/validation of the CALIPSO-derived PSC occurrence statistics to long-term PSC statistic from ground-based measurements in the Antarctic and Arctic.***

We agree that it is important to link the CALIPSO satellite-derived PSC statistics to the long-term ground-based measurements. To that end, a detailed comparison of observations of PSC occurrence and composition from CALIOP and the ground-based lidar at McMurdo is the focus of a separate paper that has just been submitted for review in ACPD (Snels et al., acp-2018-589). The paper is collaboration between the CALIOP team and colleagues at the Institute of Atmospheric Sciences and Climate (CNR-ISAC) in Italy. In general, point-to-point comparison between ground-based and satellite based lidar measurements is extremely difficult due to the intrinsic differences in the observation geometries and the imperfect overlap of the observed areas. Therefore this study employs a statistical approach to compare the satellite- and ground-based observations from a 5-year period (2006-2010). The ground-based McMurdo lidar data have been processed using a detection and classification algorithm which closely follows the CALIOP v2 algorithm in order to avoid a bias due to different classification schemes. The relative occurrences of the four PSC compositions, STS, NAT mixtures, enhanced NAT mixtures and ice, averaged over the five year overlap period are very similar for ground-based and CALIOP observations, and also the vertical distribution compares well.

We have also compared CALIOP PSC occurrence frequencies for 2006-2017 with published PSC climatologies for McMurdo and Ny-Ålesund ground-based lidars (Di Liberto et al., 2014; Massoli et al., 2006). A summary of these comparisons is now included in Section 6 of the revised manuscript. For these comparisons, we took the subset of CALIOP observations within $\pm 1.5°$ latitude and $\pm 15°$ longitude of McMurdo and Ny-Ålesund. The CALIOP occurrence frequency distribution for the McMurdo coincidences is very similar to the ground-based McMurdo distributions for both the 2006-2010 period (Figure 3 in Di Liberto et al., 2014) and 1995-2001 period (Figure 3 in Massoli et al., 2006), illustrating both the consistency between the satellite and ground-based lidar datasets and the regularity of the Antarctic PSC seasons. The CALIOP distribution for the Ny-Ålesund coincidences is qualitatively similar to the ground-based Ny-Ålesund distribution for the 1995/96-2002/03 period (Figure 3 in Massoli et al., 2006), but the lack of agreement in absolute sighting frequencies is not surprising given the large interannual variability in Arctic PSCs and the lack of temporal overlap between the satellite and ground-based datasets.

*-The authors compare their v1 and v2 composition schemes only for theoretical calculations. The paper would benefit from an example case for both the v1 and v2 classification, i.e. plots such as Figures 3 and 4 with real data, for instance the PSC observation presented in Figure 2.*

We are very appreciative of this suggestion by the referee and have added 2-D histogram figures (new Figures 5 and 6) showing the distribution of CALIOP PSC observations in the v1 and v2 classification optical spaces. The observations are from the period 10-18 July 2008, which includes the orbit shown in Fig. 2 and renumbered Fig. 11. The choice of this time period clearly illustrates that by accounting for denitrification and dehydration, the v2 PSC classification scheme captures the separation between ice and NAT mixture PSCs much better than the v1 scheme.

*-Page 11: There is an abundance of balloon-borne in-situ measurements of particle size distributions in PSCs, e.g. Schreiner et al. (2003), Voigt et al. (2003), Deshler et al. (2003). A discussion of these measurements should be provided. Also, why have those data not been considered in the estimation of SAD and VD?*

The in situ PSC particle data in Schreiner et al. (2003), Voigt et al. (2003), and Deshler et al. (2003) all appear to have been obtained under mountain wave conditions, so it is not clear that they are representative of PSCs as a whole. However, we did calculate $\beta_{particulate}$, SAD, and VD based on bimodal lognormal size distribution fits to in situ optical particle counter measurements within three PSC layers presented in Deshler et al. (2003). These calculated points are now included in renumbered Figs. 9 and 10 and show that our estimates of SAD and VD are reasonable. We also have been provided a draft of a manuscript presenting a comprehensive analysis of coincident in situ PSC particle size distributions and ground-based lidar measurements from McMurdo. Empirical relationships between $\beta_{particulate}$ and SAD and VD derived from those data are quite similar to the theoretical relationships we use and give us more confidence in our estimates of SAD and VD.

*- Where does the information on the height of the tropopause used in the retrieval come from? How reliable are the values? It seems excessive to dismiss any data within 4 km of the tropopause height.*

Tropopause height information is based on the MERRA-2 "blended" tropopause values that are included in the CALIOP Level 1b lidar data product files. The MERRA-2 "blended" tropopause is the lower (in altitude) of the temperature-based ("thermal") tropopause and potential vorticity (PV) based ("dynamic") tropopause (Bosilovich et al., 2016; Ott et al., 2016). In practice, the MERRA-2 "blended" tropopause is usually the "dynamic" tropopause in mid- and high-latitudes, but switches to the "thermal" tropopause in the tropics. The tropopause is often difficult to locate in the polar regions, especially during the polar nights (e.g., Highwood et al., 2000), so any tropopause height determination in these regions should be used with caution. It is our experience that the transition from upper tropospheric cirrus to PSCs is often ambiguous with no clear separation across the reported tropopause. Therefore, using the MERRA-2 tropopause as a hard boundary to separate tropospheric cirrus from PSCs will lead to misclassifications. Hence, we make no explicit attempt to distinguish tropospheric cloud from stratospheric cloud in the 8.5-30 km altitude range over which we produce the CALIOP PSC cloud mask. However, we do tag each

observation in the database with a feature flag that identifies its altitude location relative to the reported "blended" tropopause as one of three possibilities: (1) below the tropopause, (2) between the tropopause and tropopause + 4 km, or (3) above the tropopause + 4 km. This allows data users to perform some crude separation between troposphere and stratosphere as desired. Only in the PSC spatial volume analyses (renumbered Figs. 16 and 23) and composition pie charts (renumbered Figs. 12 and 26) do we exclude data within 4 km of the tropopause because inclusion of the omnipresent cirrus would distort the statistics on the temporal evolution of PSC occurrence and relative fraction of ice PSCs. We have improved the v2 composition classification algorithm in the lower stratosphere/upper troposphere by incorporating observed Aura MLS $H_2O$ and $HNO_3$ abundances in the determination of the optical space composition boundaries at these lower altitudes. However, cirrus clouds will be identified simply as 'ice' clouds in our algorithm. We now include the location of the MERRA-2 tropopause in many of the figures as a guide to the reader for the approximate upper extent of cirrus. We have also added a discussion in the revised manuscript of the difficulty in using the tropopause to separate cirrus from PSCs.

*- Figure 15c shows the highest occurrence rate of ice PSCs below 15 km height. I would expect to see those higher up, particularly as Figure 16 shows that the frost point temperature is only reached above 15 km height. Can cirrus still be misclassified as ice PSC?*

As discussed above, we do not explicitly separate tropospheric ice clouds from stratospheric ice clouds due to the uncertainty in the location of the tropopause. To put better focus on PSCs, we limit the lower altitude in (renumbered) Fig. 17 to 12 km, near the climatological maximum MERRA-2 tropopause as shown in (renumbered) Fig. 15. The higher occurrence rates of ice clouds below 15 km are indeed likely due cirrus clouds. Using the MERRA-2 "blended" tropopause height as the discriminator does not completely filter out cirrus and enhanced ice occurrence rates are still present below 15 km. However, if we use the tropopause height + 4 km as the discriminator, the high ice frequencies associated with cirrus below 15 km completely disappear. We added some sentences in the description of (renumbered) Fig. 17c to mention that the ice cloud statistics do include some cirrus in the lower altitudes.

*- Please add a figure on the occurrence rate of different PSC types with altitude for the Arctic.*

Due to the high degree of interannual variability in PSC occurrence in the Arctic, the 11-year mean depictions of PSC occurrence and composition do not accurately represent the seasonal evolution in any given year and, in fact, may provide a misleading picture of PSC occurrence. Therefore we avoided including such representations in our original manuscript. However, based on referee suggestions, we now include these in the revised manuscript with appropriate caveats. In terms of the climatology of Arctic PSC composition, we feel that it is meaningful to show only the composite season-long vertical profile of relative spatial coverage (composition-specific area normalized by total PSC area). This is now included as new Figure 24. STS and NAT mixtures are the major Arctic PSC compositions as expected, with STS (NAT mixtures) predominant above (below) 24 km. Ice becomes the predominant composition at altitudes below 14 km, but this likely reflects an imprecise separation of ice PSCs from upper tropospheric cirrus due to uncertainty in the tropopause height.

*Minor comments:*

*- Please be precise in the naming of your parameter to avoid misunderstandings as in my initial comment.*

We have attempted to be more precise in our parameter notation by explicitly using perpendicular and parallel symbols. To avoid confusion, we use the subscript "∥" to denote parallel, the subscript "⊥" to denote perpendicular, and the subscript "particulate" to denote particulate in the revised manuscript.

*- What is the explanation for the change of the crosstalk value in the 2008 Antarctic and 2008-09 Arctic winters, and then again in 2015?*

In response to a suggestion by Anonymous Referee #3, we have moved the discussion of crosstalk (CT) to Appendix A. We have added a figure to the Appendix showing the record of CT as derived from measurements of molecular depolarization ratio and have added text as to the likely cause of the increase in CT in the 2008 Antarctic and 2008-09 Arctic winters. There was no change in CT in 2015. There have been no regular measurements of molecular depolarization since March 2015, so we assumed a constant value of CT since that point, as stated in the Appendix.

*- Please provide the equation used to calculate the particle linear depolarisation ratio.*

Done. This is Equation 2 in the revised manuscript.

*What is a typical total backscatter ratio for the stratospheric background aerosol layer from CALIPSO? Is it comparable to ground-based measurements?*

During the lifetime of CALIPSO, the total backscatter ratio for the stratospheric aerosol layer in the polar regions has ranged from around 1.04 to 1.1, depending on the aerosol loading in a particular year. Vernier et al. (2009) found excellent agreement between CALIOP stratospheric aerosol scattering ratios and those from ground-based lidar, however these comparisons were primarily confined to lower latitudes. We have compared the CALIOP total backscatter ratio values used in the PSC algorithm to represent background stratospheric aerosol levels with profiles of total backscatter ratio from ground-based lidars at Dumont d'Urville and McMurdo in the Antarctic and Eureka in the Arctic. These comparisons indicate CALIOP and the ground-based lidars generally agree to within 5-10%.

*- Page 8, line 7: Spheroids with aspect ratios smaller than 1 should be oblates. Why not use a mixture of prolates and oblates as done by Reichardt et al. (2002, 2014)?*

We followed the terminology used by Engel et al. (2013) and Mishchenko and Travis (1998), where the aspect ratio was defined as the ratio of the horizontal to rotational axes lengths, i.e. prolate spheroids have aspect ratios < 1. We did show calculations for oblate spheroids with an aspect ratio of 1.2 in Pitts et al. (2009), which are similar to the results shown in the

present paper, but with smaller theoretical maximum values of $\delta_{particulate}$. $\delta_{particulate}$ is not used for composition classification in our v2 algorithm, so we feel there is no need to repeat the theoretical calculations with a mixture of prolates and oblates.

*- Page 8, line 8: What is the maximum value of the particle linear depolarisation ratio and how often have they been observed? I would like to know that these values are not outliers.*

Engel et al. (2013) were referring to best agreement with maximum $\delta_{particulate}$ values of 0.3-0.4 for NAT mixtures and 0.4-0.6 for ice. The 2-D histogram in Figure 5 shows that there are many CALIOP observations in these regions of the optical space, i.e. that they are not outliers.

*.- Page 10, line 21: Note that there is literature of lidar ratios in PSCs from ground-based observations, e.g. Reichardt et al. (2004).*

Our theoretical lidar ratios are consistent with layer-averaged values for $S_{particulate}$ at 355 nm shown in Reichardt et al. (2004), e.g. values from 67-82 sr for PSCs with small scattering ratios and values of 20-35 sr for PSCs with large scattering ratios. We have added text accordingly to the revised manuscript.

*- Page 13, line 24: Are you referring to millions of PSC profiles or indeed millions of PSC observations (i.e. individual clouds)? Please note that your profiles in the same don't provide independent measurements and that a single cloud might be present over several orbits. This means that your statistics are biased towards oversampled clouds. Please comment.*

Here an 'observation' refers to a single CALIOP 5-km horizontal x 180-m vertical resolution measurement sample along a CALIOP orbit track. The spatial extent of PSCs as seen in the CALIOP orbit curtains can be hundreds to even thousands of kilometers horizontally by many kilometers thick. So an individual observation obviously doesn't represent a single independent cloud measurement. However, the distance over which PSCs have reasonably homogeneous physical properties as indicated by the variability in optical parameters is much smaller than the distance between opposite clear air boundaries. For instance, Kent et al. (1997) found a mean horizontal and vertical scale of homogeneity within cirrus clouds to be 20-25 km and 0.6-0.8 km, respectively. Although the scales of homogeneity may be different for PSCs, we are confident that they are much smaller than the distance between clear air boundaries. Although CALIOP may sample the same 'cloud' in adjacent profiles or orbits, each observation represents a different measurement sample volume with potentially different particle ensembles. So yes there likely is correlation between adjoining observations and if we were attempting to count clouds, overlap would be problematic. However, in the context of the discussion on page 13, we list the total number of 'observations' to highlight the vast quantity of data acquired over nearly the whole polar region over 11+ years that goes into these analyses. In deference to the referee, we have reworded the sentence to read: "These composite histograms, which incorporate over 11 years of CALIOP PSC measurements, demonstrate behavior consistent with theoretical expectations for each composition class, providing confidence that the v2 composition classification scheme is robust.

*- Page 19, line 7: Again, are you referring to profiles or individual clouds? Please carefully revise your use of numbers of observations/profiles.*

See discussion above. We have replaced "observations" with "measurement samples" to avoid implications that individual clouds are being observed with each measurement.

*- Page 20: Why has the comparison to SAM II not been performed for Arctic PSCs?*

The SAM II/CALIOP analyses are an attempt to assess the possibility of a long-term trend in PSC occurrence. The extremely high variability of Arctic PSC occurrence makes it very difficult to detect any long-term trend in the Arctic and thus we omitted this analysis from the original manuscript. We now include the Arctic results in Section 6 in the revised manuscript. Overall, the CALIOP occultation and SAM II records of Arctic column PSC occurrence frequency are very different, especially in December and January. However, we feel that this is not a long-term trend, but simply a result of comparing measurements from different time periods in a region with very high interannual variability.

*- Figure 2: There is no gap between tropospheric and stratospheric clouds. Are they separated, e.g. through the feature classification or the height of the tropopause? If it's the height of the tropopause, please add the respective data to the figure. If it's through the feature mask, please add a subplot with the feature mask.*

Figure 2 shows a standard browse image of the CALIOP Level 1b total attenuated backscatter along an orbit similar to what is provided on the CALIPSO web site (https://www-calipso.larc.nasa.gov/products/lidar/browse_images/production/). Only Level 1 processing has been performed at this point (geolocation and conversion to sensor units, e.g. total attenuated backscatter coefficient). Feature detection and classification is part of the Level 2 processing and the corresponding feature mask for this orbit is shown in (renumbered) Figure 11. Figure 2 is shown to simply illustrate the high resolution sampling capability of CALIOP in the polar troposphere and stratosphere. For context, we have added the MERRA-2 "blended" tropopause altitude to Figure 2.

*Figure 9: please provide information on the height of the tropopause.*

We have added the MERRA-2 'blended' tropopause altitude to the figure.

*- Figure 13: please remove all data points below the tropopause.*

Given the uncertainty in the reported tropopause location (see discussion above), using the MERRA-2 tropopause to filter out tropospheric cloud produces less than pleasing results with significant amounts of cirrus remaining. To completely eliminate cirrus contamination, we must filter all clouds within several kilometers of the tropopause, which filters some PSCs as well. To avoid this conundrum, we feel it is best to not attempt to filter out upper tropospheric cirrus from the climatology, except in cases where cirrus contamination explicitly skews the statistics. In the context of the climatology, the temporal and spatial extent of the underlying cirrus provides additional information on cloud occurrence in the polar upper troposphere/lower

stratosphere that may be useful to potential users of the database. As mentioned previously, we do include a feature flag in the database that tags each PSC measurement sample with its location relative to the MERRA-2 tropopause. For added context, we have included the climatological daily maximum MERRA-2 polar tropopause height to the figure.

*- Figure 19: This figure reveals the same effect as shown in Figure 12 and described on page 14, line 27, though not for all years. This should be discussed. Note that it has been reported previously by Fromm et al. (2003) and Achtert et al. (2012).*

Thank you for pointing this out. Yes, although not as common as in the Antarctic, upper tropospheric forcing events also produce deep synoptic-scale cloud layers in the Arctic that can extend from the troposphere into the stratosphere, as previously noted by Fromm et al. (2003) and Achtert et al. (2012). There are several years in (renumbered) Figure 21 that indicate this phenomenon. We have added a brief discussion of this to the text.

*- Figure 22: I would have expected a larger fraction of wave ice in the Arctic due to the stronger wave activity, e.g. triggered by Greenland and the Scandinavian mountains. With generally lower temperatures in the Arctic, does the threshold for wave ice require adjustment?*

Pitts et al. (2011) found that intense mountain-wave induced PSCs can be distinguished as a subset of CALIOP ice PSCs through their distinct optical signature in $R_{532}$ and lidar color ratio (the ratio of 1064-nm to 532-nm aerosol backscatter coefficients). In general, lidar color ratio is an indicator of particle size; cirrus and tropospheric clouds have color ratios around 1, indicating large particles, while smaller aerosol particles have lower color ratios (Liu et al. 2004). We do not use the color ratio in an absolute sense in our PSC algorithm because of calibration issues and the generally low signal-to-noise (SNR) of 1064-nm CALIOP data in the stratosphere (Hunt et al., 2009). However, an abrupt change in color ratio occurs for ice PSCs with very high values of $R_{532}$ (where the 1064-nm data are more reliable), which we feel is a clear signature of mountain wave PSCs. However, it is important to note that the CALIOP wave-ice PSC class is not all-inclusive; some additional CALIOP ice PSC observations are likely associated with mountain waves, but do not meet our strict ($R_{532} > 50$) wave-ice identification criterion, e.g. observations immediately upwind or downwind of the location of peak backscatter. We have added this caveat to the revised manuscript.

At first glance, the fraction of wave ice in the Arctic seemed small to us as well. However, published studies support the CALIOP statistics. For instance, Alexander et al. (2013) quantified the role of orographic waves on PSC occurrence for four Arctic (2006-07 to 2009-10) and four Antarctic (2007-2010) winter season and found that 37% of Antarctic days and 12% of Arctic days were orographic-wave active. Also, Dörnbrack and Leutbecher (2001) compiled a 20-year climatology of meteorological conditions necessary to produce mountain wave PSCs over Scandinavia and concluded that mountain wave PSCs occurred only about 3 days/month on average (~10%) with a maximum in January. In comparison, CALIOP detected wave ice over Scandinavia on about 6 days/month during January 2006-2017, which is not inconsistent with the published

findings. In terms of relative occurrence, CALIOP wave ice comprises about 2.5% of all ice PSC measurement samples from the Arctic, but only about 0.4% of all ice PSC measurement samples from the Antarctic. So in this sense, wave ice is a larger fraction of total ice PSC occurrence in the Arctic than in the Antarctic.

The referee brings up an interesting point regarding the wave ice threshold in the Arctic compared with the Antarctic. The analysis in Pitts et al. (2011) was based on four years of combined data from both the Arctic and Antarctic with no attempt to separate hemispheres. So to adequately address the referee's point, more detailed analyses separated by hemisphere would be required. This is beyond the scope of the current paper, but will be considered in future work.

Bosilovich, M. G., R. Lucchesi, and M. Suarez, 2016: MERRA-2: File Specification. GMAO Office Note No. 9 (Version 1.1), 73 pp, available from http://gmao.gsfc.nasa.gov/pubs/office_notes.

Ott, L. E., Duncan, B. N., Thompson, A. M., Diskin, G., Fasnacht, Z., Langford, A. O., Lin, M., Molod, A. M., Nielsen, J. E., Pusede, S. E., Wargan, K., Weinheimer, A. J., Yoshida, Y., Frequency and impact of summertime stratospheric intrusions over Maryland during DISCOVER-AQ (2011): New evidence from NASA's GEOS-5 simulations, J. Geophys. Res., Atmos., 121, 3687-3706, doi:10.1002/2015JD024052, 2016.

Highwood, E. J., Hoskins, B. J., Berrisford, P., Properties of the arctic tropopause, Q. J. Roy. Met. Soc., 126, 1515-1532, https://doi.org/10.1002/qj.49712656515, 2000.

Alexander, S. P., Klekociuk, A. R., McDonald, A. J., Pitts, M. C., Quantifying the role of orographic gravity waves on polar stratospheric cloud occurrence in the Antarctic and Arctic, J. Geophys. Res. Atmos., 118, 11493-11507, doi:10.1002/2013JD020122, 2013.

Dörnbrack, A., Leutbecher, M., Relevance of mountain waves for the formation of polar stratospheric clouds over Scandinavia: A 20 year climatology, J. Geophys. Res., 106, D2, 1583-1593, 2001.

**Response to Anonymous Referee #2**

Included below are our responses to the comments from Anonymous Referee #2 on our ACPD paper. The specific referee comments are given in bold italics and our responses in plain text. We thank the referee for their insightful comments which have resulted in a much improved manuscript.

***Specific Comments:***

***P2, L20 'sedimentation of large NAT particles (Molleker et al., 2014)':***
***Please cite also the original paper(s) referring to the large NAT particles as reason for the denitrification.***

This was an oversight. We have added citations to both the Fahey et al. (2001) and Northway et al. (2002) papers.

***P5, L16 'interpolated to the CALIOP PSC orbit grid using a weighted average of the two nearest MLS profiles': How has this weighting been performed?***

The MLS gas species data are interpolated to the CALIOP orbit grid using an average of the two nearest profiles weighted by the inverse of their relative distance from the CALIOP grid point. We have added this additional information to the text.

***P5, L28 'The data are then corrected for molecular and ozone attenuation using the MERRA-2 molecular and ozone number density profiles': Are the ozone profiles from MERRA-2 or those from the MLS retrievals? If MERRA, how strong does it affect the preprocessing if the 'real' (MLS) profiles would be used?***

The ozone attenuation correction uses MERRA-2 ozone number density profiles. Since 2004, MERRA-2 assimilates Aura MLS profile and OMI total column ozone which has resulted in a marked improvement in the MERRA-2 ozone product. During the period 2004-2014, the differences between MERRA-2 and ozonesonde values are around 5% (Gelaro et al., 2017; Wargan et al., 2017). Comparisons of Aura MLS profile ozone products with ground-based lidar and ozonesonde measurements from sites across the globe also indicate agreement within 5% or better (Hubert et al., 2016). So it appears that the MERRA-2 ozone product is of similar quality as Aura MLS. Therefore, changing the source of ozone from MERRA-2 to MLS would have negligible impact on the ozone attenuation correction. Also note that typical column ozone optical depths in the stratosphere are only on the order of 0.02, so the reduction in two-transmittance due ozone attenuation is about 4%. Therefore, even if one assumes an error in ozone abundance as large as 10% (which is double the uncertainty of MERRA-2), the error in the ozone attenuation correction would only be about 0.5%.

***P6, L11: Is the second equation really correct, or should it read: beta'_perp = beta'_perp,meas – beta'_par x CT ?***

This was a typographical error and the equation should read as the referee suggested. This has been corrected in Appendix A in the revised manuscript.

**P6, L17 '[unc_par] . . . [unc_perp]':**
**1. 'par' should everywhere be exchanged by 'para' not to confuse it with 'particles'**

To avoid confusion, we use the subscript "∥" to denote parallel, the subscript "⊥" to denote perpendicular, and the subscript "particulate" to denote particulate in the revised manuscript.

**2. 'unc' is a strange variable. Could it not be called delta_beta', so that the direct connection with beta' becomes clear?**

In the revised manuscript, we adopted the ISO naming convention for measurement uncertainty in quantity x, i.e. u(x).

**P7, L26 'is fixed at 10 cm−3': Could you provide a reason for this value (e.g. a citation) and how large its variation could be?**

Based on in situ data on total particle concentrations in the polar stratospheres (Wilson et al., 1990; Campbell and Deshler, 2014; Weigel et al., 2014), we feel that 10 $cm^{-3}$ is a good choice for "nominal" conditions and that 5 $cm^{-3}$ and 15 $cm^{-3}$ represent reasonable bounds on the variability. We have cited these papers in the revised manuscript.

**P9, L21 'Points with CI_NS > 1 are presumed to be PSCs containing non-spherical particles.': This seems to be a 2-sigma limit – could you describe more clearly what it means in percentage of the whole data points: how many of the 'non-spherical' particles might be spherical and vice-versa?**

$CI_{NS} > 1$ is roughly equivalent to $\beta_\perp$ values more than 2 median deviations larger than the median $\beta_\perp$ for the background aerosol. Histograms of $\beta_\perp$ for v2 STS and NAT mixtures overlap somewhat and suggest that 10-15% of data points in either class could be misclassified. We have added text accordingly to the revised manuscript.

**P9, L28 'detected through gas-phase uptake of HNO3 as observed by MLS': Regarding your analysis of volume-density later in the manuscript: can you confirm, that the amount of HNO3 uptake seen by MLS is in accordance with the detection limit of CALIOP?**

Simulations in Lambert et al. (2016) indicate that the amount of $HNO_3$ uptake observed by MLS in sub-visible PSCs is consistent with the detection limits of CALIOP. Figure 12 of that paper shows that very low number density ($5 \times 10^{-5}$ $cm^{-3}$) NAT particles remain undetectable by CALIOP throughout their growth history even though there may be uptake of several ppbv $HNO_3$ into the particles. Therefore, CALIOP may not detect NAT during the same time that MLS observes a substantial depletion of gas-phase $HNO_3$. On the other hand, higher number density ($5 \times 10^{-4}$ $cm^{-3}$) NAT particles become detectable by CALIOP and via MLS HNO3 gas phase depletion on similar time scales. We note in the manuscript that all derived quantities

(including VD) for non-features (including sub-visible PSCs) are retained in the v2 data product. So it is possible to compare our estimates of VD with the $HNO_3$ uptake observed by MLS in the sub-visible PSCs analyzed by Lambert et al., (2012; 2016), but we have not done this comparison.

*P12, L18 '8.3%':  In Fig. 10 the number seems 5.8% (?)*

This is a typographical error- 5.8% is the right number.  We have corrected this in the revised manuscript.

*P12, L17..:  Apart from the referenced modelling work by Zhu et al., 2018, which additional arguments are there to support the strong increase in the detected ice-PSCs in v2 compared to v1?*

As discussed by Pitts et al. (2013), a clear deficiency of v1 was the misclassification of ice as NAT mixtures under denitrified/dehydrated conditions because the NAT/ice boundary was based on a fixed abundance of 10 ppbv $HNO_3$ and 5 ppmv $H_2O$.  Therefore, the strong increase in ice detections in v2 is expected since it uses a much more appropriate NAT/ice boundary based on MLS observed abundances of $HNO_3$ and $H_2O$.  Additional supporting evidence for the v2 approach is provided by Spang et al. (2018), who show excellent agreement between CALIOP v2 and MIPAS areal coverage of ice PSCs over the 2009 Antarctic season.

*P13, L17..:  Is the explanation for the bimodal distribution of NAT-mixtures also confirmed by any modelling work which could be referenced here?*

We are not aware of any specific modeling studies that have examined the bimodal distribution of NAT mixtures.  However, in Pitts et al. (2013) we explored the source of this bimodality by examining the temperature histories along ten-day backward isentropic trajectories initiated at CALIOP NAT mixture observation points during 2009-10 Arctic winter.  The trajectories were produced using the Chemical Lagrangian Model of the Stratosphere (CLaMS) (McKenna et al., 2002) with wind and temperature fields from the European Center for Medium-range Weather Forecasts (ECMWF) analyses. From these temperature histories, we were able to estimate the time the air parcel arriving at each CALIOP NAT mixture observation point was exposed to temperatures below $T_{NAT}$.  The results clearly indicated that NAT mixture observations that occur at temperatures near NAT equilibrium had been exposed to temperatures below $T_{NAT}$ for much longer time periods than the non-equilibrium NAT mixtures that were observed at colder temperatures closer to $T_{STS}$.

*P14, L22 'persisting until early October':  From your Fig. 12, PSC are also often seen until mid and even end of October.*

The referee is correct that PSCs often are observed by CALIOP until mid and even end of October.  We have modified the text to reflect this.

*P14, L30..:  Is there any possibility to distinguish the PSCs from upper tropospheric cirrus (e.g. in terms of volume/surface density?  Do they 'separate' when plotting against potential temperature, if it is mainly caused by an upward displacement of isentropic surfaces, a separation might be possible.*

This is an interesting topic that we want to explore further. In particular, we would like to examine particle characteristics in the polar upper troposphere/lower stratosphere when deep cloud decks are present. Given the likely sharp gradients in condensables in this region, we suspect the particle characteristics could vary significantly. In very limited analyses, we have not seen a separation of tropospheric cirrus and PSCs when plotted against potential temperature. Although beyond the scope of the present paper, we plan on pursuing this in future research.

*P20, L10 'Then, we integrated the occurrence frequencies': Does 'integrated' just mean 'summed up'?*

Yes the occurrence frequencies are compiled at 1-km vertical increments and just summed over altitudes between 14 and 30 km. So a more representative descriptor would be 'total column occurrence frequency'- we have changed this in the manuscript.

*P20, L16: 'However, note that the SAM II occurrence frequencies are higher than those of CALIOP early in the PSC season.' May this difference in the early PSC period also be caused by how the sightings in case of SAM II are counted? During this time, there is a clear maximum at around 18 km and a minimum below. As SAM II is a limb-sounder, the sightings below might be influenced by the higher PSCs through which the SAM II line-of-sight passes. Has this been taken into account in the comparison?*

This is a good point. Although the occultation weighting functions are highly peaked near the tangent altitude, it is likely that at least to some extent an overlying cloud layer will impact the retrieved extinction values at lower tangent altitudes, with the cloud layer smeared out below the true level of occurrence. We don't think that this is a large effect, but it could increase the cloud occurrence frequency below the true level of maximum occurrence. We don't know of any quantitative approach to account for this in our statistics. Note that based on more comprehensive analyses presented in the revised Section 6, SAM II does appear more sensitive to tenuous clouds such as would be observed early in the season.

*P20, L17: 'This may be a reflection of the greater sensitivity of the limb-viewing occultation measurements to the onset of PSCs when liquid droplets first began to deliquesce and/or when low number density NAT particles form that are below the CALIOP detection thresholds'. How far south are the soundings by SAM II in mid-May? Aren't the 'sub-visible' PSCs identified by MLS more in the centre of the vortex where SAM II has not been observing?*

PSCs observed by SAM II early in the season were at lower latitudes than the sub-visible PSCs seen by MLS early in the season near the center of the vortex, so they are not the same PSCs. However, the principle is the same, i.e., both SAM II and MLS had/have greater sensitivity to tenuous PSCs than CALIOP. When the sensitivity of SAM II is adjusted to mimic that of CALIOP as we describe in Section 6 in the revised manuscript, there is no statistically significant difference between the two instruments in early season PSC sighting frequencies, so we have deleted the text referring to this difference in the revised manuscript.

*P22, L24: 'The estimates assume liquid particles (binary H2SO4-H2O or STS) only and thus have large uncertainties when NAT mixtures or ice are present' It would be helpful for the reader to repeat at this point that in case of NAT or ice the values are very probably lower limits of surface and volume density.*

We have added a sentence in both Sections 5 and 7 reiterating that our estimated SAD is likely an upper (lower) limit for the actual SAD in NAT mixture (ice) PSCs, and that our estimated VD is likely a lower limit for the actual VD in ice PSCs and in most NAT mixtures.

*P33, Fig. 3: To better understand the differences between the new and the old composition classification, either the separation lines of v2 should be included in Fig. 3 or those of v1 in Fig. 4.*

Fig. 3 (v1 optical space) now includes a new curve indicating the separation between enhanced NAT mixtures and NAT mixtures in v2. The new curve is noted in the figure caption and is also mentioned in the revised manuscript.

*P33, Fig. 3, caption: 'and symbol sizes are proportional to NAT' Does 'size' mean the symbol area or the diameter?*

Since particle size is not an important topic for either Fig. 3 or Fig. 4 (as noted by Anonymous Referee #3), we have made all symbols in these figures the same size to eliminate any confusion.

*P44, Fig. 14: The colours are partly difficult to distinguish, could different symbols be used, at least for the case of similar colours?*

Figs. 16 and 23 in the revised manuscript now use different symbols and colors so that individual years can be distinguished more easily.

*P54, Fig. 24: Please plot the results from both instruments in one graph. As it is now, the comparison in relation to the text is difficult.*

Results from CALIOP and SAM II are now plotted in the same graphs, one for the Antarctic (Fig. 32) and one for the Arctic (Fig. 33).

*Technical:*

*P2, L5: 'of season' -> 'of the season'*

Changed.

*P6, L9,11: numbering of equations missing*

Changed.

*P14, L20: 'Hence, it not' -> 'Hence, it is not'*

Changed.

*P16, L17: 'from the near' -> 'from near'*

Changed.

*P17, L24: 'about the pole' -> 'around the pole'*

Changed.

*P20, L6: delete blank between 'degree' and 'N' or 'S'*

Done.

*P21, L26: 'that more are more' -> 'that are more'*

Changed.

*P32, Fig. 2 caption: 'in yellow': in Fig. 1 this appears more light green than yellow*

We remade the figure and highlighted the orbit of interest in green.

*P41, Fig. 11 caption: '1 Antarctic' -> '12 Antarctic' (?)*

Changed.

*P43 and P49: color bar legend: '(x 10ˆ6 km^2)' should read either '(10ˆ6 km^2)' or '/(10ˆ6 km^2)'*

Changed to '(10^6 km^2)'

**Response to Anonymous Referee #3**

Included below are our responses to the comments from Anonymous Referee #3 on our ACPD paper. The specific referee comments are given in bold italics and our responses in plain text. We thank the referee for their insightful comments which have resulted in a much improved manuscript.

*Minor comments:*

***To my mind the manuscript is missing a short section/paragraph on comparisons with PSC measurements of other sensors. Some references on comparisons are given at various places of the paper, but I recommend to summarise the comparisons results at one specific place of the manuscript. This would better highlight the quality and reliability of the PSC detection and classification methods of the CALIOP instrument.***

Section 6 now includes a summary of comparisons of CALIOP with MIPAS PSC results and ground-based lidar PSC observations at McMurdo and Ny-Ålesund, as well as a more comprehensive quantitative comparison with the historical SAM II PSC record.

***I would recommend to move 3.1.1. and 3.1.2 into an appendix. A reduction on technical details in section 3 would be desirable for none-expert readers.***

We agree with the referee's recommendation, we have moved the material in these sections to Appendix A. We also added a figure to the appendix in response to questions from Anonymous Referee #1 about changes in crosstalk.

***Page 9, line 5: I am wondering that the MIPAS observations show a NAT belt on 2008- 05-29 and 2008/06-01/02 but no indication on May 30. Usually the NAT belt is devolving slowly over a couple of days starting with a small area of NAT/ice activity followed by a downstream formation of a belt-like structure in the next days (e.g. Höpfner et al., 2006). Please clarify, if May 30 is really NAT-free (maybe a typo?) in the MIPAS observations. Is it possible that MIPAS just misses the small NAT area from the day before due to a sampling issue? This potential mismatch may bias your definition of the empirical sub-class of 'enhanced NAT mixtures'.***

MIPAS data from 2008-05-30 showed the presence of NAT in just one observation at one altitude, so we originally grouped those data with data from 2008-05-27/28 (which showed zero NAT observations) to characterize observations without the NAT belt. We repeated our analysis excluding CALIOP data from 2008-05-30 altogether and obtained results nearly identical to our original analysis. So, we conclude that our results as presented in the paper are robust. We did discover that our description of CALIOP NAT mixture statistics for 2008-05-27/28/30 was not correct in the original manuscript. It now reads

as follows: "… only about 2% of CALIOP NAT mixture data from those days had $R_{532}>2$ and $\beta_\perp>2\times10^{-5}$ km$^{-1}$sr$^{-1}$" instead of the original version "…about 98% of CALIOP NAT mixture data from those days had $R_{532}\leq2$ and $\beta_\perp\leq2\times10^{-5}$ km$^{-1}$sr$^{-1}$."

*Section 3.6: The percentages of Figure 10 suggests that mainly enhanced NAT mixtures of Version 1 are classified in Version 2 as ice. Is this correct, or is the effect caused by misclassification of the former Mix-2 and Mix-2enhanced classes. Can you quantify the partitioning between the two V1 classes into the V2 ice class?*

Further analysis indicates that about 75% of the additional ice PSCs in Version 2 come from a reclassification of Version 1 Mix2-enhanced PSCs, and the remaining 25% come from a reclassification of Version 1 Mix1 and Mix2 PSCs. The text has been modified accordingly.

*Section 3.7: The temperature difference between STS and ice in the T −Tice histogram for the maximum position (∆T 1-1.5K) looks unexpectedly small to me. I would expect from the equilibrium curves, for example presented in Fig. 5 of Pitts et al. (2013), higher temperatures for STS. Can you please clarify and/or explain in more detail how you defined TSTS based on Carslaw et al. (1995).*

The roughly 1.5 K difference between the peaks of the STS and ice histograms in T-T$_{ice}$ space is a reflection of the denitrification that occurs during both Antarctic and Arctic winters. For example, Fig. 5(a) of Pitts et al. (2013) shows that the STS equilibrium curve lies about 3.5 K above T$_{ice}$ at 15 ppbv HNO$_3$, but actually approaches T$_{ice}$ at very low HNO$_3$ mixing ratios. T$_{STS}$ is defined using Carslaw et al. (1995) with equilibrium HNO$_3$ and H$_2$O mixing ratios equal to those observed by MLS.

*Page 13, line 8: Is the 'strong' statement regarding the positive tail in the PSC distribution (that this is due to warm biased temperatures associated with wave ice events not fully resolved in MERRA-2 fields) based on a detailed analysis or 'only' one plausible explanation. Uncertainties of the threshold lines between Ice and NAT may cause a similar tail in the distribution. Please commend and clarify.*

Pitts et al. (2011) found that anomalously warm temperatures in ice PSC distributions from the 2009-10 Arctic winter were associated with wave ice events. Given that the MERRA-2 meteorological analyses are computed on a grid with horizontal resolution of approximately 0.5° x 0.625° (55.55 km x 23.75 km at 70° latitude), it is likely that small scale temperature perturbations will not be fully resolved. Hoffman et al. (2017) found that MERRA-2 reanalyses were only able to reproduce about 30% of the standard deviations of the temperature fluctuations observed on long-duration superpressure balloon flights over Antarctica. So it is possible that at least part of the warm tail is associated with poorly resolved temperature perturbations associated with wave ice events. But as the referee correctly points out, it is also plausible that misclassification of NAT mixtures as ice due to measurement noise could produce a similar tail. Overall there are not a lot of anomalously warm ice observations, so it is not a significant issue. We have reworded the text to offer both possibilities as plausible explanations for the positive tail.

*The authors may think about to skip Figure 16, which is partly redundant to Fig. 17. For example, Fig. 17 includes by far more quantitative information than Fig. 16 due to the choice of the vertical and horizontal coordinates.*

We agree with the referee that there is some redundancy between original Figs. 16 and 17. However, we have decided to not skip the former for the benefit of readers who are more comfortable with geographic coordinates and because including both figures clearly illustrates that more quantitative information comes directly from the coordinate transformation.

*Section 6: To my mind the SAM II - CALIPSO comparison would profit by some more detailed descriptions and analyses. The information on SAM II measurements are very limited. For a profound comparison of the PSC occurrence frequencies it would be necessary to discuss the detection limits of both instruments (I guess based on extinction or density thresholds). The authors should discuss similarities and differences between the two sensors as well.*

Section 6 now includes a more detailed description of the SAM II measurements and how the CALIOP PSC record was sub-sampled to mimic the SAM II measurement geometry. There is also a comprehensive discussion of the steps we took to put the two data sets on equal footing to establish their relative sensitivities before comparing their multi-year PSC sighting statistics.

*Technical corrections:*

*page 14, line 20: 'Hence, it is not ...'*

Changed.

*p21, L7: please explain 'DMPs'*

The Version 2 (v2) derived meteorological products (DMPs) (Manney et al., 2007; Manney et al., 2011a) consist of gridded meteorological data and derived products, such as equivalent latitude and vortex edge location, that are calculated from the GEOS5-based MERRA-2 reanalyses and interpolated to the locations and times of the MLS measurements. The Aura MLS Derived Meteorological Products (DMPs) acronym is defined in Section 2, first paragraph. A brief description of the DMP data is provided in Section 2.2.

*P23, L26: For completeness the authors may like/need to add a reference to the SAM II dataset as well.*

Good point, we've added the appropriate references to the SAM II dataset.

*page 31/32; Fig. 1/2 caption: CALIOP curtain of Fig. 2 looks on my printout and screen greenish and not yellow. Please check.*

We remade the figure and highlighted the orbit of interest in green.

*Figure 3: 'The symbols size are proportional to volume-equivalent radii of NAT and ice'. This fact is hard to see in Figure 3 and may cause the effect, that the Mix1 calculations are hiding all STS results. Is the particle size an important topic for this figure? If not, keeping the figure more simple the interpretation of the figure might be easier for the reader. Is the the particle radius also an issue in Figure 4? If yes, this is not obvious from the caption and the text passages in the corresponding section.*

Particle size is not an important topic for either Fig. 3 or Fig. 4, so we have made all symbols in these figures the same size. STS results are now clearly visible in Fig. 3, but are off scale in Fig. 4 because theoretical $\delta_{particulate}=0$ for spherical particles.

*Figure 4: The authors may explain the grey box in the figure caption (S/N issue) or reference to the details in the corresponding section.*

We have added an explanation for the grey box in the Figure 4 caption.

*Figure 18: Starting at 5% occ. freq. with the colour bar looks a bit extreme. Especially, if this is leading to the strong statement of section 4.1.2 'with essentially no STS' occurrence in the the deep vortex during September. Please clarify, if this statement is an 'artefact'.*

We remade the original Figure 18 (renumbered Figure 20) using 1% occ. freq. as the minimum contour. The original minimum contour of 5% may have been misleading in terms of STS occurrence, although the large difference between STS and NAT mixtures is a robust feature. We have modified the text describing (renumbered) Figure 20 accordingly to state that STS observations are widespread during June at this level, but more limited afterwards with occurrence frequencies generally less than 10% in the interior of the vortex during July-August and less than 5% during September. NAT mixtures, on the other hand, are relatively widespread over much of the vortex at this level, especially during July and August when occurrence frequencies exceed 35%.

There is a solid black contour line on each of the color panels in original Figure 15 (renumbered Figure 17) that corresponds to the range of days and altitudes where PSCs (of any composition) were observed in at least six of the twelve Antarctic seasons. This provides an indication of the climatological temporal and altitude extent of the PSC season. For STS and NAT mixtures (Figs 17a-b), PSC onset in at least six years occurred by approximately 20 May. The onset of ice PSCs (Fig. 17c) is delayed until temperatures drop below the frost point which is typically mid-June. The contour is primarily included to illustrate when the onset of ice occurs relative to the other compositions. We have revised the caption to Figure 17 to better explain this contour.

[revised manuscript text omitted]